# Evaluation of natural aerosols in CRESCENDO-ESMs: Mineral Dust

Ramiro Checa-Garcia[1], Yves Balkanski[1], Samuel Albani[8], Tommi Bergman[5], Ken Carslaw[2], Anne Cozic[1], Chris Dearden[10], Beatrice Marticorena[3], Martine Michou[4], Twan van Noije[5], Pierre Nabat[4], Fiona M. O'Connor[7], Dirk Olivié[6], Joseph M. Prospero[9], Philippe Le Sager[5], Michael Schulz[6], and Catherine Scott[2]

[1]Laboratoire des Sciences du Climat et de l'Environnement, CEA-CNRS-UVSQ, IPSL, Gif-sur-Yvette, France
[2]Institute for Climate & Atmospheric Science, School of Earth & Environment, University of Leeds, United Kingdom
[3]LISA, Universités Est-Paris & Diderot-Paris, France
[4]CNRM, Université de Toulouse, Météo-France, CNRS, Toulouse, France
[5]Royal Netherlands Meteorological Institute, De Bilt, Netherlands
[6]Norwegian Meteorological Institute, Oslo, Norway
[7]Met Office Hadley Centre, Exeter, United Kingdom
[8]Department of Environmental & Earth Sciences, University of Milano-Bicocca, Italy
[9]Department of Atmospheric Sciences, University of Miami, USA
[10]Centre for Environmental Modelling & Computation (CEMAC), School of Earth & Environment, University of Leeds, UK

**Correspondence:** Ramiro Checa-Garcia (ramiro.checa-garcia@lsce.ipsl.fr)

**Abstract.** This paper presents an analysis of the mineral dust aerosol modelled by five Earth System Models (ESMs) within the Coordinated Research in Earth Systems and Climate: Experiments, kNowledge, Dissemination and Outreach (CRESCENDO) project. We quantify the global dust cycle described by each model in terms of global emissions, together with, dry and wet deposition, reporting large differences in ratio of dry over wet deposition across the models not directly correlated with the range of particle sizes emitted. The multi-model mean dust emissions with 5 ESMs is 2836 $\mathrm{Tg\,yr^{-1}}$ but with a large uncertainty due mainly to the difference in the maximum dust particle size emitted. The multi-model mean of the subset of four ESMs without particle diameters larger than 10 $\mathrm{\mu m}$ is 1664 ($\sigma$=651) $\mathrm{Tg\,yr^{-1}}$. Total dust emissions in the simulations with identical nudged winds from reanalysis give us better consistency between models, i.e. this multi-model mean global emissions with 3 ESMs is 1613 ($\sigma$=278) $\mathrm{Tg\,yr^{-1}}$, but 1834 ($\sigma$=666) $\mathrm{Tg\,yr^{-1}}$ without nudged winds and same the models. Significant discrepancies in the globally averaged dust mass extinction efficiency explain why even models with relatively similar global dust load budgets can display strong differences in dust optical depth. The comparison against observations has been done in terms of dust optical depths based on MODIS (Moderate Resolution Imaging Spectroradiometer) satellite products, showing a global consistency in terms of preferential dust sources and transport across the Atlantic. The global localisation of source regions is consistent with MODIS, but we found regional and seasonal differences between models and observations when we quantified the cross-correlation of time-series over dust emitting regions. To faithfully compare local emissions between models we introduce a re-gridded normalisation method, that also can be compared with satellite products derived from dust events frequencies. Dust total deposition is compared with an instrumental network to assess global and regional differences. We find that models agree with observations within a factor of 10 for data stations distant from dust

sources, but the approximations of dust particle size distribution at emission contributed to a misrepresentation of the actual range of deposition values when instruments are close to dust emitting regions. The observed dust surface concentrations also are reproduced to within a factor of 10. The comparison of total aerosol optical depth with AERONET (AErosol RObotic NETwork) stations where dust is dominant shows large differences between models, although with an increase of the inter-model consistency when the simulations are conducted with nudged-winds. The increase in the model ensemble consistency also means a better agreement with observations, which we have ascertained for dust total deposition, surface concentrations and optical depths (against both AERONET and MODIS retrievals). We introduce a method to ascertain the contributions per mode consistent with the multi-modal direct radiative effects, that we apply to study the direct radiative effects of a multi-modal representation of the dust particle size distribution that includes the largest particles.

# 1 Introduction

Mineral dust is a key element of the Earth system. It plays an important role in our planet's energy budget, in both the long-wave (LW) and the short-wave (SW) spectrum, by direct radiative effects and feedbacks on the climate system (Knippertz and Stuut, 2014a). It also contributes significantly to the global aerosol burden. Kok et al. (2017), based on models and observations, estimated that global emissions are $1700 \, \mathrm{Tg \, yr^{-1}}$ (with a range between $1000\text{-}2700 \, \mathrm{Tg \, yr^{-1}}$ and particle diameters up to 20 µm) which indicates that mineral dust, together with sea spray, have the largest mass emission fluxes of primary aerosols. Furthermore, it is transported by the atmospheric flow from emission source regions to distant remote regions up to thousands of kilometres (Kaufman et al., 2005; Li et al., 2008). When it is deposited over the ocean (Schulz et al., 2012) dust constitutes a source of minerals, in particular iron (Wang et al., 2015; Mahowald et al., 2005; Mahowald, 2011) and phosphorus (Wang Rong et al., 2014), therefore it indirectly participates in the carbon cycle and the ocean removal of carbon dioxide from the atmosphere (Gruber et al., 2009; Shaffer et al., 2009). When dust is deposited over land it impacts on ecosystems (Prospero et al., 2020) and snow albedo (Painter et al., 2007). In the troposphere dust contributes to heterogeneous chemical reactions (Tang et al., 2017; Dentener et al., 1996; Perlwitz et al., 2015; Bauer, 2004) and ice nucleation (Tang et al., 2016; Atkinson et al., 2013; Hoose and Möhler, 2012; Prenni et al., 2009) but also behaves as cloud condensation nuclei (Bègue et al., 2015), presenting additional interactions with precipitation (Solomos et al., 2011). Air quality studies link dust concentrations with health effects (Monks et al., 2009) but also with visibility (Mahowald et al., 2007). Additionally, transport and deposition of dust plays a role in the design and maintenance of solar energy stations in semi-desert areas (Piedra et al., 2018), whereas at the Earth's surface fine dust particles (diameter smaller than 2.5 µm) can cause long-term respiratory problems (Pu and Ginoux, 2018a; Longueville et al., 2010). At regional scales dust has been reported to influence the West African (Strong et al., 2015; Biasutti, 2019) and Indian monsoons (Sharma and Miller, 2017; Jin et al., 2021).

**Table 1.** Main characteristics of the CRESCENDO models used in this study and the simulation experiments analyzed: PD (Present Day), PDN (Present Day with nudged winds), PI (Pre-Industrial aerosol and chemistry forcings). Resolution is given in degrees (longitude x latitude), and all dust emissions are interactively driven by wind speed. DPSD stands for Dust Particle Size Distribution, detailed information for each model is given in Supplement, Tables S.MD.8 and S.MD.9. To describe the modelling of largest particles we defined two classifiers: (D10) to differenciate those schemes that explicity aim to model diameters larger than $> 10 \mu m$. (BM20), if a specific bin or mode for particles larger than $20 \mu m$ is defined (Yes), is not included (Not) or is joint into a single mode/bin with smaller particles than $20 \mu m$ particles (Mix). $\kappa^{DUST}$ means the refractive index used for mineral dust aerosols. For additional information of the dust schemes and their implementation in the Earth System Models key References are given.

| Model Full-Name | Short-Name | Resolution | Levels | Experiments | DPSD | Large-Particles | | $\kappa^{DUST}$ | References |
| | | | | | | D10 | BM20 | Dust Refraction Index | |
| --- | --- | --- | --- | --- | --- | --- | --- | --- | --- |
| IPSL-CM6-INCA5 | IPSL | 2.50x1.25 | 79 | PD, PDN, PI | modes: 1 | No | No | $1.520 - i1.47 \cdot 10^{-3}$ | [1] |
| CNRM-ESM2-1 | CNRM-3DU | 1.40x1.40 | 91 | PD, PDN, PI | bins: 3 | Yes | No | $1.51 - i8.0 \cdot 10^{-3}$ | [2] |
| CNRM-ESM2-1-CRESC | CNRM-6DU | 1.40x1.40 | 91 | PD, PDN, PI | bins: 6 | Yes | Mix | $1.51 - i8.0 \cdot 10^{-3}$ | [2] |
| NorESM1.2 | NorESM | 1.25x0.94 | 30 | PD, PDN, PI | modes: 2 | No | No | $1.530 - i2.40 \cdot 10^{-3}$ | [3] |
| EC-Earth3-AerChem | EC-Earth | 3.00x2.00 | 34 | PD, PI | modes: 2 | No | No | $1.517 - i1.09 \cdot 10^{-3}$ | [4] |
| UKESM1 | UKESM | 1.87x1.25 | 85 | PD, PI | bins: 6 | Yes | Yes | $1.520 - i1.48 \cdot 10^{-3}$ | [5] |
| IPSL-CM6-INCA5-4DU | IPSL-4DU | 2.50x1.25 | 79 | Special PDN | modes: 4 | Yes | Yes | $1.520 - i1.47 \cdot 10^{-3}$ | [6] |

*Dust Schemes description*: (1) Schulz et al. (1998), (2) Michou et al. (2020), (3) Zender et al. (2003), (4) Tegen et al. (2002), (5) Woodward (2001), (6) Albani et al., 2020; in prep.

*Earth System Model description*: (1 & 6) Boucher et al. (2020), (2) Séférian et al. (2019), (3) Kirkevåg et al. (2018), (4) van Noije et al. (2020), (5) Sellar et al. (2019); Mulcahy et al. (2020).

As a consequence, the dust cycle is actively analysed on regional (Pérez et al., 2006; Konare et al., 2008) and global scales, based on observations and models, covering aspects related to optical properties, mineral composition, emission processes, transport and deposition (Tegen and Fung, 1994). Current global models represent reasonably well the atmospheric lifetime of dust particles with a diameter of less than 20 μm (Kok et al., 2017), supporting a consistent modelling of the dust atmospheric cycle: emission, transport and deposition. Very large dust particles with diameters of several tens of micrometers are, however, seldomly represented in these models, and have become an active area of research (van der Does et al., 2018; Di Biagio et al., 2020).

Detailed comparisons between observations and models indicate that the latter are not yet capturing the full dust spatial and temporal distribution in terms of its various properties. This is due to the fact that current Earth system models are limited to approximate phenomenological descriptions of the dust mobilisation (Zender et al., 2003). These dust emissions schemes are based on either a saltation process (Marticorena and Bergametti, 1995) or a brittle fragmentation model (Kok, 2011), but in both cases the momentum transfer between the wind in the boundary layer and the soil particles is conditioned by erodibility or surface roughness parameters, which sometimes are simply scaled to be in agreement with observations of aerosol index and/or aerosol optical depth. These constraints allow the models to reproduce reasonably well the dust optical depth (Ridley et al., 2016) but cannot fully constrain the whole range of the dust particle size distribution. This explains the considerable differences in surface concentrations and vertical deposition fluxes when global models are evaluated against dust observations

**Table 2.** CRESCENDO-ESM experiments analysed: PD (Present Day), PDN (Present Day with nudged winds), PI (Pre-Industrial aerosol and chemistry forcings). The sea-surface temperatures (SSTs) and ice cover are prescribed based on CMIP6-DECK-AMIP (Durack and Taylor, 2018). The solar forcing is using the input4MIPs dataset (Matthes et al., 2017) but NorESM uses the previous dataset. The gas and aerosol emissions are consistent with CMIP6 but depending on the complexity of the gas-phase species, ozone can be prescribed with either ozone concentrations from a previous full chemistry simulation or the input4MIPs ozone forcing dataset (Checa-Garcia et al., 2018; Hegglin et al., 2016). Wind fields used for the specified dynamics are obtained from re-analysis of ERA-Interim (Dee et al., 2011).

|  | PD | PDN | PI |
|---|---|---|---|
| Time Period | 2000-2014 | 2000-2014 | 2000-2014 |
| SSTs and ice cover | prescribed | prescribed | prescribed |
| Aerosol Precursors | Present-Day | Present-Day | 1850 |
| Anthropogenic Emissions | Present-Day | Present-Day | 1850 |
| Solar Forcing | Present-Day | Present-Day | Present-Day |
| Wind Fields | modelled | prescribed | modelled |

at regional and local scales. These challenges increase in regions with strong seasonal cycles and sparse vegetation cover, that require a description of the evolving vegetation, like the Sahel or semi-arid regions. Other difficulties emerge when the anthropogenic component of atmospheric dust has to be ascertained, as it requires land use change and agricultural activities to be considered. Optical properties of mineral dust aerosols are another field of research as both the refractive index and the particle shape introduce uncertainties on the estimation of scattering and absorption properties (Nousiainen, 2009). Finally, the total mass of mineral dust emitted to the atmosphere is mostly conditioned by a few events with intense surface winds, as the dust emission flux has a non-linear dependence on the wind speed, which the models pursue to capture. Actually, the meteorological phenomena conditioning these events exhibit regional dependencies, e.g. in West Africa deep convection (Knippertz and Todd, 2012) and nocturnal low-level jets (Heinold et al., 2013; Washington and Todd, 2005) have been found to be key drivers. Recently, Yu et al. (2019) reported differences in the frequency of dust events between the Gobi (very high frequency of dust events in March and April) and Taklamaklan (more than half of events from May to September) deserts, which can be interpreted by a larger role in dust activation of the nocturnal low-level jet in Taklamaklan (Ge et al., 2016).

The relevance of dust on the Earth system implies that most climate models have introduced parametrization schemes to describe properly the dust cycle in the last two decades. Woodward (2001) describes the parametrization implemented in the Hadley Centre climate model, Miller et al. (2006) introduces the NASA Goddard dust model, Schulz et al. (1998) and later Schulz et al. (2009) show the implementation of dust emissions in the INteraction of Chemistry and Aerosols (INCA) module of the IPSL model. Pérez et al. (2011) for the BSC-DUST model, and more recently other models either incorporate new dust schemes or improve on previous ones, e.g. Albani et al. (2014) and Scanza et al. (2015) in the CAM climate model, LeGrand et al. (2019) for the GOCART (Goddard Chemistry Aerosol Radiation and Transport ) aerosol model, Klingmüller et al. (2018) in the EMAC atmospheric-chemistry climate model, Colarco et al. (2014) in the NASA GEOS-5 climate model, Astitha et al. (2012) and Gläser et al. (2012) in the ECHAM climate model. Therefore comparisons to ascertain how the models are

**Table 3.** Observations used for the comparison of the CRESCENDO models against observations indicating the spatial and temporal scales considered. Loadings and Mass Extinction Efficiency (MEE) were derived from model results only and are compared between them. L=local, N=Network, G=Global, R=Regional, A=Annual, M=Monthly, CM=Monthly-Climatology, CA=Annual-Climatology, TS=Time-Series-Avaliable.

| Diagnostic | Dataset | Spatial | Temporal | Reference | Comments |
|---|---|---|---|---|---|
| Aerosol Optical Depth | AERONET | (L, N) | (A, M, TS) | (Giles et al., 2019) | Aeronet v3 |
| | MODIS | (G, R ) | (A, M) | (Sayer et al., 2014) | DeepBlue-v6 |
| | MISR | (G, R ) | (A, M) | (Diner et al., 2002) | |
| Ångström Exponent | AERONET | (L, N) | (A, M, TS) | (Giles et al., 2019) | Aeronet v3 |
| | MISR | (G) | (A, M) | (Diner et al., 2002) | |
| Dust optical depth | AERONET dusty | (L, N) | (A, M, TS) | (Giles et al., 2019) | Subset of AERONET |
| | MODIS DOD | (G, R) | (A, M) | (Pu and Ginoux, 2018b) | See Supplementary |
| | IASI dust | (G, R) | (A, M) | (Peyridieu et al., 2013) | Near-Infrared |
| Surface concentration | UMOAC | (L, N) | (CA, CM) | (Prospero and Nees, 1986) | Filter Collectors |
| | Mahowald-2009 | (L, N) | (CA) | (Mahowald et al., 2009) | |
| | INDAAF-PM10 | (L) | (TS, CA) | (Marticorena et al., 2017) | INDAAF dataset |
| Dust deposition flux | Network-H2011 | (N) | (CA) | (Huneeus et al., 2011) | Compilation dataset |
| | Network-SET-M | (N) | (CA) | (O'Hara et al., 2006; Vincent et al., 2016) | Compilation dataset |
| Wet/dry deposition flux | INDAAF-dep | (L) | (TS,CM) | (Marticorena et al., 2017) | INDAAF dataset |

improving the description of dust related processes are needed to make progress in the above challenges. A broad comparison of 15 AeroCom models (including both climate models and chemistry transport models) in terms of dust has been conducted by Huneeus et al. (2011) and more recently a comparison of dust optical depth in 7 CMIP5 (Coupled Model Intercomparison Project phase 5) climate models (Pu and Ginoux, 2018b). Albani et al. (2014) show a detailed comparison of several dust

schemes of the CAM climate model. However, as the evolution of ESMs and dust schemes continues, in parallel with the availability of longer and new/refined observations, exhaustive comparisons of dust cycles modelling, covering scales from the global to the local, are still needed.

This study aims to carry out an extensive comparison between observations and five Earth system models from the Coordinated Research in Earth Systems and Climate: Experiments, kNowledge, Dissemination and Outreach (CRESCENDO) project

which aims to develop the current European ESMs through targeted improvements to a range of key processes, in particular natural aerosols and trace gases. We compare the ESMs against observations in terms of optical properties (dust optical depth, Ångström exponent), surface concentration, wet and dry deposition, and dust emissions, and how these aspects evolve in time and space. The paper is structured as follows: Sect. 2 describes the models analysed, which is followed by Sect. 3 describing the observational datasets used, and the methods (Sect. 4). The results of the comparison are presented first at the global scale

(Sect. 5.1), showing also its climatological spatial patterns (Sect. 5.2), followed by sections describing: dust emission (Sect. 5.3), dust deposition (Sect. 5.4), dust optical depths (Sect. 5.5) and surface concentrations (Sect. 5.6). These results are then discussed in Sect.6 where the main conclusions are also summarised. Our final summary of future research recommendations

is in Sect. 7. The supplementary information is a single document but organised according with the several sections of the main paper: Supplement MD has additional information in sections 2 (models) and 3 (datasets). Supplement GL complements Sect. 5.1. The other supplement parts refer to each of the diagnostics analysed.

## 2 Models description

Five different Earth System Models (Table 1) constitute the CRESCENDO-ESM ensemble: CNRM-ESM2-1, NorESM1.2, EC-Earth3-AerChem, IPSL-CM6-INCA5 and UKESM1 with 2 different dust schemes for CNRM-ESM2-1 and IPLS-CM6-INCA5 (hereafter we refer to each model with the short-names in Table 1). This ensemble covers the two main methods to describe the dust particle size distribution: binned/sectional and multi-modal log-normal.

In the sectional methodology the full size distribution is divided into a fixed number of bins, while inside each bin the size distribution is considered invariant. For CNRM-ESM2-1 two different dust schemes based on two different sets of bins have been evaluated, see Table S.MD.8 for further details, named here CNRM-6DU (with 6 bins) and CNRM-3DU (with 3 bins). The UKESM model includes 6 bins, with both UKESM and CNRM-6DU covering also particles with diameters larger than 20μm, with two bins in the case of the UKESM model and one bin in the case of the CNRM-6DU model.

In the case of modal description the evolution of the size distribution is controlled by balance equations of mass and number concentrations of each mode, as they effectively constrain a log-normal distribution with fixed width. In CRESCENDO there are two main approaches: EC-Earth and NorESM are considering bi-modal size distributions (with one fine/accumulation mode and one coarse mode) mixed with other aerosols, whereas IPSL is considering an externally mixed single dust coarse mode (see Table S.MD.9). The limit between coarse and fine particles is located at about 1 μm (while accumulation refers to fine particles from 0.1 μm to 1 μm). Denjean et al. (2016) aimed to estimate the typical parameters of a multi-modal description of the dust size distribution but confined to the range of sizes typical of accumulation and coarse modes. Recent experiments are also including larger particles (Ryder et al., 2018, 2019). A new analysis by Adebiyi and Kok (2020) proposes that the coarse mode, and more specifically those particles with diameter larger than 20 μm are important to better understand the global dust cycle (often referred to as super-coarse and giant dust particles). Therefore, we also compared the CRESCENDO-ESMs modal dust schemes, with a new dust scheme of the IPSL model with 4 insoluble dust modes (Albani and et al, 2021) whose properties are based on the FENNEC campaign (Rocha-Lima et al., 2018; Di Biagio et al., 2020). Table S.MD.9 shows the modal approaches in CRESCENDO, and how they compare with the IPSL-4DU.

To better describe the CRESCENDO ensemble diversity in the modelling of the coarse mode (large particles), two classifiers are introduced in Table 1: one to differentiate those dust schemes that aim to include particles with diameters larger than $10\mu m$, and the other one to indicate whether the model explicitly has a bin/mode for particles with diameters larger than $20\mu m$.

All the models provide standard approaches that estimate dust mobilisation based on a velocity threshold, information on soil texture (clay/silt), erodibility factors (including soil moisture or accumulated precipitation) and prescribed vegetation cover. Conceptually, a fraction of the horizontal flux of dust particles, dominated by sandblasting, is actually transformed into a vertical flux with a mass efficiency factor and then effectively transported by the atmosphere. EC-Earth emissions are

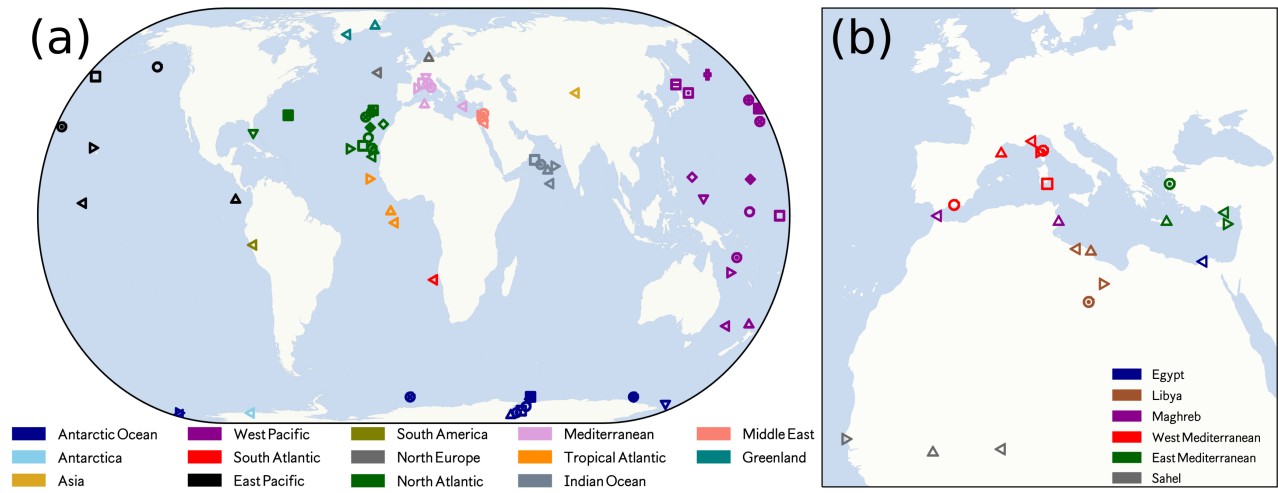

**Figure 1.** Panel (a): Map with the stations of the dataset named Network-H2011 which collects annual dust deposition fluxes for multiple years (Huneeus et al., 2011). Panel (b): Map with the stations of the dataset named Network-SET-M which collects additional station data in the Mediterranean region where observations have been reported by O'Hara et al. (2006) and Vincent et al. (2016), and station data over the Sahel (Marticorena et al., 2017). The different colours represent the region where each station belongs to.

calculated following the scheme described by Tegen et al. (2002) based on the horizontal dust flux proposed by Marticorena and Bergametti (1995), which is also used in the UKESM dust scheme (Woodward, 2001). The NorESM emissions are estimated with the Dust Entrainment And Deposition (DEAD) model (Zender et al., 2003). The IPSL dust emission has been described by Schulz et al. (2009, 1998), and the CNRM-3DU model (Nabat et al., 2012) used also Marticorena and Bergametti (1995)

with an emitted size distribution based on Kok (2011), while the CNRM-6DU is a revised version of the CNRM-3DU dust scheme.

Although none of the models have implemented an explicit mineralogical description of dust particles, the optical refractive index effectively accounts for global average of the mixture of minerals present in the mineral dust aerosol. Therefore, those optical properties are representative for the global mineralogical composition rather than a description of the soil-type depen-

145 dence of the mineralogy that would imply local differences on emitted optical properties. This approximation is considered to drive specific biases in those regions where the fraction of hematite or goethite minerals induce larger values of optical absorption, as shown by (Balkanski et al., 2007, 2021). The refractive index, expressed as $\kappa^{DUST} = n - ki$, of each model is shown in Table 1. They have similar values for the real component, but the imaginary component, although small, can be different by a factor of 2 which implies discrepancies in mass absorbing efficiency. Beyond the refractive index, the optical model used to

150 estimate the key optical properties is another factor of diversity.

In all the models the particle size is described by the geometric diameter, where the dust particles with irregular shapes are modelled by spherical particles with the same effective volume. Regarding optical properties they are calculated based on Mie scattering, this approximation is reasonable as far as the orientation of the particles is randomly distributed, but any

physical process that breaks this hypothesis, like preferential transport of specific geometries or physical processes that promote a specific orientation of the particles, will imply a bias in the methodology. The geometry of the particles also affects the gravitational settling, and therefore the transport of particles with specific geometries (Li and Osada, 2007) and their lifetime in the atmosphere. Recently, Huang et al. (2020) have estimated that the asphericity increases gravitational settling lifetime by 20% for both fine and coarse modes. Additionally, the spherical approximation is considered to underestimate the optical extinction of mineral dust (Kok et al., 2017). This hypothesis also affects the actual area of the global mineral dust surface which is important in heterogeneous chemistry (Bauer, 2004) and influences tropospheric chemistry.

## 2.1 Model experiments

Because the models have interactive dust emissions, wind fields play a prominent role in dust emission and transport (Timmreck and Schulz, 2004). Therefore, this study contrasts two different present-day forcing experiments: one with winds generated by the dynamical part of the climate model (named PD), and the other nudged to re-analysed winds (named PDN) from ERA-Interim (Dee et al., 2011). The historical greenhouse gases concentrations are consistent with (Meinshausen et al., 2017). The models IPSL and IPSL-4DU were run without explicit gas-phase interactive chemistry activated, therefore they use the CMIP6 ozone forcing database (Checa-Garcia et al., 2018). The CNRM-ESM2-1 has explicit chemistry in the stratosphere and upper-atmosphere (Michou et al., 2020). A last simulation where aerosols and chemistry emissions are prescribed for 1850 (named PI) is presented as well, see Table 2. All the simulations are from 2000 to 2014 plus at least 1 year of spin-up (except NorESM-PDN that covers 2001 to 2014). All the simulations implement prescribed sea surface temperatures (SSTs) of present-day conditions according to input4MIPs dataset (Durack and Taylor, 2018). The solar forcing implemented by all the models is derived from the dataset of Matthes et al. (2017). The comparison between the PD and PDN experiments inform about the role of wind fields to explain model diversity. The difference between PD and PI dust emissions allow us to evaluate whether the effects in the climate system due to non-dust emissions have a discernible impact on the global dust cycle (as both PD and PI have been prescribed with the same SSTs). A summary of the properties of the model experiments is given in Table 2.

## 3 Observational datasets

The observational data-sets used to assess the performance of the CRESCENDO ESMs in their representation of mineral dust are based on a compilation of ground-site and satellite measurements. Table 3 summarises the different available datasets used, and the spatial and temporal scales applied in the analysis. Additionally, this table includes datasets representative of either a monthly or a yearly climatology (respectively referred to as CM and CA in Table 3). In this section these datasets are briefly described, but we refer to the original publications for further details. For those datasets with specific pre-processing the additional details are given in the supplementary material.

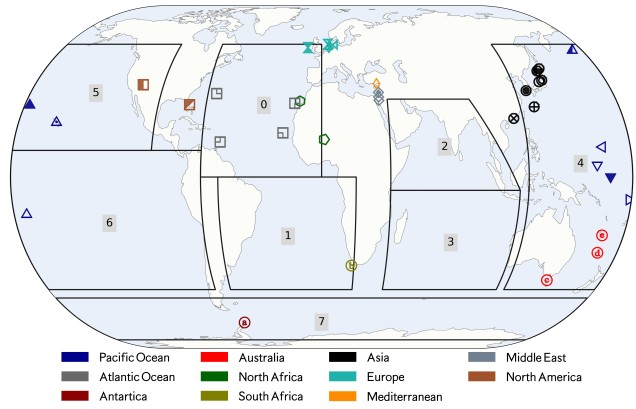

**Figure 2.** Map with 36 stations where surface concentrations were monitored by UMOAC (University of Miami Oceanic Aerosols Network) and also those described by Mahowald et al. (2009). Colours represent the region where each station belongs to. The regions correspond to those used for the regional analysis of dust deposition over the ocean: North Atlantic (0), South Atlantic (1), North-Indian Ocean (2), South-Indian Ocean (3), Pacific West (4), Pacific North-East (5), Pacific South-East (6) and Antarctic Ocean (7). For each of the oceanic regions a land-mask is also applied to filter inland grid-cells.

## 3.1 Surface Deposition flux

This dataset comprises deposition flux observations described in Huneeus et al. (2011), composed from several measurement campaigns over land and ocean (Figure 1 panel a), and named hereafter *Network-H2011*, plus an additional set of measurements at stations in the Mediterranean and Sahel regions (Figure 1 panel b), named hereafter *Network-SET-M* for which data values are shown in the Table S.MD.5.

The set *Network-H2011* gives deposition fluxes estimated from sedimentation corresponding to the DIRTMAP (Dust Indicators and Records of Terrestrial and MArine Palaeo-environments) database (Kohfeld and Harrison, 2001), while direct measurements of deposition fluxes were acquired during the SEAREX campaign (Ginoux et al., 2001) mostly in the Northern Hemisphere. Mahowald et al. (2009) describes 28 sites where dust deposition is inferred assuming a 3.5% fraction of iron. The compilation also includes observations of deposition fluxes deduced from ice core data according to Huneeus et al. (2011). The dataset covers a range of total dust flux deposition from $10^{-3}$ to $0.5 \cdot 10^3 \, \mathrm{g\,m^{-2}yr^{-1}}$ but without a homogeneous distribution of values over this range. Only two stations have observational values larger than $100 \, \mathrm{g\,m^{-2}yr^{-1}}$ and the bulk set of stations comprised values of between 0.1 and 75 $\mathrm{g\,m^{-2}yr^{-1}}$.

The dataset *Network-SET-M* includes field measurements for 20 additional stations located in the Mediterranean and Sahel regions to represent both deposition near to dust sources (O'Hara et al., 2006), as well as at intermediate distances from them (Vincent et al., 2016). The values in this dataset ranges from 4.2 to 270 $\mathrm{g\,m^{-2}yr^{-1}}$ and allow us to visualise regional differences in the dust deposition flux. The INDAAF (International Network to study Deposition and Atmospheric composition in Africa) stations (Marticorena et al., 2017) provide us with an estimation of the inter-annual variability which is large in the Sahel region (see the Table S.MD.7)

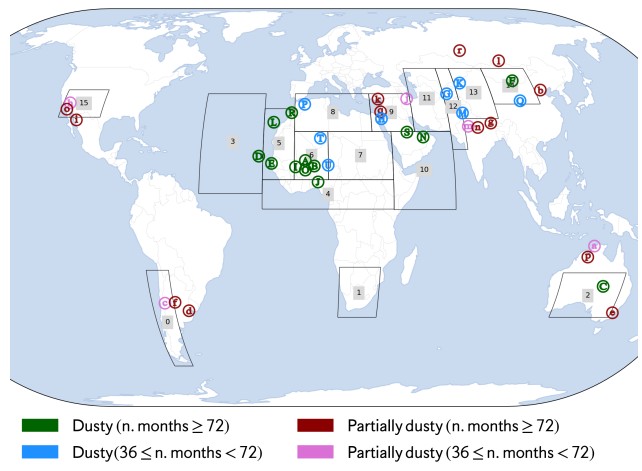

**Figure 3.** Map showing the 39 dusty stations from AERONET, classified in two groups: 21 dust-dominated stations (uppercase letters), and 18 stations where dust is important but not necessarily dominant (lower-case letters). The colour allows differentiating also the number of months in the observed time-series. The regions for the preferential dust emission sources (plus Mid-Atlantic region) are indicated with numbered boxes. The region number correspond to the names of the regions to which they belong: South-America (0), South-Africa (1), Australia (2), Mid-Atlantic (3), Gulf of Guinea (4), Western Sahara (5), Mali/Niger (6), Bodélé/Sudan (7), North Sahara (8), North Middle-East (9), South Middle=East (10), Kyzyl Kum (11), Thar (12), Taklamakan (13), Gobi Desert (14), North-America (15).

## 3.2 Surface concentrations

The first part of the climatological dataset for dust concentrations (see Table S.MD.4) at the surface has been adopted from estimations done by the University of Miami Oceanic Aerosols Network (UMOAN) whose instruments are filter collectors deployed in the North Atlantic and Pacific Oceans (Prospero and Nees, 1986; Prospero and Savoie, 1989). This dataset provides climatological monthly averages with a standard deviation that represents inter-annual variability. The second part of the climatological dataset is based on yearly values from the station data shown in Mahowald et al. (2009). The dataset comprises of 36 stations with values from $5 \cdot 10^{-2}$ to $100\ \mu\mathrm{g\,m^{-3}}$ distributed within the full range of values but grouped in clusters correlated with the geographical regions they belong to.

## 3.3 INDAAF stations of data

The multi-instrument network was deployed in the framework of the African Monsoon Multidisciplinary Analysis, and belongs to the INDAAF set of data-stations. Marticorena et al. (2010) described the collocated measurements of wet and dry deposition, as well as, surface concentrations (of particulate matter smaller than 10 μm) at three stations in the Sahel region, see Tables S.MD.6 and S.MD.7 and Figure 1 panel (b). The stations also measured precipitation, wind velocity and surface temperature. Additionally, in the same locations there are AERONET sun-photometers to measure aerosol optical depths.

**Table 4.** Given the mass mixing rations $X_s$, airmass $a_{mass}$, optical depths $\tau_s$ per species $s$ and air density $\rho_{air}$. We indicate here the method used to estimate other diagnostics. (i,j) are the coordinates/index of each cell grid, l represents the level/layer. $A(i,j)$ is the area of (i,j) grid cell, $l_0$ represents the surface layer. The units refer to those of original CRESCENDO diagnostics.

| Diagnostic | Symbol | Equation | Units |
|---|---|---|---|
| Grid cell area | $A(i,j)$ | Diagnostic provided by models | $\mathrm{m}^2$ |
| Mass mixing ratio | $X_s(i,j,l)$ | Diagnostic provided by models | $\mathrm{kg\,kg}^{-1}$ |
| Airmass | $a_{mass}(i,j,l)$ | Diagnostic provided by models | kg |
| Optical depth at 550nm | $\tau_s(i,j)$ | Diagnostic provided by models | - |
| Grid cell loadings | $L_s(i,j)$ | $\sum_l \left[ X_s(i,j,l) \cdot a_{mass}(i,j,l) A(i,j)^{-1} \right]$ | $\mathrm{kg\,m}^{-2}$ |
| Total column load | $TL_s$ | $\sum_{i,j} L_s(i,j)A(i,j) = \sum_{i,j,l} X_s(i,j,l) \cdot a_{mass}(i,j,l)$ | kg |
| Surface concentrations | $\widetilde{x_s}(i,j)$ | $X_s(i,j,l_0) \cdot \rho_{air}(i,j,l_0)$ | $\mathrm{kg\,m}^{-3}$ |
| MEE at 550nm (†) | $m_s^{ee}(i,j)$ | $\tau_s(i,j)L_s(i,j)^{-1}$ | $\mathrm{kg}^{-1}\,\mathrm{m}^2$ (‡) |

† MEE: Mass Extinction Efficiency. ‡ The MEE shown in the analysis has units $\mathrm{g}^{-1}\,\mathrm{m}^2 = 10^{-3}\mathrm{kg}^{-1}\,\mathrm{m}^2$.

## 3.4 AERONET optical properties

The AERONET (Aerosol Robotic Network) database implemented in our comparisons relies on the Version 3 (Level 2.0) algorithm. Based on this new algorithm the entire database of observations was reprocessed in 2018 (Giles et al., 2019). The database comprises aerosol optical depths and Ångström exponents, as well as, fine and coarse optical properties obtained with a new cloud-screening quality control scheme. The actual division threshold between fine and coarse particles is ascertained by the inversion algorithm that aims to differentiate aerosol particles from ice crystals and it lies between 0.44 and 0.99 µm.

The network database provides daily data, allowing for events analysis, and there is also a monthly time resolution dataset, used here to examine decadal, yearly and seasonal properties. We processed data from 300 stations of the full network to explore general properties. For the dust analysis we selected those stations where all the models together considered dust to be an important contributor to the aerosol composition (at the geographical location of the AERONET station). This subset is named here dusty set of stations, which are shown in Figure 3. It comprises 39 stations divided into two subsets: those stations where the dust has a *dominant* role in terms of the optical depth ($\tau_{440}^{dust} > 0.5\tau_{440}^{all-aer}$ for all models and all the months of the year, where $\tau_{440}^{all-aer}$ refers to optical depth at 440 nm of all aerosols and $\tau_{440}^{dust}$ is the optical depth of mineral dust aerosols at 440 nm), and those where the dust is *important* although not necessarily dominant for all the models (even if the dust optical depth from a single model contributes more than 50% of the total aerosol optical depth). The first subset comprises of 21 stations, and it is denoted with upper-case letters in Figure 3. The second comprises 19 stations, and it is denoted with lower-case letters. The dusty stations set over Africa is consistent with the stations analysed by Huneeus et al. (2011) based on Bellouin et al. (2005) criteria, but it has been extended with stations in Australia, South-America, North-America and Asia consistent with Klingmüller et al. (2018). Figures with the seasonal cycle of aerosol optical depth of the dusty dominant

**Table 5.** Statistics used to inter-compare models and observations and perform model inter-comparisons. $N$ indicates the number of observations or sample size. When the analysis refers to a global performance of the model over a set of instruments, N represents the number of stations. When the statistical analysis is done over a time series of values, N represents the number of time samples usually corresponding to a specific location. Pearson Correlation Coefficient ($\rho$), bias ($\delta$), normalised bias ($\delta_N$), Ratio of standard deviations ($\Sigma$), Normalised mean absolute error ($\theta_N$) and Root mean square error (RMSE=$\eta$).

| Statistic Estimator |
| --- |
| $\rho = Cov(log_{10}X, log_{10}Y)/(\sigma(log_{10}X)\sigma(log_{10}X))$ |
| $\delta = N^{-1}\sum_{i=1}^{N}(x_i^{(mod)} - x_i^{(obs)})$ |
| $\delta_N = \sum_{i=1}^{N}(x_i^{(mod)} - x_i^{(obs)})/(\sum_{i=1}^{N} x_i^{(obs)})$ |
| $\Sigma = \sigma_{mod}/\sigma_{obs}$ |
| $\theta_N = \sum_{i=1}^{N}\left| x_i^{(mod)} - x_i^{(obs)}\right|/(\sum_{i=1}^{N} x_i^{(obs)})$ |
| $\eta = N^{-1}\sqrt{\sum_{i=1}^{N}(x_i^{(mod)} - x_i^{(obs)})^2} = RMSE$ |

and important stations that highlight the classification criteria are shown in the supplement material (Figures S.DOD.10 and S.DOD.11).

## 3.5 MODIS dust related products

Interactions between dust and radiation are defined through three optical properties: dust optical depth (DOD), single scattering albedo ($\omega$) and the asymmetry parameter which defines the ratio of the radiation scattered forward over the radiation scattered backward. For the dust coarse mode, the dust optical depth can be estimated using the Moderate Resolution Imaging Spectroradiometer (MODIS) enhanced deep-blue (DB) aerosol optical depth (Sayer et al., 2014) as done by Pu and Ginoux (2018b) with the additional support of the MODIS product of single-scattering albedo ($\omega$) and Ångström exponent ($\alpha$). The rationale of the method relies on the properties of these three optical parameters applied to aerosol particles. First, $\alpha$ is very sensitive to particle size, so there are parametrisations of aerosol optical depth that use it to separate each mode contribution. Second, aerosols with low absorption and large scattering like sea-salt have $\omega \simeq 1$, whereas mineral dust is considered an absorbing aerosol. Third, the dependency of $\alpha(\lambda)$ on wavelength contains a signature of the aerosol composition. Given this information, we have considered 2 different MODIS dust optical depth related datasets. One of them is a pure filter of aerosol optical depth to differentiate those pixels where dust is expected to be the dominant contribution to aerosol optical depth, but without an attempt to estimate the actual fraction of mineral dust, so it is considered here as an upper threshold of the actual DOD of the coarse mode (because particles of dust with diameters below 1 μm are thought to contribute less than 10% to the total dust optical depth). The other method aims to explicitly separate sea-salt, and proceeds to rescale the aerosol optical depth to ascertain an actual value of DOD, and according to Pu and Ginoux (2018b) it may be considered a lower-bound of the DOD. Additional information and a comparison of these created products are given in the supplementary information, see Figures S.MD.2 and S.MD.3.

**Table 6.** Global dust mass balance, dust loading, dust optical depth (DOD), mass extinction efficiency (MEE) and lifetime for each model and each experiment available. CNRM has two configurations one specific for CRESCENDO referred as CNRM-6DU and another for CMIP6 denoted as CNRM-3DU. The UKESM is not diagnosing the dust sedimentation separately and dry deposition flux diagnostics accounts for all removal of dust except for wet deposition. The units are $Tg yr^{-1}$ for emissions and depositions tendencies, $Tg$ for Load, $m^2 g^{-1}$ for MEE and days for lifetime. MEE is calculated as the mean of the MEE(x,y) field, while $\widehat{MEE}$ is the ratio of DOD and Load mean fields. $\Delta$ represents the ratio of the Net (Emission-Total Deposition) relative to emission in %. $\mathcal{R}_{dep}$ is the ratio of total dry (including gravitational settling) over total wet deposition. MM-mean shows the multi-model mean for each experiment (and each variable) and MM-$\sigma$ the estimated multi-model standard deviation. Note that some statistical estimations (indicated with ‡) related to the deposition are not including the UKESM model as we cannot separate gravitational settling from other dry deposition processes. Due to the larger values of the $\Delta$ parameter, CNRM-6DU is not included in the statistics noted with ‖ and ‡.

| Model | Exp. | Emi. [Tg yr⁻¹] | Dep. [Tg yr⁻¹] | Net [Tg yr⁻¹] | Δ % | Dry Dep. [Tg yr⁻¹] | Wet Dep. [Tg yr⁻¹] | Sedim. [Tg yr⁻¹] | $\mathcal{R}_{dep}$ - | DOD - | Load [Tg] | MEE [m²g⁻¹] | $\widehat{MEE}$ [m²g⁻¹] | Lifetime [day] |
|---|---|---|---|---|---|---|---|---|---|---|---|---|---|---|
| CNRM-3DU | PD | 2605.2 | 2679.6 | -74.5 | -2.86 | 1708.1 | 753.8 | 217.8 | 2.55 | 0.011 | 13.3 | 0.63 | 0.44 | 1.9 |
| EC-Earth | PD | 1126.6 | 1126.7 | -0.12 | -0.01 | 367.8 | 493.2 | 265.7 | 1.28 | 0.029 | 11.7 | 1.86 | 1.27 | 3.8 |
| IPSL | PD | 1557.5 | 1558.9 | -1.44 | -0.1 | 329.3 | 968.3 | 261.3 | 0.61 | 0.026 | 16.4 | 0.82 | 0.82 | 3.8 |
| NorESM | PD | 1368.2 | 1368.3 | -0.09 | -0.01 | 84.0 | 275.7 | 1008.6 | 3.96 | 0.023 | 7.2 | 2.86 | 1.63 | 1.9 |
| UKESM | PD | 7524.4 | 7527.6 | -3.21 | -0.04 | 6566.3† | 949.8 | - | 6.91 | 0.011 | 18.1 | 0.5 | 0.31 | 0.9 |
| MM-mean | PD | 2836.4‖ | 2852.2‖ | - | - | 622.3‡ | 622.8‡ | 438.5‡ | - | 0.02‖ | 13.32‖ | 1.33‖ | 0.89‖ | 2.5‖ |
| MM-σ | PD | 2680.8‖ | 2680.5‖ | - | - | 734.7‡ | 302.1‡ | 380.9‡ | - | 0.008‖ | 4.25‖ | 1.01‖ | 0.556‖ | 1.3‖ |
| CNRM-3DU | PI | 2651.5 | 2730.2 | -78.7 | -2.97 | 1728.7 | 781.0 | 220.4 | 2.49 | 0.012 | 13.4 | 0.63 | 0.44 | 1.8 |
| EC-Earth | PI | 1145.8 | 1145.4 | 0.44 | 0.04 | 374.4 | 511.6 | 259.4 | 1.24 | 0.027 | 11.6 | 1.7 | 1.17 | 3.7 |
| IPSL | PI | 1551.7 | 1553.2 | -1.49 | -0.1 | 330.6 | 961.0 | 261.5 | 0.62 | 0.027 | 16.7 | 0.82 | 0.82 | 3.9 |
| NorESM | PI | 1407.3 | 1407.5 | -0.21 | -0.01 | 86.8 | 287.4 | 1033.2 | 3.90 | 0.023 | 7.4 | 2.75 | 1.56 | 1.9 |
| UKESM | PI | 7421.9 | 7413.6 | 8.25 | 0.11 | 6475.6† | 938.0 | - | 6.90 | 0.01 | 17.4 | 0.49 | 0.29 | 0.9 |
| MM-mean | PI | 2835.6‖ | 2850.0‖ | - | - | 630.13‡ | 635.3‡ | 443.6‡ | - | 0.02‖ | 13.3‖ | 1.28‖ | 0.87‖ | 2.4‖ |
| MM-σ | PI | 2627.4‖ | 2622.4‖ | - | - | 743.23‡ | 296.5‡ | 393.5‡ | - | 0.008‖ | 4.06‖ | 0.95‖ | 0.52‖ | 1.3‖ |
| CNRM-3DU | PDN | 1812.1 | 1888.7 | -77.62 | -4.28 | 1290.6 | 435.1 | 164.0 | 3.34 | 0.011 | 11.6 | 0.63 | 0.46 | 2.3 |
| IPSL | PDN | 1295.3 | 1297.1 | -1.77 | -0.13 | 268.8 | 813.1 | 215.2 | 0.60 | 0.024 | 14.8 | 0.82 | 0.82 | 4.2 |
| NorESM | PDN | 1733.6 | 1733.4 | 0.12 | 0.01 | 115.7 | 345.5 | 1272.2 | 4.02 | 0.029 | 9.1 | 2.87 | 1.61 | 1.9 |
| MM-mean | PDN | 1613.7‖ | 1640.1‖ | - | - | 558.4‖ | 531.2‖ | 550.4‖ | - | 0.02‖ | 11.8‖ | 1.44‖ | 0.96‖ | 2.8‖ |
| MM-σ | PDN | 278.5‖ | 307.1‖ | - | - | 638.7‖ | 248.2‖ | 625.5‖ | - | 0.009‖ | 2.86‖ | 1.24‖ | 0.59‖ | 1.2‖ |
| CNRM-6DU | PD | 3542.2 | 4134.7 | -592.5 | -16.7 | 1283.9 | 2108.9 | 741.9 | 0.96 | 0.023 | 32.6 | 0.55 | 0.36 | 3.4 |
| CNRM-6DU | PI | 3887.3 | 4552.0 | -664.7 | -17.0 | 1415.2 | 2319.1 | 817.7 | 0.96 | 0.025 | 35.2 | 0.56 | 0.36 | 3.3 |
| CNRM-6DU | PDN | 1278.4 | 1507.3 | -228.8 | -17.9 | 499.5 | 716.8 | 290.9 | 1.10 | 0.011 | 15.2 | 0.56 | 0.38 | 4.3 |

† Values including the sedimentation. ‡ Statistic is not including the UKESM. ‖ Statistic is not including UKESM and CNRM-6DU.

## 3.6 MISR aerosol optical depth derived products

The Multi-angle Imaging Spectro-Radiometer (MISR) is a sensor on-board the Terra satellite which takes advantage of its multi-angle measurement capabilities. It is able to ascertain the presence of non-spherical particles on the aerosol products at four different wavelengths. The optical depth at several wavelengths has been used to compute the Ångström exponent between Mar-2000 and Dec-2014 of MISR, and compare with the models' Ångström exponent based on the same information. This product gives us information on how the models represent the spectral dependence of optical depth. Our computation using the 446 nm and the 672 nm wavelengths, has been compared with the MISR Ångström exponent product to validate our computations, see Figure S.GL.8.

## 4 Methods

As part of this study we calculated several diagnostics not directly provided by the different models. Table 4 shows how they have been estimated together with their units. Regarding the statistical methods, Table 5 shows the metrics used for the comparison of the CRESCENDO models with the comprehensive suite of observations. The skill of the models to represent the dust optical depth over dust source regions has been calculated based on the Pearson correlation. Given that this statistic is not robust, because its instability in the presence of outliers (Li et al., 2006), and only representative of linear relationships, the skill is also estimated based on the Spearman rank correlation to ensure the robustness of the results. For the other comparisons beyond the skill, the scatter-plots are informative of the quality of the Pearson correlation estimator.

For the comparison against the networks of instrument used: one monitoring surface concentrations, two for total deposition and one that retrieves dust optical depth, we proceed with the same methodology. For each observation, we chose the model value of the corresponding variable in the grid pixel to which this measuring station belongs. Given the different area covered by the grid cell and the grid point location of the in-situ measurements, there is an underlying representation error. However, the observational datasets of total deposition and surface concentrations at point based sites are climatological estimations which can be representative of larger areas. The values for the parameters discussed here are time averaged over the 15-year simulations and hence the produced fields are smooth over sub-grid scales.

The surface concentration and total deposition comparisons are presented as scatter-plots together with three associated statistics: the Pearson correlation (evaluated in log-scale), the bias and the RMSE (root mean square error). These last two metrics can be used to characterise quantitative differences between each model and the observations. Additional statistics are summarised in Tables 11, 12 and 13 including the normalised bias and the normalised mean absolute error which help us understand how the models differ when scaled to the observation values.

## 5 Results

The results are divided into six different subsections. First a comparison at the global scale summarises the main properties of the global dust cycle in the models analysed, which is complemented with an overview of the spatial pattern of the temporal

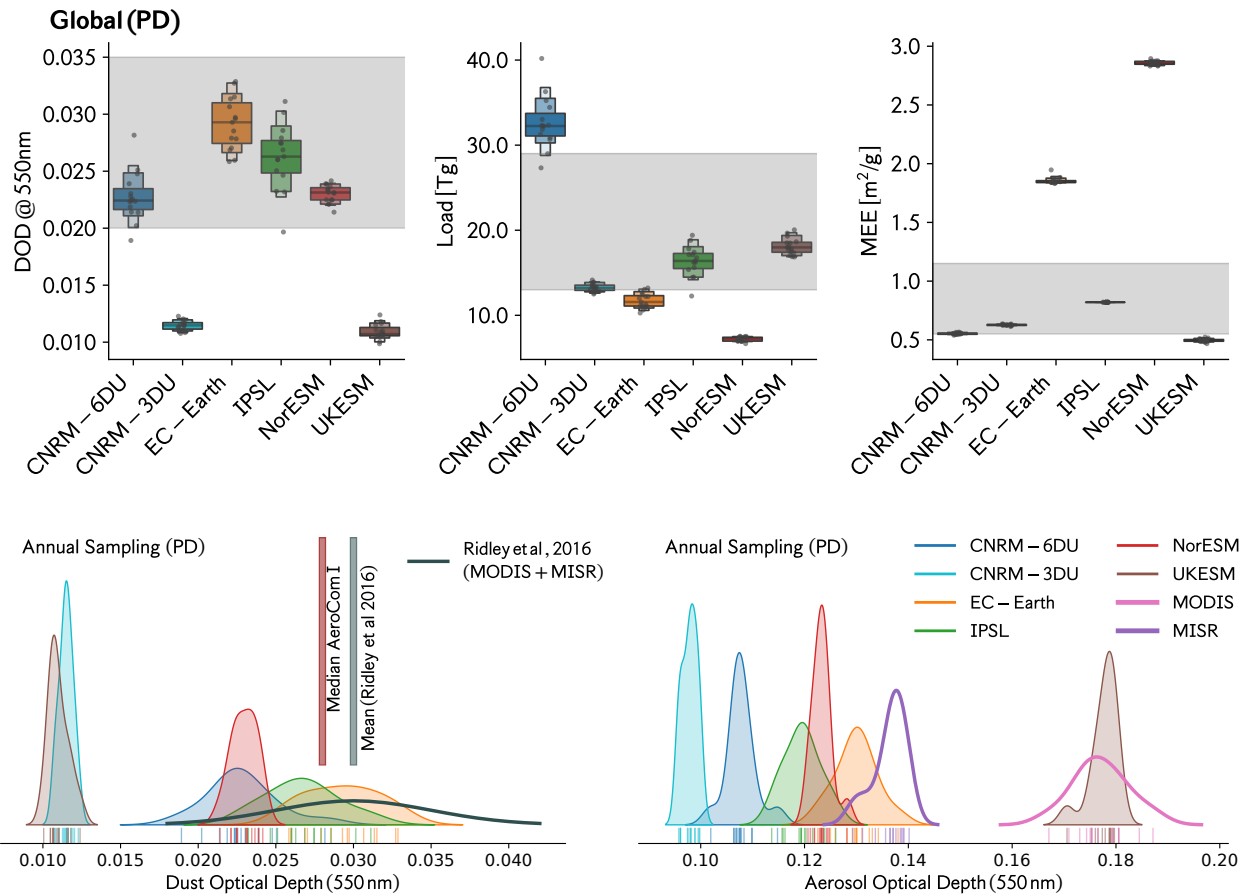

**Figure 4.** *Top panel*: Global dust cycle values for PD experiment. The gray shaded region represents the expected interval range based on Kok et al. (2017) for dust particles with diameter up to 20 μm for Dust Optical Depth (DOD), Load and mass extinction efficiency (MEE). The grey dots over the box-plot represent each of the annual values. *Bottom left panel* represents the estimated distribution of global dust optical depth annual values (our sample values per model are represented by the coloured vertical marks just above the x-axis). The *bottom right panel* is the analogous for all aerosols optical depth. Both distributions are normalized and vertical axis represents a probability. For both the models and the observations (MISR and MODIS) the estimates are for time-period 2000-2014. Additional analysis analogous to top panel but constrained over different regions are in Supplementary material (Figures S.GL.1 and S.GL.2).

mean of the 15 years of simulation (based on monthly values) for each of the climate models of the study. The next four sections are detailed analysis of the dust properties: emission, deposition, optical depth and surface concentrations. Each one is described at the regional scale and compared against a network of instruments and/or satellite retrievals when available. In all the cases, the PD experiment simulations have been taken as the baseline of the inter-comparison and shown in the main paper. The results for the other experiments (PDN and PI), if not present in the main paper, are shown in the supplementary material. The case of nudged wind simulations (PDN) is used to ascertain the role of modelled surface winds on inter-model differences, whereas the simulation with PI emissions help us to evaluate the possible role of prescribed emissions.

## 5.1 Global dust properties

The global dust cycle has been analysed in terms of global climatological values and complemented by a study of the role of the particle size distribution on the direct radiative effects (based on the IPSL model with 4 dust modes).

The dust particle size distribution is physically constrained by emission, transport and deposition (wet and dry), whereas, other aerosol processes like aerosol nucleation, condensation and coagulation have a minor role on the evolution of the size distribution (Mahowald et al., 2014). Therefore, the first step to describe the global atmospheric dust cycle in climate models consists of a characterisation of the emission and deposition fluxes at the surface. This analysis is complemented by the analysis of two size-integrated properties: the dust optical depths and loadings. Other phenomena present in the Earth System dust cycle more relevant for paleoclimate studies, like those derived from the stabilisation of dust deposition on the surface on long time-scales, are not considered in this work.

The global dust budget is analysed for the whole time period of the simulations over the three different simulations considered: PD, PDN and PI. Table 6 presents the mean global values of each model. It describes the dust mass balance in terms of emission, dry and wet deposition. A parameter $\mathcal{R}_{dep}$ is defined to represent the ratio of total dry to total wet deposition. In addition, $\Delta$ represents the fraction (%) of the emissions not deposited relative to the total emission. This last parameter is used to ascertain if the dust cycle from emission to deposition is consistent in terms of global mass conservation, or, to the contrary, whether the model transport introduces any inconsistency in the modelled dust cycle. In particular, the parameter $\Delta$ is used to decide those models and experiments that will be included in the multi-model ensemble mean to ensure internal consistency in the ensemble.

In this regard, the mass budget of the CNRM-3DU model is closed to within $\Delta \simeq 3\%$ as its dynamical core is based on a semi-Lagrangian method (Voldoire et al., 2012, 2019) which is not fully mass conservative in terms of its tracers. In the case of the PDN experiment there is an increase to $\Delta \simeq 4.3\%$, because the excess of mass in the deposition with respect to the emissions is similar for all the experiments, but the emissions of CNRM-3DU decrease with nudged winds by 30%. The deposition value therefore is biased by an approximately constant amount of $75\ \mathrm{Tg\,yr^{-1}}$ independently of the wind field. Given that in any case the value of $\Delta < 5\%$, then we have included the CNRM-3DU model in the ensemble means. In the case of the CNRM-6DU model the consequences of its dynamical core properties are the same, hence there is also a bias. However, it is close to $600\ \mathrm{Tg\,yr^{-1}}$ in total deposition, producing a value of $\Delta$ larger than $15\%$. Therefore, this model is not included in the ensemble means. In both cases, the CNRM-3DU and the CNRM-6DU models the bias in total deposition implies an excess

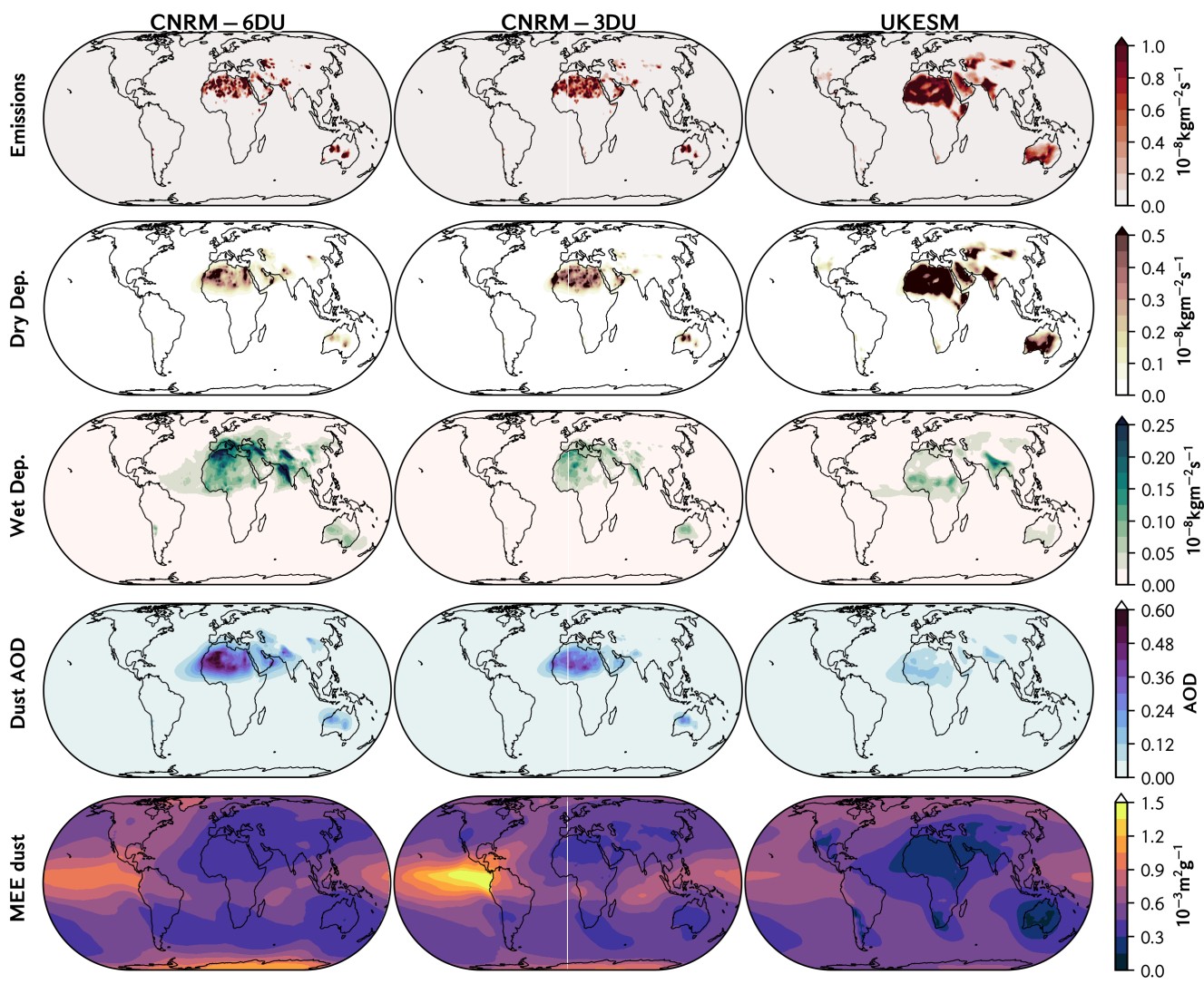

**Figure 5.** CRESCENDO-ESMs global maps describing dust properties (averaged over the 15 years): emission tendency, deposition tendencies, dust optical depth and mass extinction efficiency. The models included have a bin-based dust parametrisation, these models are: CNRM-6DU, CNRM-3DU and UKESM models. The equivalent figures for PI and PDN experiments are shown in Supplementary Material: Figure S.GL.3 and Figure S.GL.4 respectively.

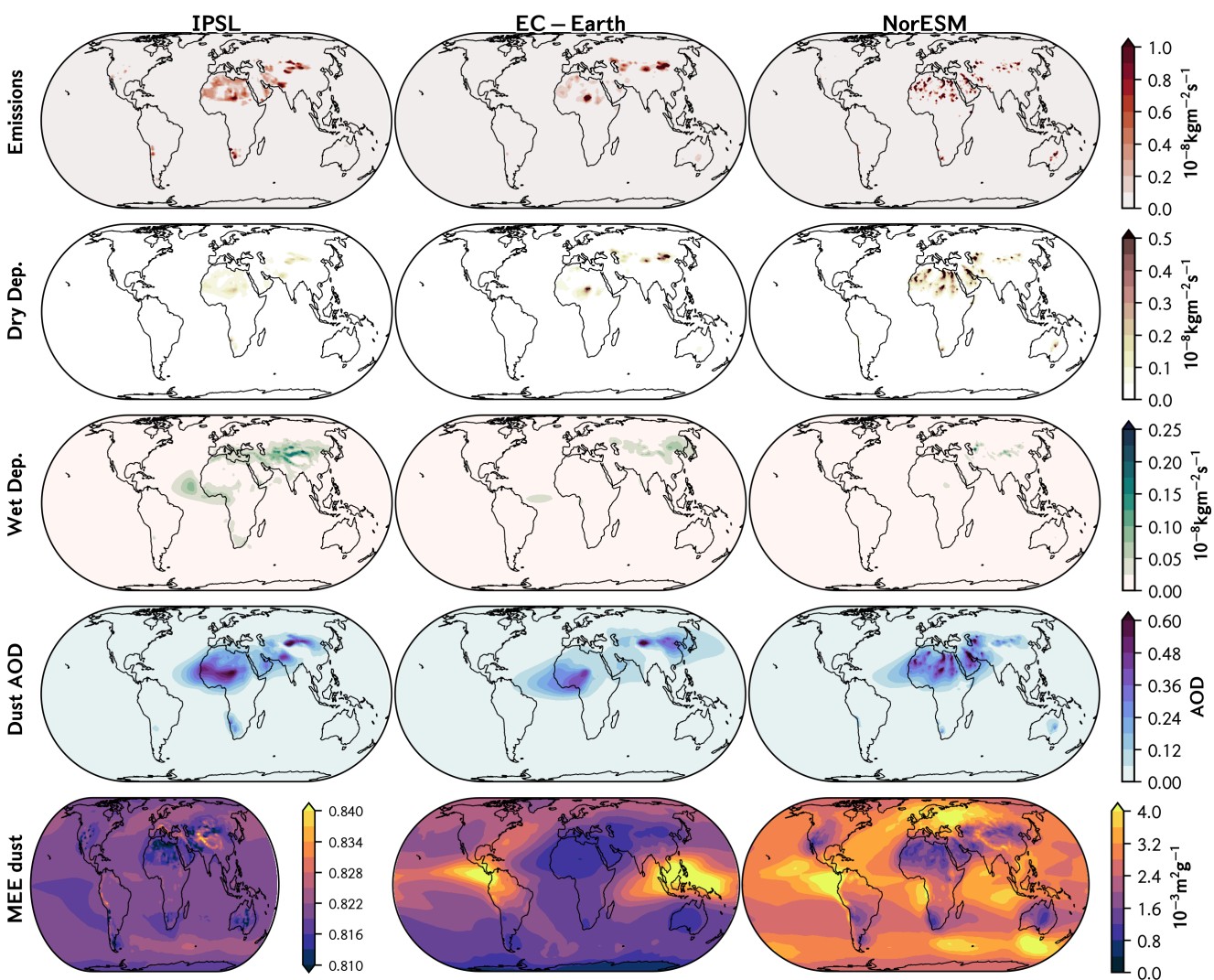

**Figure 6.** CRESCENDO-ESMs global maps of dust properties (averaged over the 15 years): emission tendency, deposition tendency, dust optical depth and mass extinction efficiency. The models included have a modal based dust parametrisation, these models are: IPSL-INCA, NorESM and EC-Earth. The equivalent figures for PI and PDN experiments are shown in Supplement Material: Figure S.GL.5 and Figure S.GL.6 respectively.

**Table 7.** Direct Radiative Effects (DRE) at the top of the atmosphere (TOA) and surface (SRF) without clouds in long-wave (LW) and short-wave (SW) for the IPSL model with 4 dust modes as described by Albani and et al (2021). For each mode the value from each method *in* and *out* and their mean value (of both methods) is indicated (the mean value in italics). Both methods are described in the Appendix A, the *method in* adds each specific mode to a case without any mode of dust, the *method out* removes that specific mode to a case with all the modes of dust. Values in italics represent those derived from other values of the table. The value of the sum of the 4 modes is not equal to the value of the multi-modal DRE of dust for each method in/out individually. But the mean of both methods in and out is consistent with the multi-modal DRE.

| Dust DRE | TOA LW [W m$^{-2}$] | | | TOA SW [W m$^{-2}$] | | |
|---|---|---|---|---|---|---|
| | *in* | *out* | *Mean* | *in* | *out* | *Mean* |
| Mode $m_1$ | 0.0074 | 0.0063 | *0.0069* | -0.1360 | -0.0932 | *-0.1146* |
| Mode $m_{2.5}$ | 0.0399 | 0.0349 | *0.0375* | -0.2737 | -0.2300 | *-0.2518* |
| Mode $m_7$ | 0.0913 | 0.0848 | *0.0881* | -0.0779 | -0.0440 | *-0.0609* |
| Mode $m_{22}$ | 0.0110 | 0.0087 | *0.0099* | 0.0188 | 0.0139 | *0.0163* |
| $\sum$ modes | *0.1497* | *0.1348* | *0.1422* | *-0.4689* | *-0.3533* | *-0.41* |
| Multimodal | | | 0.142 | | | -0.41 |

| Dust DRE | SRF LW [W m$^{-2}$] | | | SRF SW [W m$^{-2}$] | | |
|---|---|---|---|---|---|---|
| | *in* | *out* | *Mean* | *in* | *out* | *Mean* |
| Mode $m_1$ | 0.0194 | 0.0142 | *0.0168* | -0.2367 | -0.1854 | *-0.2110* |
| Mode $m_{2.5}$ | 0.1180 | 0.0910 | *0.1045* | -0.6413 | -0.5378 | *-0.5895* |
| Mode $m_7$ | 0.3217 | 0.2831 | *0.3024* | -0.6615 | -0.5548 | *-0.6082* |
| Mode $m_{22}$ | 0.0540 | 0.0371 | *0.0455* | -0.0653 | -0.0442 | *-0.0547* |
| $\sum$ modes | *0.5131* | *0.4253* | *0.4692* | *-1.6047* | *-1.3223* | *-1.4635* |
| Multimodal | | | 0.467 | | | -1.45 |

of mineral dust in the atmosphere not consistent with the actual modelled emissions. A further complication is that the bias leads to other biases in variables like concentrations, load and optical depths. For this reason the CNRM-6DU model is not used in our analysis to draw conclusions about the dust cycle. But it is kept in the other analyses to be compared with future developments of the model that improve/fix the mass conservation, and subsequently highlight better the implications of these kinds of numerical instabilities in dust modelling. For the other models $\Delta < 0.1\%$, with NorESM and EC-Earth presenting values closest to zero.

The multi-model mean global emissions for the PD and PI simulation experiments are 2836 Tg yr$^{-1}$ and 2835 Tg yr$^{-1}$ respectively, with standard deviations of 2680 and 2627 Tg yr$^{-1}$. The PDN experiment shows an ensemble mean value of 1614 Tg yr$^{-1}$ which is significantly smaller because of the models included (see Table 1), but also because of an important decrease in the CNRM-3DU total emissions. Indeed, the decrease in emissions with nudged winds is even higher in the CNRM-6DU. As a consequence, our ensemble mean value for the PDN experiments agrees well with recent estimations (Kok et al., 2017)

when large particles (diameter $\leq$ 20 μm) are not included. But it also agrees well with previous estimations of 1500 $\mathrm{Tg\,yr^{-1}}$ based on the DEAD model (Zender et al., 2003) for particles with diameters smaller than 10μm. At the same time, when nudged winds are used (PDN ensemble), the standard deviation of total emissions (278 $\mathrm{Tg\,yr^{-1}}$) is significantly smaller than for the PD or PI cases. For the PD experiment, the multi-model ensemble mean total emission, for the same models as available for PDN, has a mean value of 1843 $\mathrm{Tg\,yr^{-1}}$ with a standard deviation of 544 $\mathrm{Tg\,yr^{-1}}$ significantly larger than the standard deviation of the PDN experiment. Therefore, nudged winds decrease model diversity in terms of global emissions. Indeed, the CNRM-6DU and CNRM-3DU models have total emissions with nudged winds similar to the CRESCENDO-ESMs ensemble mean, but they produce higher emissions without nudged winds-field, i.e. 2600 $\mathrm{Tg\,yr^{-1}}$ in CNRM-3DU model (diameters up to 10 μm), and 3500 $\mathrm{Tg\,yr^{-1}}$ for CNRM-6DU (diameters up to 100 μm, see Table 1). These values are similar to the 3000 $\mathrm{Tg\,yr^{-1}}$ reported by Tegen and Fung (1994) for particle sizes between 0.1 and 50 μm. Finally, due to the presence of particles with diameters up to 62 μm, the UKESM model has notably higher emissions (although in this case we can't assess the role of surface winds).

This higher value of total emissions due to large particles is not directly correlated with the modelled dust load in the atmosphere. The reason is that the lifetime of dust particles in the atmosphere depends on the size and these large particles sediment faster. For instance, the UKESM model has monthly mean global loading values close to the other models, and the smaller lifetime of dust in the atmosphere (less than 12 hours, a characteristic value of largest particles). In fact, the dry deposition of larger particles for UKESM (which for this model includes sedimentation) is truly dominant, resulting in a wet deposition close to other models, like IPSL, without the largest particles modelled. On the contrary the CNRM-6DU wet deposition is two times larger than that of the UKESM or IPSL models in the PD simulation (being CNRM-6DU the only model for which wet deposition exceeds total dry deposition) but close to IPSL with nudged winds. Because larger particles are deposited faster by gravitational settling, it is expected that $\mathcal{R}_{dep}$ would be larger for those models including the largest particles, but it is only obvious for the UKESM model. For the CNRM-6DU model that is not the case. EC-Earth has double the value of $\mathcal{R}_{dep}$ of IPSL, and NorESM is 6 times larger. Previously, Shao et al. (2011) reported values for $\mathcal{R}_{dep}$ of between 1.03 and 8.1 also uncorrelated with the size range of the dust particles modelled. The multi-model ensemble mean for total dry deposition, without gravitational settling is 622 $\mathrm{Tg\,yr^{-1}}$ for the PD experiment and 558 $\mathrm{Tg\,yr^{-1}}$ for PDN. In the case of wet deposition, we estimated 623 and 531 $\mathrm{Tg\,yr^{-1}}$ for the multi-model mean for the PD and PDN experiments respectively. Despite the similar values of our ensemble mean, the standard deviation of dry deposition is more than two times that from wet deposition. To summarise, each of the processes: sedimentation, wet deposition and dry deposition (without sedimentation) has a similar contribution in the ensemble mean for all the experiments, but this is masking strong differences in these three properties from each of the models.

As explained above, the impact of the largest particles on the global behaviour of loading and dust optical depth is considered less important than coarse particles (up to 10 μm), so this hypothesis allows us to compare all models with observational constraints that rely on optical depth measurements. Figure 4 (top panel) compares the PD experiment with the Kok et al. (2017) proposed values of dust optical depth and total load, where we also derive the mass extinction efficiency (MEE) field as the ratio of dust optical depth to loading field, see Table 4.

**Table 8.** First part of the table: Emissions [Tg yr$^{-1}$] for Present Day (PD) and their contribution fraction to the total global emissions. Globally, over Land and over coastline pixels. (*) Denotes those models with modelled bin diameters larger than 20$\mu m$. Sahara desert emissions and its percentage over total emissions is obtained from the sum of the regions: Western Sahara, Mali, Bodele and North Sahara, so it is not including Sahel. Second part of the table: Emissions [Tg yr$^{-1}$] for Present Day (PD) simulations. Over 16 different regions, see Figure 3. In brackets the order of the 10 regions with the largest emissions. The multi-model ensemble mean (MM-mean) includes the mean values ± the standard deviation for all the models, and (‡MM-mean) for all the models without UKESM. On the Supplementary Information (Section E), Tables E1 to E4 have the analogous information for the PI and PDN experiments. Ensemble mean of emissions are including CNRM-6DU.

| | CNRM-6DU (PD) | | CNRM-3DU (PD) | | EC-Earth (PD) | | IPSL (PD) | | NorESM (PD) | | UKESM (PD) | | MM-mean (PD) | | ‡MM-mean (PD) | |
|---|---|---|---|---|---|---|---|---|---|---|---|---|---|---|---|---|
| Global Earth | 3542.2 | (*) | 2605.2 | | 1126.6 | | 1557.5 | | 1368.2 | | 7524.4 | (*) | 2954 | (± 2415) | 2040 | (± 1012) |
| Land | 3377.4 | (*) | 2526.1 | | 1111.0 | | 1550.9 | | 1343.6 | | 7506.4 | (*) | 2903 | (± 2410) | 1982 | (± 948) |
| Ocean (Coast) | 164.8 | (*) | 79.1 | | 15.6 | | 6.6 | | 24.6 | | 18 | (*) | 52 | (± 61) | 58 | (± 66.) |
| Sahara Desert | 2071.5 | (58%) | 1734.2 | (66%) | 445.2 | (39%) | 715.4 | (46%) | 651.8 | (48%) | 4339.5 | (58%) | 1660 | (± 1466) | 1124 | (± 728) |
| North. Hemis. | 3135.3 | (88%) | 2292.7 | (88%) | 1072.9 | (95%) | 1377.6 | (88%) | 1256.1 | (92%) | 6614.9 | (89%) | 2625 | (± 2104) | 1827 | (± 870) |
| South. Hemis. | 406.9 | (12%) | 312.5 | (12%) | 53.7 | (5%) | 179.9 | (12%) | 112.1 | (8%) | 909.5 | (11%) | 329 | (± 313) | 213 | (± 145) |
| South America | 17.4 | | 13.3 | | 11.3 | | 36.9 | | 9.0 | | 18.2 | | 18 | (± 10) | 18 | (± 11) |
| South-Africa | 5.8 | | 17.0 | | 2.8 | | 113.8 | (5) | 31.8 | | 30.3 | | 34 | (± 41) | 34 | (± 46) |
| Australia | 343 | (4) | 235.4 | (5) | 35.9 | (9) | 10.7 | | 59.3 | (8) | 691.2 | (5) | 229 | (± 261) | 137 | (± 145) |
| Western Sahara | 242.5 | (6) | 296.1 | (4) | 52.1 | (6) | 87.4 | (7) | 95.8 | (4) | 788.8 | (4) | 260 | (± 276) | 155 | (± 108) |
| Mali/Niger | 382.4 | (3) | 323.2 | (3) | 49.5 | (7) | 83.4 | (9) | 69.5 | (7) | 841.2 | (3) | 292 | (± 304) | 182 | (± 158) |
| Bodele/Sudan | 540.4 | (2) | 569.4 | (1) | 259.6 | (2) | 305.8 | (1) | 190.6 | (3) | 1852.2 | (1) | 620 | (± 623) | 373 | (± 171) |
| North Sahara | 906.2 | (1) | 545.5 | (2) | 85.0 | (5) | 238.8 | (2) | 295.9 | (1) | 857.3 | (2) | 488 | (± 339) | 414 | (± 321) |
| North-MiddleEast | 253.7 | (5) | 112.8 | (9) | 17.0 | | 28.1 | | 146.1 | (4) | 303.7 | (8) | 144 | (± 117) | 112 | (± 97) |
| South-MiddleEast | 208.0 | (8) | 195.9 | (6) | 39.5 | (8) | 68.1 | | 83.7 | (5) | 441.1 | (6) | 173 | (± 149) | 119 | (± 77) |
| Kyzyl Kum | 230.3 | (7) | 118.7 | (7) | 118.8 | (3) | 142.4 | (4) | 198.7 | (2) | 377.4 | (7) | 198 | (± 99) | 162 | (± 50) |
| Thar | 136.3 | | 56.1 | (8) | 19.1 | | 86.2 | (8) | 13.9 | | 288.7 | (9) | 100 | (± 103) | 62 | (± 51) |
| Taklamakan | 15.2 | | 15.7 | | 104.6 | (4) | 153.0 | (3) | 35.5 | (9) | 75.0 | | 67 | (± 55) | 65 | (± 61) |
| Gobi | 140.2 | (9) | 36.2 | (9) | 269.8 | (1) | 113.1 | (6) | 80.6 | (6) | 230.3 | | 145 | (± 89) | 128 | (± 88) |
| North-America | 0 | | 1.1 | | 2.3 | | 28.4 | | 6.1 | | 57.1 | | 15.8 | (± 23) | 7.6 | (± 12) |
| Gulf-of-Guinea | 2.5 | | 1.4 | | 0.0 | | 0.0 | | 1.9 | | 69.3 | | 12.5 | (± 27.8) | 1.2 | (± 1.1) |

‡ These statistics exclude UKESM.

Figure 4 indicates that, aside from the CNRM-6DU model, all models have dust loadings smaller than 20 Tg with the loading of NorESM half that of the ensemble median value. As already noted above, the load of CNRM-6DU model is subject of a bias due to the artificial mass introduced during the transport. Therefore, the set of models included in our ensemble mean (Table 6) agrees with the AeroCom Phase I models where the fine dust dominates with a total load ensemble mean value of 15 Tg.

Based also on AeroCom Phase I, Huneeus et al. (2011) reported a MEE multi-model median of 0.72 $\mathrm{m^2g^{-1}}$, similar to the global MEE value of 0.6 $\mathrm{m^2g^{-1}}$ used by Pu and Ginoux (2018b) to compare DOD and dust loadings of CMIP5 models. Recently, Adebiyi et al. (2020) estimated a mean over 13 observational stations giving a value slightly smaller than 0.6 $\mathrm{m^2g^{-1}}$. Our estimation of MEE shows that EC-Earth and NorESM depart from that value, whereas the other models remain reasonably close to the Pu and Ginoux (2018b) hypothesis and the AeroCom Phase I median value. The larger MEE values of the EC-Earth

and the NorESM models can be due a combination of factors: they have the lowest dust loadings, and both are not modelling particles larger than about 8 μm, also in the case of NorESM the imaginary part of the refractive index is also the largest of all the models analysed. Our results highlight that the MEE depends on the modelled dust particle size distribution (in particular the presence of large particles) but with a significantly smaller inter-annual variability than dust optical depth and loading. This fact explains its use for ad-hoc relationships between dust optical depths and loadings with a constant factor (Pu and Ginoux,

2018b).

We note that the global mean values for the models, as shown in Figure 4 (top panel) are partially influenced by ocean or land regions with low dust loadings. To complement this analysis, we present two additional comparisons in the supplementary material. The first is shown in Figure S.GL.1, for the case when only values over land are considered. The second is shown in Figure S.GL.2 for the case when the annual values are estimated over the dust belt that covers most of the Sahara and the

Middle-East. Both Figures still indicate important differences between models.

To further understand the properties of dust optical depth, we calculated the distribution of values for each model with a kernel density estimation based on the histogram of the annual global values of dust optical depth. The results shown in Figure 4 (bottom left panel) indicate the presence of two main groups for our model ensemble: the first one centred around a value close to 0.01, and the second one around 0.025, a value closer to the proposed constraint. The solid black line shows the

distribution of dust optical depth at 550 nm proposed by Ridley et al. (2016), and the vertical lines indicate the mean of that distribution and the AeroCom Phase I median value. The EC-Earth model agrees actually in both central value and typical inter-annual variability (as represented by the width of the distributions). These results should be also interpreted in the context of the total aerosol optical depth (AOD), Figure 4 (bottom right panel). We observe that the UKESM has the lowest values of dust optical depth but actually the largest values of total aerosol optical depth, with similar global mean values to those

obtained by MODIS at 550 nm but with a narrower distribution. The EC-Earth model has AOD values slightly smaller than MISR estimates but with similar inter-annual variations. The bottom right panel of Figure 4 indicates model discrepancies in the magnitude of the inter-annual variability (as measured by the width of the distribution) and an overall underestimation of AOD at 550 nm with respect to these satellite platforms.

A specific PDN experiment with the IPSL model was run for 5 years (2010 to 2014) to analyse how the representation

of the dust size distribution influences the dust cycle. In this simulation, named IPSL-4DU, the dust scheme is based on 4

dust insoluble modes ($m_1, m_{2.5}, m_7$ and $m_{22}$, where the number indicates the MMD (mass median diameter) value of that log-normal mode) covering the whole range of sizes from 0.1 to 100 μm and nudged winds were used. The results shown in the Supplement Table S.GL.7 are consistent with the impact of larger particles on dust emissions and loadings in UKESM, and allow us to discuss the role of each mode independently. The total emissions for IPSL-4DU are dominated by the largest particles, those of mode $m_{22}$, but are promptly removed from the atmosphere through their sedimentation which is very rapid compared with the typical lifetime of mineral dust, as shown in Table 6. When comparing the total load for each mode, actually the coarse size mode $m_{2.5}$ is more abundant than $m_{22}$. Amongst all the modes, mode $m_7$ has the largest contribution, with 2/3 of the total, which is comparable to the large particles represented in the CNRM-6DU model, consistent with Adebiyi and Kok (2020). Note that the dust loads in CNRM-6DU model are larger than in CNRM-3DU, to a degree that cannot be explained solely by the larger emissions of CNRM-6DU. An explanation for this difference is that the bin that includes particle sizes from 2.5 to 20 μm in CNRM-3DU is split into different bins in the CNRM-6DU model, which have different life times in the atmosphere, and that non-conservative transport could create larger aerosol mass in the CNRM-6DU configuration. In contrast to emissions, optical properties are dominated by the contributions of accumulation to coarse size particles compared to the largest particles of mode $m_{22}$ that does not play a large role in its contribution to aerosol extinction. Those values are then used for assessments about modal contributions to direct radiative effects.

Mineral dust aerosol interaction with solar and terrestrial radiation results in both absorption and scattering of light. These interactions are strongly dependent on dust mineralogical composition and particle size distribution, hence they differ regionally (Ginoux, 2017; Kok et al., 2017). We estimated the respective roles of the different modes (that represent different particle size ranges), and note that in the case of multi-modal distributions the estimations of direct radiative effects (DRE) by each mode is, somewhat, non-linear (Di Biagio et al., 2020). This is illustrated by the sum of the contribution of the DRE from each mode which is not exactly equal to the multi-modal dust contribution. Appendix A shows how, with an estimation of DRE per mode based on the combination of two different methods, we determined modal values of DRE that, when combined, come close to the multi-modal DRE estimation. This is summarised in Table 7 where the estimates per-mode DRE for each method are shown together with their mean. The sum of these mean values per mode is now consistent with the multi-modal DRE. It is remarkable how the estimations of DRE at TOA-SW (top of the atmosphere in the short-wave) for $m_7$ for each method differ by a factor of 2. The non-linear effects at the surface in the SW are also important with differences in the sum of the 4 modes between methods of 0.3 Wm$^{-2}$.

The analysis of direct radiative effects (DRE) by mode, shown in Table 7, indicates that the largest particles (mode $m_{22}$) have a minor impact on the DRE in both LW and SW according to IPSL-4DU model. In contrast, the inclusion of the mode with the smallest particles contributes to SW cooling although it is the coarse size mode that dominates the net direct radiative effects at the top of the atmosphere. At the surface however, the mode $m_7$ has the largest effect on both SW and LW but its net contribution (LW+SW) is smaller than the coarse mode $m_{2.5}$. It is important to note that the DRE shown in Table 7 is estimated without scattering in the LW (only absorption). To neglect the LW scattering in the case of mineral dust implies an underestimation of TOA-DRE-LW (Dufresne et al., 2002), mostly in cloud conditions.

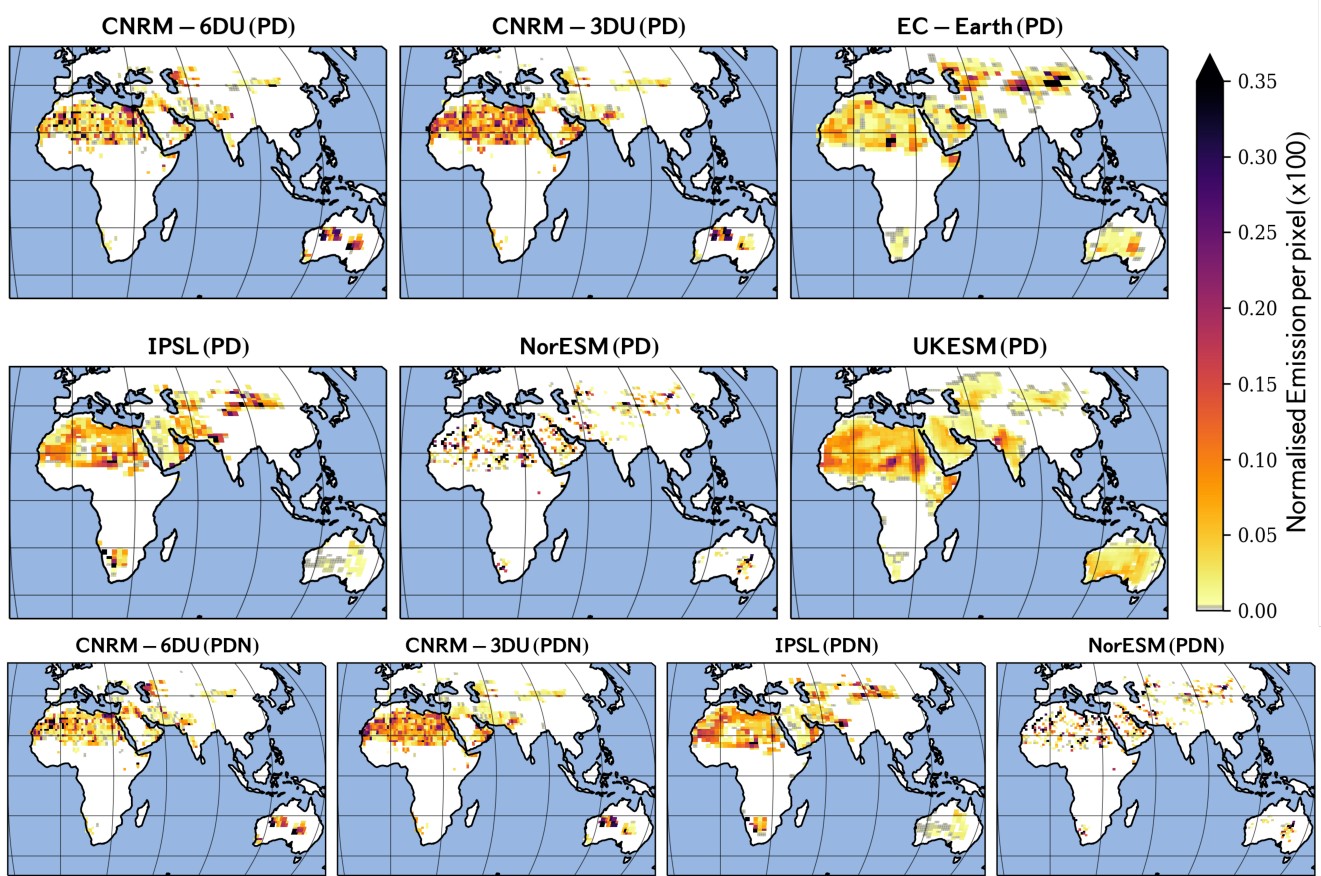

**Figure 7.** Normalized map of emissions (x100) over NorESM grid resolution. On the top: experiment with present day aerosol and chemistry forcings (PD), on the bottom the PDN experiment. We used a conservative near-neighbour interpolation to create emission maps that preserve global values on higher resolutions, then the maps were normalized to have a common comparison scale. The color-bar represents the normalized emission tendencies per grid with range [0,100]. The figure S.E5 is the correspondent of this figure for PI experiment.

## 5.2 Dust global spatial patterns

A global picture of the dust cycle is shown in Figures 5 and 6, which describe temporal mean properties of dust in CRESCENDO ESMs (PD simulations) over the 15 years. The spatial resolution and vertical levels of the models are introduced in Table 1.

First, those models that have a sectional representation of the DPSD (CNRM-6DU, CNRM-3DU and UKESM) are shown in Figure 5. For all these models, emission and dry deposition show strong spatial correlations because gravitational settling of large particles occurs close to dust sources, whereas wet scavenging dominates the deposition process over the oceans. The extension of regional emissions over the Sahel and Somalia is more pronounced for UKESM than for the CNRM models. Although the Chalbi Desert in Kenya is also a location for emission in the CNRM models, the extent over which emissions occur in the UKESM is significantly larger. The figure also suggests differences in deposition for the CNRM models: the

CNRM-3DU model has higher values of dry deposition than CNRM-6DU but the opposite is true for wet deposition. These
differences in wet deposition are pronounced over the North Atlantic and the Indian Ocean. In contrast, wet deposition is more
intense over the Sahel and the Indian sub-continent in the UKESM model which indicates the strong role of the monsoon at
scavenging dust. It is also noticeable that the CNRM-3DU annual mean wet deposition decreases from West to East over the
Indian Ocean while the inverse is true for UKESM. Despite systematic smaller values for UKESM optical depth compared
to CNRM-3DU, they have rather similar spatial distributions, except in Australia. Analogous of figures 5 for PI and PDN are
shown in Figures S.GL.3 and S.GL.5, respectively. The figures of the PI experiment demonstrate no differences with the PD
experiment, but the PDN experiment for CNRM models show smaller values of deposition and optical depth but with similar
spatial patterns, due to the decrease of their dust emissions with nudged winds.

Second, the models with a modal description of the DPSD (IPSL, EC-Earth3-AerChem and NorESM) are shown in Figure 6.
Dust emissions from EC-Earth are more intense in Asia than for the other models whereas EC-Earth has the smallest emissions
from the Northern Sahara. This causes the trans-Pacific transport of dust to peak in this model compared to others, and the
transport across the Atlantic to be smaller. Northern Sahara emissions from NorESM are more localised but with larger peak
values. Like for sectional models, dry deposition correlates well spatially with emissions whereas wet deposition dominates
over oceanic regions. EC-Earth shows both larger wet deposition and optical depth over East Asia extending into the Sea of
Japan. For all models with a modal scheme, wet deposition over the Indian ocean is mostly occurring over its Western part.
Analogous of figures 6 for the PI and PDN experiments are shown in Figures S.GL.4 and S.GL.6, respectively. Here, the results
of PI and PDN draw a picture with similar global properties of dust cycle to the PD experiment.

## 5.3  Dust emissions

The dust emission rate is defined as the surface mass flux of mineral dust in the vertical direction $F_d$. This flux is derived in
climate models as a function of surface winds but there are different schemes depending on the complexity of the descrip-
tion. Shao and Dong (2006) classify all dust emission schemes in three different categories named $\alpha, \beta$ and $\gamma$ schemes. The
$\alpha$-schemes are those where $F_d$ is directly described in terms of the wind speed (with a non-linear function including a friction
velocity threshold) with an imposed empirical size distribution at emission. IPSL-INCA uses this approach. The $\beta$-schemes
instead estimate the vertical flux from the dust horizontal mass-flux which itself can be parameterised depending on a geograph-
ical erodibility factor and the surface wind. Although this erodibility factor depends on soil properties and moisture, sub-daily
global patterns of dust emission are tightly correlated with wind fields, and therefore with the atmospheric general circulation
(Shao et al., 2011). Examples of $\beta$-schemes are those described by Zender et al. (2003) and Woodward (2001) that are used
respectively by NorESM and UKESM models. It is also used in the EC-Earth model whose horizontal flux is estimated with the
scheme described by Marticorena and Bergametti (1995) which distributes particles in four bins with values up to 8 µm. Those
values are mapped in the modes described in the Table S.MD.9. In the case of UKESM the horizontal flux is also calculated
based on Marticorena and Bergametti (1995) into 9 bins of diameters between 0.064 to 2000 µm but mapped for transport into
6 bins described in Table S.MD.9. Similarly the CNRM models have a drag partition according to Marticorena and Bergametti
(1995) but the size distribution at emission follows that defined by (Kok, 2011). The $\gamma$-schemes aim to describe the physical

**Table 9.** Total wet deposition [Tg yr$^{-1}$] for Present Day (PD) simulations. Over oceanic regions, see Figure 2. The numbers in brackets show the fraction of global deposition over the ocean. The numbers in parentheses indicate the ranking order of contribution to the global total wet deposition by region from the highest to the lowest. The equivalent Tables for the PI and PDN experiments are in Supplementary Information: Tables S.DD.1 and S.DD.2, respectively.

| | CNRM-3DU (PD) | | EC-Earth (PD) | | IPSL (PD) | | NorESM (PD) | | UKESM (PD) | | MM-$\mu \pm \sigma^{\dagger}$ | | CNRM-6DU (PD) | |
|---|---|---|---|---|---|---|---|---|---|---|---|---|---|---|
| Global Earth | 753.8 | | 493.2 | | 968.3 | | 275.7 | | 949.8 | | 688±300 | | 2108.9 | |
| Land | 541.3 | | 272.9 | | 575.7 | | 203.9 | | 673.6 | | 453±200 | | 1397.1 | |
| Ocean | 212.5 | [28%] | 220.3 | [45%] | 392.6 | [40%] | 71.8 | [26%] | 276.1 | [29%] | 235±120 | | 711.8 | [33%] |
| North Atlantic | 65.4 | (1) | 61.7 | (2) | 156.1 | (1) | 23.7 | (1) | 103.4 | (1) | 82±50 | (1) | 207.4 | (1) |
| South Atlantic | 5.1 | (5) | 14.6 | (5) | 47.0 | (2) | 2.5 | (4) | 11.3 | (4) | 16±18 | (4) | 9.1 | (6) |
| North-Indian Ocean | 47.8 | (2) | 16.6 | (4) | 36.5 | (4) | 16.2 | (2) | 33.1 | (3) | 30±14 | (3) | 187.2 | (2) |
| South-Indian Ocean | 13.9 | (4) | 4.1 | (6) | 18.5 | (5) | 2.4 | (5) | 11.1 | (5) | 10±7 | (5) | 39.3 | (4) |
| Pacific West | 21.1 | (3) | 70.5 | (1) | 39.1 | (3) | 7.3 | (3) | 41.5 | (2) | 36±24 | (2) | 93.6 | (3) |
| Pacific North-East | 0.2 | (8) | 21.0 | (3) | 12.2 | (6) | 1.0 | (6) | 10.2 | (6) | 8.9±8 | (6) | 2.9 | (7) |
| Pacific South-East | 2.5 | (6) | 3.0 | (7) | 3.8 | (8) | 0.9 | (7) | 5.9 | (7) | 3.2±2 | (8) | 9.9 | (5) |
| Antarctic Ocean | 2.2 | (7) | 2.5 | (8) | 7.3 | (7) | 0.6 | (8) | 4.3 | (8) | 3.4±3 | (7) | 5.4 | (8) |
| Ocean. North. Hemis. | 162.9 | | 188.5 | | 287.4 | | 59.2 | | 218 | | 183±80 | | 569.1 | |
| Ocean. South. Hemis. | 49.5 | | 31.8 | | 104.2 | | 12.5 | | 58.1 | | 51±30 | | 142.1 | |

$^{\dagger}$ Statistic is not including CNRM-6DU.

process driving the size resolved vertical flux but they require additional information of the underlying soil properties and are not used by CRESCENDO-ESM.

Despite the different schemes all of them agree that the regions where most dust is uplifted are subtropical arid and semi-arid regions. Such regions are characterised by atmospheric stability and scarce rainfall. This global pattern is however modulated by the Inter-Tropical Convergence Zone (ITCZ) oscillations, monsoons, and orography, as visible in Figures 5 and 6. Because the Himalayan mountains filter the water-vapour transport from the Indian Ocean all the models have important dust sources in Northern Asia (such as the Taklamakan and Gobi deserts) but the specific location of Asian sources, and their relative contribution to global emissions differs significantly between models.

Nowadays, we understand how regional climate influences the dust emissions and their variability, together with the atmospheric systems linked to dust emission episodes. But dust emission modelling still constitutes an active research field (Shao, 2008). In particular, the dust particle size distribution (DPSD) at emission is critical for a better description of the global dust cycle (Mahowald et al., 2014) but its modelling needs to be improved for three main reasons: (1) because there is not an unified approach; (2) because there are discrepancies in the role of wind speed at emission for larger dust particles (Alfaro et al., 1998, 1997); and (3), because the quantitative link between soil properties and dust emission fluxes still needs additional research.

Regardless of the several sets of parametrisations of DPSD at emission (Kok, 2011; Alfaro and Gomes, 2001; Shao, 2001, 2004), the actual modelling of dust in global climate models is highly influenced by a balance of the different elements involved (vertical flux at small scale, soil erodibility, wind fields), which explains that during last decade the estimation

**Table 10.** Total dry deposition [Tg yr$^{-1}$] for Present Day (PD) simulations. Over oceanic-regions, see Figure 2. The numbers in brackets show the fraction of global deposition over the ocean. The numbers in parentheses indicate the ranking order of contribution to the global total dry deposition by region from the highest to the lowest. The ensemble mean (and standard deviation) includes all the models except CNRM-6DU and UKESM. The equivalent Tables for the PI and PDN experiments are in Supplementary Information: Tables S.DD.3 and S.DD.4, respectively. The ensemble statistics for Global Earth and Land is not including UKESM due to their large values of gravitational settling would drive the estimate. Over ocean regions

| | CNRM-3DU (PD) | | EC-Earth (PD) | | IPSL (PD) | | NorESM (PD) | | UKESM (PD) | | MM-$\mu \pm \sigma$ | | CNRM-6DU (PD) | |
|---|---|---|---|---|---|---|---|---|---|---|---|---|---|---|
| Global Earth | 1925.8 | | 633.5 | | 590.6 | | 1092.5 | | 6566.3 | | 1061±620$^{\ddagger}$ | | 2025.9 | |
| Land | 1678.1 | | 555.8 | | 523.1 | | 986.6 | | 6366.1 | | 936±540$^{\ddagger}$ | | 1681.1 | |
| Ocean | 247.7 | [7.7%] | 77.7 | [12%] | 67.5 | [11%] | 105.9 | [10%] | 199.4 | [3%] | 140±80$^{\dagger}$ | | 344.8 | [17%] |
| North Atlantic | 99.5 | (1) | 31.7 | (1) | 31.6 | (1) | 28.4 | (2) | 81.9 | (1) | 54±34$^{\dagger}$ | (1) | 120.3 | (1) |
| South Atlantic | 5.5 | (5) | 2.3 | (4) | 5.3 | (3) | 2.5 | (4) | 1.9 | (5) | 3.5±1.8$^{\dagger}$ | (5) | 2.3 | (6) |
| North-Indian Ocean | 63.6 | (2) | 14.3 | (2) | 13.8 | (2) | 49.5 | (1) | 51.3 | (2) | 38±23$^{\dagger}$ | (2) | 106.7 | (2) |
| South-Indian Ocean | 18.8 | (3) | 1.4 | (7) | 0.9 | (6) | 0.8 | (6) | 9.1 | (4) | 6.2±8$^{\dagger}$ | (4) | 26.2 | (3) |
| Pacific West | 11.0 | (4) | 13.3 | (3) | 2.3 | (5) | 3.9 | (3) | 12.5 | (3) | 8.6±5.1$^{\dagger}$ | (3) | 24.5 | (4) |
| Pacific North-East | 0.3 | (8) | 2.2 | (5) | 2.7 | (4) | 0.9 | (5) | 1.5 | (6) | 1.5±1.0$^{\dagger}$ | (6) | 0.4 | (7) |
| Pacific South-East | 3.0 | (6) | 0.4 | (7) | 0.5 | (7) | 0.6 | (7) | 0.6 | (7) | 1.0±1.1$^{\dagger}$ | (7) | 4.9 | (5) |
| Antarctic Ocean | 0.1 | (8) | 0.2 | (8) | 0.3 | (8) | 0.1 | (8) | 0.4 | (8) | 0.2±0.1$^{\dagger}$ | (8) | 0.2 | (8) |
| Ocean. North. Hemis. | 199.5 | | 71.3 | | 58.3 | | 98.6 | | 172.4 | | 120±63$^{\dagger}$ | | 280.9 | |
| Ocean. South. Hemis. | 48.1 | | 6.4 | | 9.2 | | 7.3 | | 26.9 | | 20±18$^{\dagger}$ | | 63.8 | |

$^{\ddagger}$ Statistic is not including CNRM-6DU and UKESM. $^{\dagger}$ Statistic is not including CNRM-6DU.

of dust emissions when online coupled with meteorological fields have improved their results significantly. On the one hand the modelled wind surface friction velocity and speed agree better with actual meteorological conditions, e.g (Knippertz and Stuut, 2014b), and on the other hand, the description of the soil surface properties has become more accurate due to both, improvements in the soil texture databases, and the use of satellite retrievals to better describe the roughness length, e.g Prigent et al. (2005); Menut et al. (2013).

All those facts explain why the comparison (Table 8) of the emissions (PD experiment) over large regions is fairly consistent among models: they agree on the main source of mineral dust being located in the Saharan desert but representing from 39% of total global emissions in the EC-Earth model to 66% in CNRM-3DU. Previous studies (Shao et al., 2011) estimated the contribution of Africa to dust emissions in the range from 50% to 68% but also including Namibia Desert emissions. The consistency is larger when we consider larger regions like hemispherical contributions where all the models show more than 85% global dust emissions from the Northern Hemisphere. When smaller regions are considered, the differences in relative contributions between models increase, which is also expected when turbulence at small scale and/or convection (Allen et al., 2015) plays a role in dust events. If we evaluate total values rather than relative contributions, the driving factor that explains differences between modelled emissions is the upper threshold of particle sizes at emission.

Dust emissions by region (which are shown in Figure 3) and their intensities (in Tg yr$^{-1}$) are listed in Table 8 for the PD experiment. The most intense source of dust for the EC-Earth model is located over the Gobi Desert, while North Sahara, a

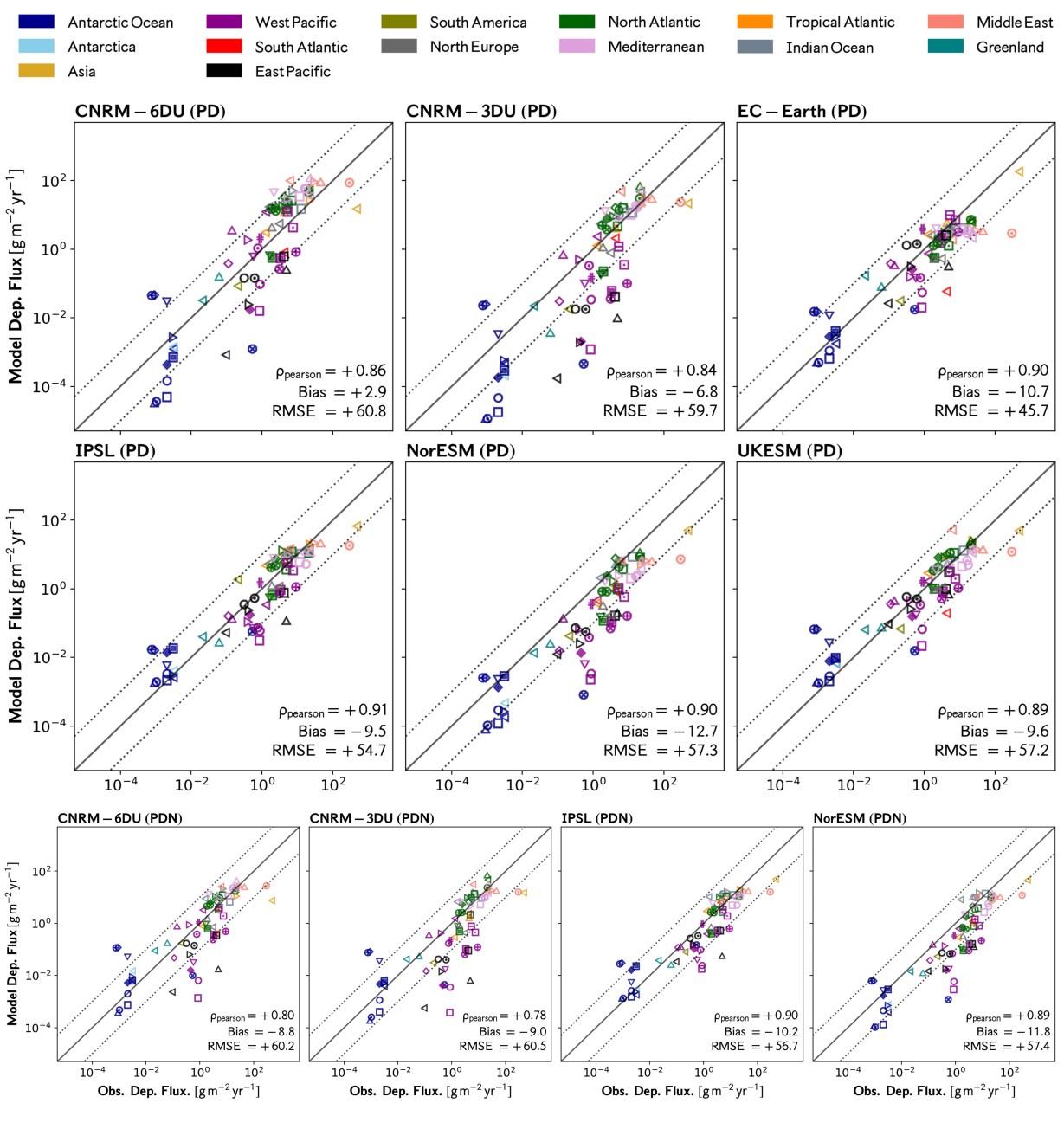

**Figure 8.** Comparison of estimated total annual deposition flux with CRESCENDO ESMs with the dataset presented by Huneeus et al. (2011), whose stations are mapped in Figure 1 (left panel). The model values taken are those from the PD experiment (top part) and the PDN experiment for bottom row. Figure S.D11 is the analogous of this figure but for the PI experiment.

key emitting region in all other models, constitutes only the 4th most intense region in emissions (after the Taklamakan and the Kyzyl-Kum). The Bodélé is remarkably an important dust source across all CRESCENDO ESMs. As expected from the analysis of dust optical depth over Asian regions, the Taklamakan, Kyzyl Kum and Thar deserts exhibit substantial differences. Regarding UKESM, it has an additional and extended dust source over the Somalia Desert (see Fig. 5) which is only a relatively small source in other models. Analogues of Tables 8 for the PDN and the PI experiments can be found in Tables S.DE.1 to S.DE.4, respectively, showing similar model differences.

If we want to compare realistically global climate model emissions over smaller regions, we need to account for the different model resolutions. We opted to display normalised emission estimations over a common grid for all the models. Our method interpolates the emission flux from each model grid to that with the highest spatial resolution (NorESM). We use a near-neighbour interpolation method which conserves the flux in each model when compared to the flux integrated over the original model resolution. This method is not introducing any ad-hoc information on how the emission tendency is distributed within the original grid-pixel. A monthly time-series of normalised emitted dust mass per grid-pixel, with respect to global monthly emissions, is produced using this method. These normalised emissions over a common grid allow us to pick up differences over locations that are caused either by the formulation of the source function or by the dust particle size distribution imposed during the emission process.

A direct comparison of dust emission maps with observations is challenging because it would require the translation of the observed frequency of dust events into a dust emission flux rate (Evan et al., 2015). Assuming the hypothesis of Evan et al. (2015) for this mapping, the hot spots of their Spinning Enhanced Visible and InfraRed Imager (SEVIRI) emission normalised product can be compared with our normalised maps (in terms of relative contribution of different pixels over North Africa). In particular they suggest that beyond Bodélé Depression an important source is at Hoggar Mountains (west of Bodélé Depression). This feature is only captured by the CNRM models.

The annual average of these monthly maps is presented in Figure 7 for PD and PDN experiments. The models CNRM-6DU and CNRM-3DU show similar values per grid-cell, which indicates the use of the same information on soil properties, but the normalised emissions although similar are not identical, reflecting the differences in dust size distribution at emission. In these models, the normalised emissions over Australia are higher than for the other models, and this difference is also appearing in the optical depths simulated at the AERONET station of Birdville. Their description of semi-desert areas in Northern India has many similarities to the IPSL model. Emission tendencies from the UKESM model extend to areas where other models do not simulate emissions, and the pattern of emissions is more smooth. In particular, significant emissions occur over the Sahel, Ethiopia, Somalia, and over India. For these regions, higher dust emissions in UKESM could have a stronger impact on African and Asian monsoons. The most granulated pattern is found for the NorESM model due to the higher resolution of the source functions implemented. The last row in Figure 7 corresponds to the normalised emission maps for the PDN experiment, and it indicates that although there are important differences between the PD and PDN experiments in terms of total emissions, see Table 8, the spatial patterns of emissions are similar once they are normalised. We can ascertain this fact by comparing the CNRM-6DU normalised emission maps for the PD and the PDN experiment. The analysis for the PI experiment is in the Supplementary Information: Figure S.DE.5.

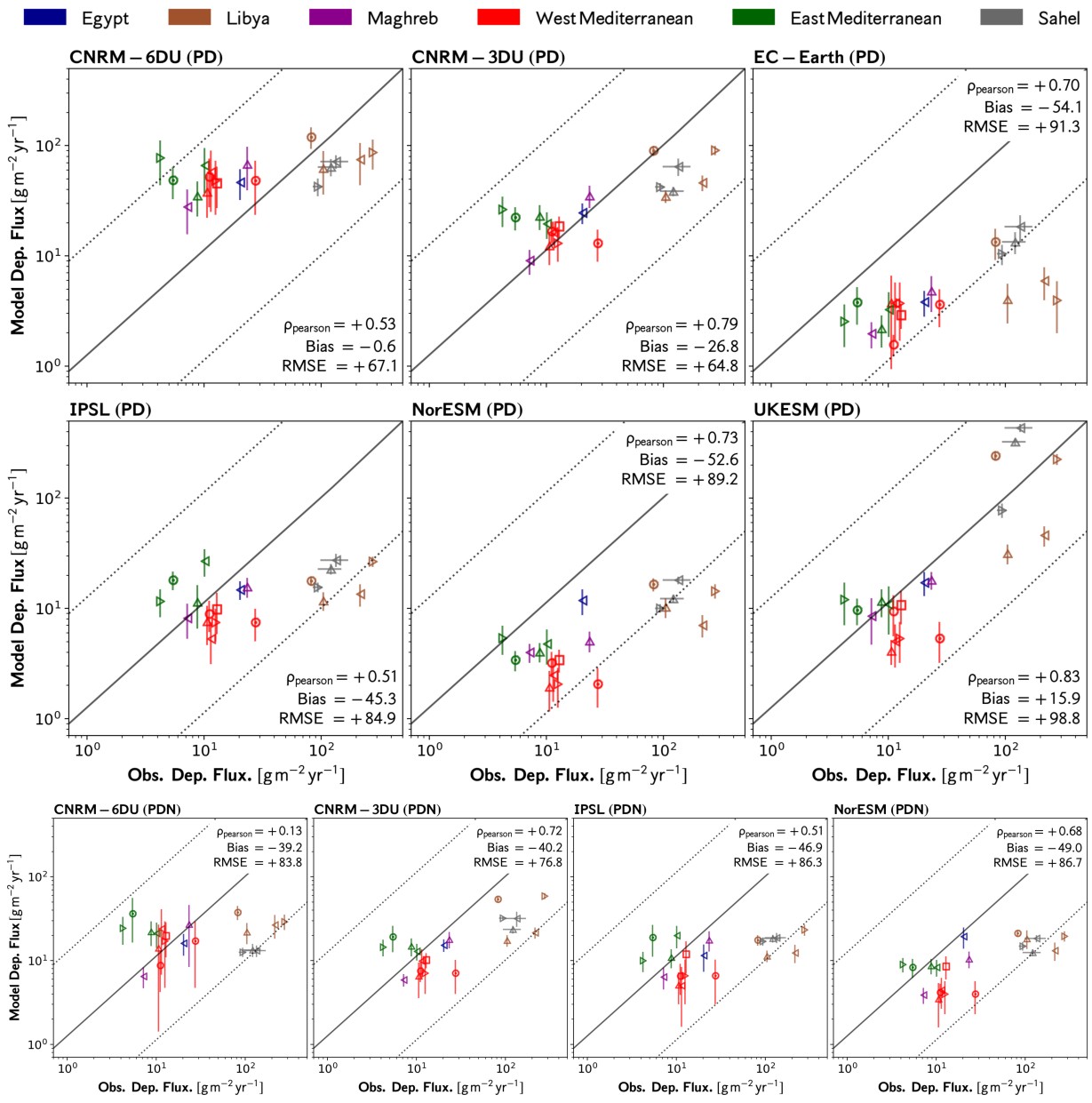

**Figure 9.** Comparison of estimated total annual deposition flux with CRESCENDO ESMs with the dataset stations shown in Figure 1 (right panel). The model values taken are those from the PD experiment (top part) and PDN experiment for bottom row. Figure S.D.11 of the Supplement is the analogous figure for the PI experiment. Vertical bars on the bottom panel represent the year to year internal variability captured by each model. The grey horizontal bars displayed for the Sahel stations represent the year to year variations in the observations.

**Table 11.** Statistical properties of the comparison of the CRESCENDO-ESMs total deposition against the network-SET-M (see Figure 1 panel b). Statistic metrics used in this table are described on Table 5. Pearson Correlation Coefficient ($\rho$), bias ($\delta$) [$\text{gm}^{-2}\text{yr}^{-1}$], normalised bias ($\delta_N$), Ratio standard deviations ($\Sigma$), Normalised mean absolute error ($\theta_N$) and Root mean square error (RMSE=$\eta$).

| Model | Exp. | Deposition Network-SET-M | | | | | |
|---|---|---|---|---|---|---|---|
| | | $\rho$ | $\delta$ | $\delta_N$ | $\Sigma$ | $\theta_N$ | $\eta$ |
| CNRM-6DU | PD | +0.53 | -0.58 | -0.01 | +0.27 | +0.90 | +67.14 |
| CNRM-3DU | PD | +0.79 | -26.83 | -0.45 | +0.31 | +0.63 | +64.79 |
| EC-Earth | PD | +0.70 | -54.12 | -0.91 | +0.06 | +0.91 | +91.26 |
| IPSL | PD | +0.51 | -45.25 | -0.76 | +0.09 | +0.83 | +84.90 |
| NorESM | PD | +0.68 | -52.10 | -0.87 | +0.07 | +0.88 | +89.01 |
| UKESM | PD | +0.83 | +15.91 | +0.27 | +1.63 | +0.88 | +98.75 |
| CNRM-6DU | PDN | +0.13 | -39.22 | -0.66 | +0.11 | +0.84 | +83.81 |
| CNRM-3DU | PDN | +0.72 | -40.25 | -0.67 | +0.19 | +0.73 | +76.79 |
| IPSL | PDN | +0.51 | -46.90 | -0.79 | +0.07 | +0.84 | +86.30 |
| NorESM | PDN | +0.62 | -48.49 | -0.81 | +0.07 | +0.83 | +86.73 |
| CNRM-6DU | PI | +0.47 | +5.22 | +0.09 | +0.29 | +0.93 | +67.54 |
| CNRM-3DU | PI | +0.74 | -23.23 | -0.39 | +0.31 | +0.66 | +63.31 |
| EC-Earth | PI | +0.66 | -54.17 | -0.91 | +0.06 | +0.91 | +91.39 |
| IPSL | PI | +0.36 | -45.81 | -0.77 | +0.10 | +0.84 | +85.98 |
| NorESM | PI | +0.76 | -52.35 | -0.88 | +0.07 | +0.88 | +88.98 |
| UKESM | PI | +0.84 | +16.05 | +0.27 | +1.65 | +0.88 | +100.8 |

## 5.4 Dust deposition

Previous studies (Huneeus et al., 2011; Albani et al., 2014) show that total deposition of dust, when compared with in-situ measurements, agrees globally only to within a factor of 10. Part of the reason is that dry and wet deposition are dependent on the dust particle size distribution, whose representation is challenging for current global climate models. Indeed, processes driving dry deposition such as turbulent motions of particles and gravitational settling are both particle size dependent, as the aerodynamic resistance and the terminal velocity due to friction depend on the effective dust particle diameter. Wet deposition during precipitation events also depends on the size of the particle (Seinfeld and Pandis, 1998) but measurements of aerosol lifetimes below clouds are scarce. Furthermore, other aerosol processes inside clouds modify the aerosol size distribution, as well as, their optical properties essentially due to potential aggregation of water-coated aerosols (Mahowald et al., 2014). Thereby, the first step of the analysis is a comparison of dry and wet deposition at a regional scale.

In fact, as the gravitational settling of large particles is dominant close to dust sources, regions remote from the main emission sources are well suited to compare models with different emission schemes, and evaluate their respective total dry and wet deposition. Close to dust sources the upper threshold of the emitted dust particle sizes plays a role in the comparison with measurements. In particular, wet deposition over oceanic regions is enhanced relative to dry deposition which motivates targeting these specific regions for comparison. Tables 9 and 10 show the regional analysis of wet and dry deposition (including the sedimentation/gravitational settling) over oceans. These results are globally consistent with those shown by Shao et al.

**Table 12.** Statistical properties of the comparison of the CRESCENDO-ESMs total deposition against the network-H2011 (see Figure 1 panel a). Statistic metrics used in this table are described on Table 5. Pearson Correlation Coefficient ($\rho$), bias ($\delta$) [$gm^{-2}yr^{-1}$], normalised bias ($\delta_N$), Ratio standard deviations ($\Sigma$), Normalised mean absolute error ($\theta_N$) and Root mean square error (RMSE=$\eta$).

| Model | Exp. | Deposition Network-H2011 | | | | | |
|---|---|---|---|---|---|---|---|
| | | $\rho$ | $\delta$ | $\delta_N$ | $\Sigma$ | $\theta_N$ | $\eta$ |
| CNRM-6DU | PD | +0.86 | +2.88 | +0.19 | +0.46 | +1.38 | +60.82 |
| CNRM-3DU | PD | +0.84 | -6.82 | -0.44 | +0.24 | +0.91 | +59.66 |
| EC-Earth | PD | +0.90 | -10.71 | -0.70 | +0.36 | +0.73 | +45.74 |
| IPSL | PD | +0.91 | -9.54 | -0.62 | +0.16 | +0.78 | +54.69 |
| NorESM | PD | +0.90 | -12.68 | -0.83 | +0.11 | +0.84 | +57.26 |
| UKESM | PD | +0.89 | -9.58 | -0.62 | +0.16 | +0.81 | +57.21 |
| CNRM-6DU | PDN | +0.80 | -8.78 | -0.57 | +0.16 | +0.83 | +60.16 |
| CNRM-3DU | PDN | +0.78 | -9.00 | -0.59 | +0.19 | +0.90 | +60.53 |
| IPSL | PDN | +0.90 | -10.23 | -0.67 | +0.13 | +0.79 | +56.67 |
| NorESM | PDN | +0.89 | -11.80 | -0.77 | +0.11 | +0.83 | +57.42 |
| CNRM-6DU | PI | +0.86 | +4.04 | +0.26 | +0.46 | +1.43 | +60.58 |
| CNRM-3DU | PI | +0.84 | -6.18 | -0.40 | +0.25 | +0.94 | +59.67 |
| EC-Earth | PI | +0.90 | -10.28 | -0.67 | +0.42 | +0.70 | +43.04 |
| IPSL | PI | +0.92 | -9.56 | -0.62 | +0.16 | +0.78 | +54.66 |
| NorESM | PI | +0.91 | -12.58 | -0.82 | +0.11 | +0.84 | +57.12 |
| UKESM | PI | +0.89 | -9.37 | -0.61 | +0.17 | +0.82 | +57.04 |

(2011). The two main oceanic regions where dust deposition occurs are the North Atlantic and the Indian Ocean even though the EC-Earth model simulated the largest dust wet deposition over the West Pacific Ocean. For all models, the fraction of dry and wet deposition over the oceans is smaller than over land. Wet deposition over oceans represents 40% and 45% of the total wet deposition for IPSL and EC-Earth, respectively. But for NorESM it represents 26% of the global wet deposition. Dry deposition over oceans ranges from 3% to 12% of global dry depositions. For the UKESM model, the dry deposition over land is 97% of the total dry deposition, due to the gravitational settling of large particles close to emission regions. Tables 9 and 10 also show slightly better consistency in the total dry deposition over oceans in the model ensemble (from 67 to 250 in $Tg\,yr^{-1}$) than in the wet deposition (72 to 392 in $Tg\,yr^{-1}$), as we are excluding CNRM-6DU from the model ensemble. Results for the PDN and the PI experiments are included in Tables S.DD.1 to S.DD.4.

### 5.4.1 Network of Dust deposition observations

Figure 8 shows the total annual deposition for the PD and PDN experiments for the locations shown on panel (a) of Figure 1, and Figure 9 shows the total annual deposition for PD and PDN experiments for the locations shown on panel (b) of Figure 1. Figures S.DD.11 and S.DD.12 show the analogues for the PI experiment. Qualitatively the global results are similar to Huneeus et al. (2011) where at most of the stations the modelled deposition is within a factor of 10 of the observed deposition flux (in the figures, the region between the dotted lines). As a consequence the estimated Pearson correlation of deposition flux calculated over log-values for the full network shows a reasonable value for all models.

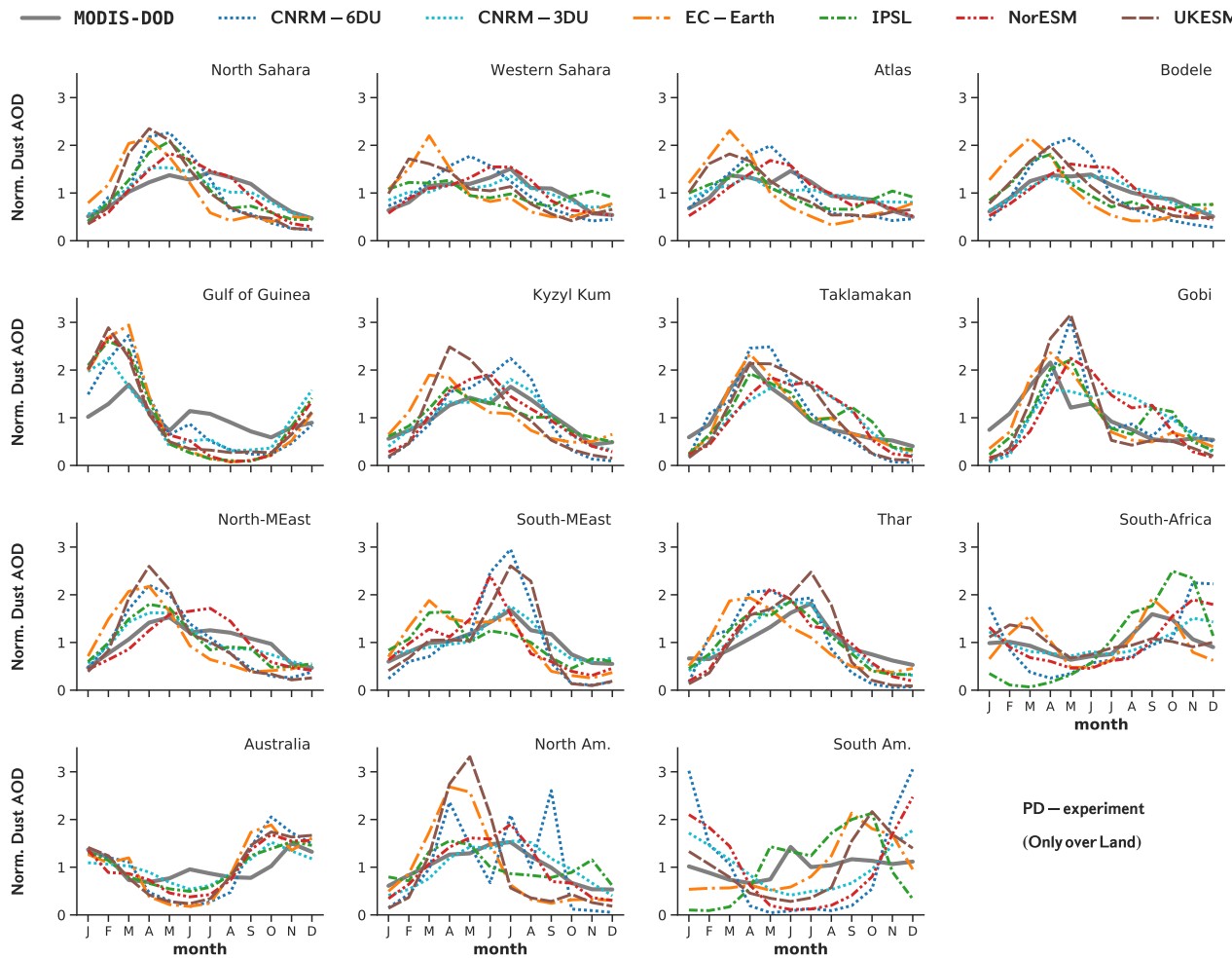

**Figure 10.** Seasonal cycle relative to the annual mean value of Dust Optical Depth as modelled by CRESCENDO ESMs over 15 regions. These seasonal cycles are compared against the DOD product of derived dust optical depth over land based on MODIS deep-blue retrievals (Pu and Ginoux, 2018b), see supplementary information for the description of how these products are derived and the analogous of this figure for PDN and PI experiments.

All the models agree that Antarctica and the Southern Ocean have the lowest values of total deposition. While UKESM
and IPSL tend to slightly overestimate the total flux in these remote regions, the CNRM models tend to underestimate the
flux. However, their most prominent property Antarctic regions is a much larger range of values than the range reported by
the observations. Additional research is need to evaluate if this is a consequence of their semi-Lagrangian model implemented
in their dynamical core which add a non-uniform bias, or instead it is just a combination of the dust source locations in the
Southern Hemisphere and wind fields modelled.

Regarding the Pacific region closer to North America (named East Pacific) NorESM, CNRM-6DU and CNRM-3DU tend
to underestimate the deposition. In the case of West Pacific region NorESM systematically underestimates the deposition flux.
Regarding CNRM models they underestimate the total deposition over the northern hemisphere part of West Pacific but not
in the southern part of West Pacific due probably to the enhanced emissions of these models over Australian deserts. All the
models except the EC-Earth model underestimate the deposition over the single Asia station, and EC-Earth model report good
values of total deposition over the northern West Pacific as it has the largest relative contributions over Gobi desert between all
the models.

All the models show a good agreement in the Atlantic region (both North and Tropical regions) and the Middle East although
the UKESM and EC-Earth models underestimate the values at the single station in the South Atlantic. The deposition fluxes
over the Indian Ocean are fairly well described by all models.

If we compare the observations against the modelled total deposition obtained from the experiment with nudged winds (last
row in Figure 8) the correlation coefficients are similar, but differences between models are reduced, specially for the CNRM
models. This is illustrated in Table 12 with a negative bias for all models (from -8.8 to -11.8 $\mathrm{g m^{-2} yr^{-1}}$), and the ratio of
standard deviations $\Sigma$ range between 0.11 and 0.19 (for PD experiment between 0.11 and 0.46). The CNRM-6DU model is the
only one with a positive bias ($\delta$ in Table 12) against the Network-H2011.

In Figure 9 we analyse the ability of the ESMs to reproduce deposition fluxes regionally and closer to sources (for the PD
and PDN experiments). We focus on the Mediterranean Sea, but we include three additional stations over the Sahel where
observational annual differences can be compared. The analysis reveals that only the UKESM model reproduces the full range
of observed deposition fluxes. All the other models underestimate total depositions fluxes over stations where fluxes exceed
100 $\mathrm{g\,m^{-2} yr^{-1}}$, and only the CNRM-3DU model estimates well the observed dust deposition in the northern Mediterranean
Sea. Over the Sahel region, the CNRM models and UKESM provide reasonable values of total deposition flux, but UKESM
overestimates the inland deposition, whereas the other models show a more consistent bias over the whole region.

The Sahel stations include horizontal bars describing the inter-annual variability over the mean values, which can be com-
pared with vertical bars describing the variability in the models. In this case EC-Earth is the model that captures best the
year-to-year differences on mean values of dust deposition flux over the inland Sahel stations. For West Mediterranean the
CNRM-3DU has the smallest bias, whereas in the full Mediterranean region UKESM and IPSL perform well in terms of
global bias.

EC-Earth and NorESM underestimate total depositions close to source regions consistent with the applied size cutoff around
8 µm of emitted particles, and CNRM-6DU overestimates the deposition on the whole Mediterranean region. For the exper-

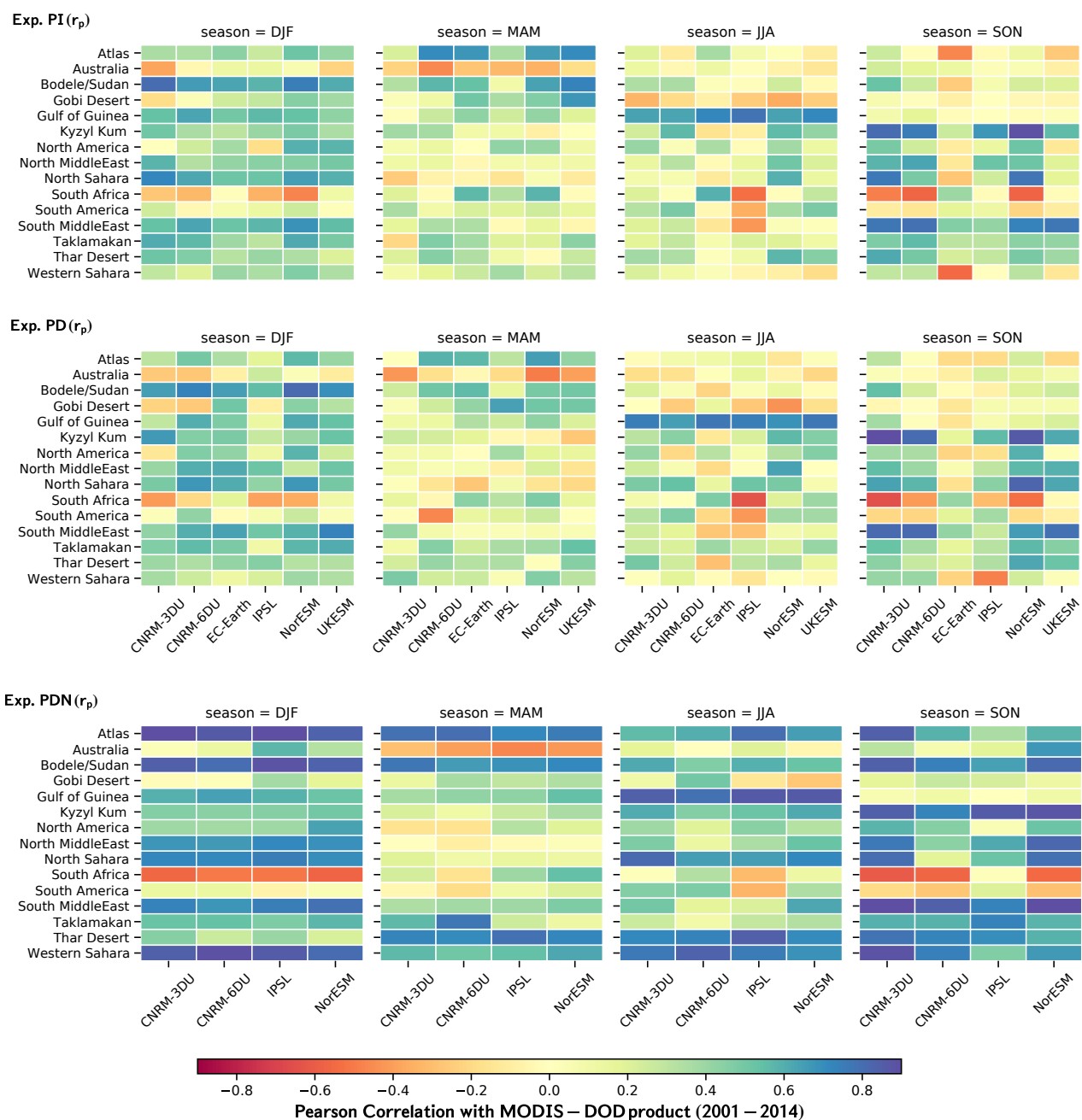

**Figure 11.** Skill of CRESCENDO-ESMs by regions calculated as the Pearson correlations between the ESM time-series of dust optical depth for each season and that from MODIS-DOD. The time interval spans from 2001 to 2014. It assess the performance of the different models to reproduce the inter-annual variability of each season against observations over dust source regions.

iment with nudged winds, we observe a better consistency between models with all of them showing similar values of total
615 deposition in the different subregions. However, this implies an underestimation over the Sahel for the CNRM-6DU model that
also has the largest inter-annual variability over the West-Mediterranean. The statistics metrics are shown in Table 11.

## 5.5 Dust optical depth

The simulated dust optical depth (DOD) by climate models has been compared previously with those retrieved through a
network of ground-based sun-photometers (Huneeus et al., 2011) but also with products derived from satellite retrievals (Pu
and Ginoux, 2018b; Peyridieu et al., 2013). There are also inter-comparisons between global climate models (Shindell et al.,
2013). The overall agreement reported by these studies between retrieved and simulated dust aerosol optical depth is within
a factor of two. Those results support the reliability of global estimations of the radiative effect from mineral dust. However,
given that it is a vertically integrated parameter, it masks larger differences present in partial column estimates.

Our study focuses first on the comparison in regions defined in Figure 3. We compare the DOD of the CRESCENDO ESMs
with satellites, as well as inter-compare simulated dust optical depth. Figure 10 shows the seasonal cycle (relative to the annual
mean value of each model) and the MODIS DOD product during the period 2001-2014, for the PD experiment (the PDN and PI
experiment are shown in Figures S.DOD.1 and S.DOD.2). We can hence analyse the seasonal amplitude relative to the annual
background signal per region for each model. The supplementary Figure S.DOD.3 shows the direct comparison of the seasonal
cycle without relative values.

Over the most prominent preferential dust source regions (first row of Figure 10), the amplitude of the seasonal variability
is systematically larger in all the models (with respect to the MODIS-DOD product) with a slight offset on the maximum
value of the seasonal cycle towards spring time, particularly over Northern Sahara. It is remarkable that in these regions
CNRM-3DU and NorESM show consistency in the seasonality with respect to MODIS-DOD, whereas EC-Earth and UKESM
show more discrepancies in the seasonal cycle in both the amplitude and the phase. The CNRM-6DU model and IPSL have
635 slight discrepancies in these 4 regions. Over the Asian deserts of Taklamakan and Gobi the seasonal maximum is reasonably
represented in the spring with a relative good agreement for EC-Earth, although the seasonality is not well represented for the
Thar Desert. The UKESM, NorESM and CNRM-3DU models overestimate summer dust optical depth over the Taklamakan
desert. A common feature between all the models is that over the Asian Desert the winter values are smaller than those of
MODIS-DOD. Previous studies (Laurent et al., 2006) concluded that the seasonal cycle of Taklamakan desert is controlled
by latter spring and summer emissions which most models capture, whereas Gobi, and the associated northern China deserts,
have maximum emissions during late winter and early spring. CRESCENDO ESMs reproduce the maximum values of DOD
in Spring for the Gobi deserts, and UKESM and EC-Earth models capture that seasonality over Taklamakan as well. Given the
structural differences in the soil properties of these Asian regions (more stony at Gobi, mostly sandy at Taklamakan) and the
additional role of snow cover over the Gobi desert, further model studies of Asian dust emissions are needed to better constrain
the way dust scheme parametrisations capture emissions in these regions. Ideally, these studies should be backed up by in-situ
surface concentration measurements. Regarding the Middle-East, the combined region of North and South Middle East is in
agreement with the Pu and Ginoux (2018b) study based on CMIP5 models.

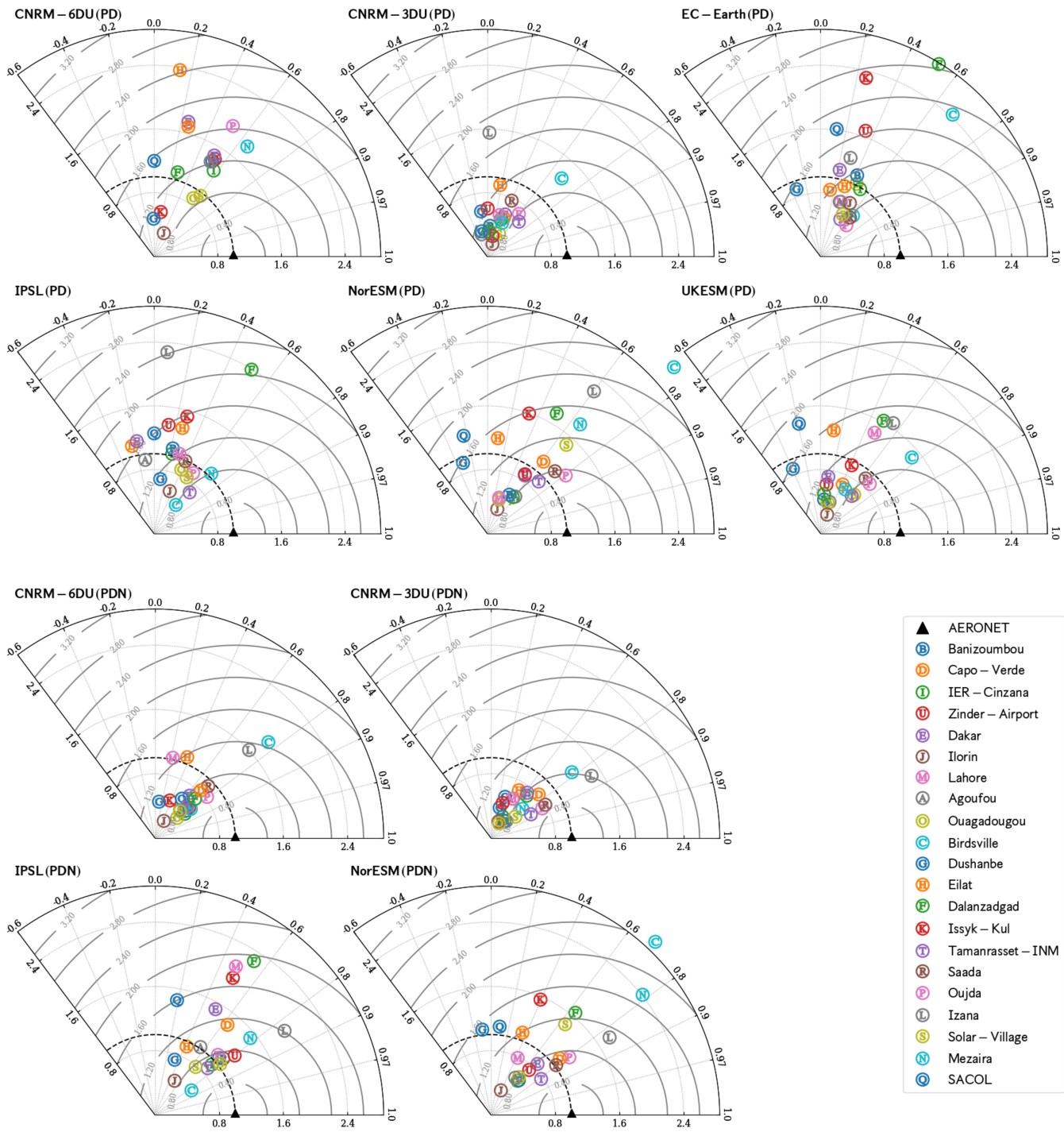

**Figure 12.** Normalised Taylor diagrams based on time series of total aerosol optical depths at 440nm. These diagrams are representing PD and PDN experiments and restricted to AERONET dusty stations shown in Figure 3 (with color green and blue).

We quantified the performance skill of the CRESCENDO ESMs by estimating the Pearson's correlation between the time-series of dust optical depth provided by each model for each of the seasons, and the same time-series of dates given by the
MODIS-DOD product for the period between January 2001 and December 2014.

Figure 11 displays the values for this Pearson's correlation. The overall assessment indicates marked differences between models for the same season and over the same region. In the case of the PD experiment (middle panel), the correlation between MODIS-DOD and CRESCENDO-ESM is positive over winter except in Australia and South Africa regions which are regions particularly challenging for the ESMs analysed as we reported negative correlations, whereas South America is one of the
regions with correlation closer to zero across all the seasons (and models). The overall correlation decreases in Spring (with respect to winter), as we notice multiple regions where the Pearson correlations are close to zero. In summer, except in the Gulf of Guinea the correlation is also smaller than in the winter season. Finally, in Autumn the performance over Middle East and the Kyzyl Kum region is improved.The better behaviour of all the models is given over Bodélé in winter season, and the Arabian region (North and South Middle-East) that shows a reasonable agreement over all year for almost all models.
Most of the features remain similar with pre-industrial aerosol-chemistry forcings (PI experiment) and the CNRM-6DU and CNMR-3DU behaves identical in the PI experiment.

The agreement with satellite platforms is significantly improved for the PDN simulations and the consistency between models is enhanced. In particular, the Saharan region shows a marked improvement in the simulated dust optical depth. Australia and South Africa are still the regions where most discrepancies are found, and South America systematically has the correlation
closest to zero.

We extended the analysis based on the Pearson correlation by using the Spearman coefficient which allows detecting non-linear correlations. The results for the Spearman rank coefficient, can be found in supplementary information Figure S.DOD.7., yield to similar conclusions, and both methods are consistent.

### 5.5.1  Network of Aerosol Optical Depth

The comparison relies on the dusty dominant AERONET stations described in Section 3.4. For each station the monthly time-series of total aerosol optical depth at 440 nm are compared with the climate model value at the grid cell where the the station is located. As we are considering dusty stations, the correlation of the time-series represents how well the seasonal cycle is captured or not, while the representation of the amplitude of the cycle is measured by the standard deviation. Therefore the ratio of standard deviations is an indication of the agreement in seasonal amplitude between the models and observations. Those
statistics are compared using the normalised Taylor diagram (Taylor, 2001). These diagrams are shown in Figure 12 for the PD and PDN simulations. The behaviour of each model with respect to the observations at a station is indicated by both its radial and angle values: the radial value indicates the normalised standard deviation with respect to observations, the angle measures the correlation between time-series.

A common result across all models comparing the PD and PDN experiments is the higher correlation for simulations with
nudged-winds, but similar normalised standard deviation for the cloud of points. With nudged winds the correlation is always positive except at one station for NorESM, a model that has a correlation larger than 0.6 for 13 stations in PDN (nine stations

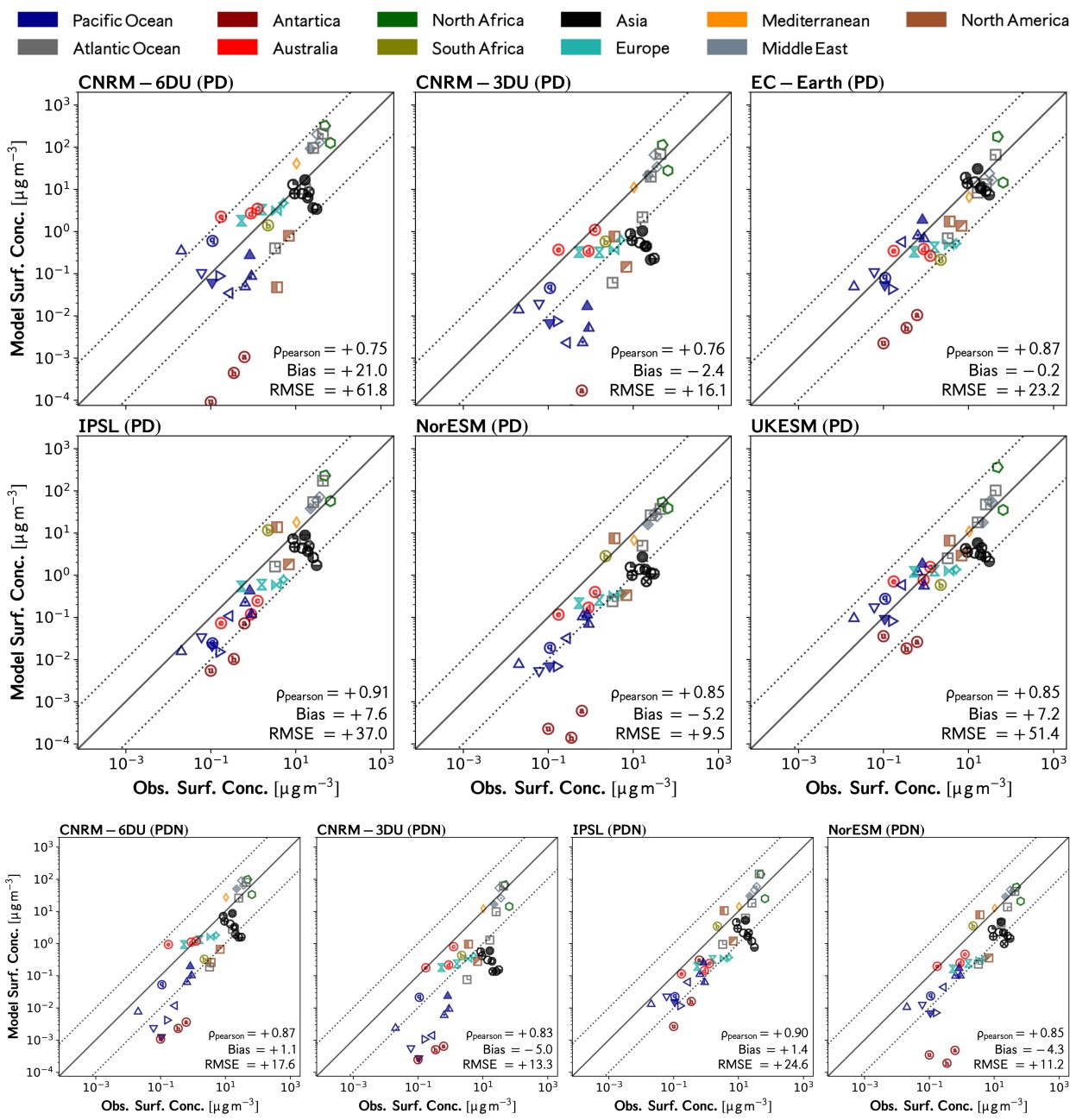

**Figure 13.** Comparison of dust surface concentrations in the models with the climatological dataset of Prospero and Nees (1986) and Prospero and Savoie (1989) for the PD and PDN experiments. The colors of the points indicate the region to which the measurement station belongs. Climatological datasets were obtained from observations over the period from 1991 to 1994. For the PI experiment see Figure S.SDC.10.

**Table 13.** Statistical properties of the comparision of the CRESCENDO-ESMs dust surface concentration with respect to the global network shown in Figure 2. Statistic metrics used in this table are described on Table 5. Pearson Correlation Coefficient ($\rho$), bias ($\delta$) [$\mu gm^{-3}$], normalised bias ($\delta_N$), Ratio standard deviations ($\Sigma$), Normalised mean absolute error ($\theta_N$) and Root mean square error (RMSE=$\eta$).

| Model | Exp. | Surface Concentration Network | | | | | |
|-------|------|--------|--------|------------|---------|------------|--------|
| | | $\rho$ | $\delta$ | $\delta_N$ | $\Sigma$ | $\theta_N$ | $\eta$ |
| CNRM-6DU | PD | +0.76 | +23.19 | +1.82 | +4.59 | +2.26 | +65.14 |
| CNRM-3DU | PD | +0.76 | -2.46 | -0.19 | +1.52 | +0.74 | +16.92 |
| EC-Earth | PD | +0.88 | -0.48 | -0.04 | +1.92 | +0.79 | +24.36 |
| IPSL | PD | +0.91 | +8.53 | +0.67 | +3.03 | +1.26 | +38.95 |
| NorESM | PD | +0.87 | -5.62 | -0.44 | +0.84 | +0.48 | +9.95 |
| UKESM | PD | +0.84 | +8.08 | +0.63 | +3.88 | +1.30 | +54.14 |
| CNRM-6DU | PDN | +0.87 | +1.33 | +0.10 | +1.70 | +0.86 | +18.59 |
| CNRM-3DU | PDN | +0.82 | -5.36 | -0.42 | +1.08 | +0.68 | +13.98 |
| IPSL | PDN | +0.89 | +1.69 | +0.13 | +2.15 | +0.98 | +25.91 |
| NorESM | PDN | +0.86 | -4.58 | -0.36 | +0.95 | +0.55 | +11.72 |

for PD). The PD experiment has only one case with correlation values around 0.8 (NorESM at Oujda), but all the models in PDN experiment have stations with correlations larger than 0.8 indicating that the seasonal cycle of optical depth is clearly improved with wind fields from reanalysis. The CNRM-6DU model has a strong change in the normalised standard deviation from PD (for which most of the stations have values larger than 1) to PDN (with most of the stations with values smaller than 1). In terms of the amplitude of the seasonal cycle, the most challenging stations for all models are in Australia (Birdsville station), Gobi Desert (Dalanzadgad and Sacol) and Izaña (close to Sahara but on an island and at high elevation). In terms of correlation Dushanbe in Thar region, and Sacol (China) are challenging. On the other hand stations like Sadaa (West-Sahara), Eilat (North-Middle-East) or Dakar are reasonably well captured by models.

## 5.6 Surface Concentrations

The stations were chosen to cover a range of dust values from low to moderate dust concentrations, mainly located at a distance from the main dust emission regions. According to the instrument location, Sahel and the West coast of North Africa (green and grey squared) together with Middle East stations (grey diamonds) report the highest values of surface concentrations, see Figure 13. The group represented by black circles represents moderate values indicating transport of dust from arid and semi-arid regions of East Asia. The lowest values correspond to Antarctica and the Pacific Ocean (blue triangles). The values of the dataset are shown in Table S.MD.4 of the Supplementary information.

The comparison between the CRESCENDO models and a network of stations that measure dust surface concentrations is shown in Figure 13 for the PD and PDN experiment and in Figure S.DSC.10 for the PI experiment. The agreement falls into the same range as previous comparisons with Community Atmospheric Model (CAM) (Albani et al., 2014) where the full range for the expected differences in annual mean values is close to 10. This range of differences between models compares well with the previous study from Huneeus et al. (2011).

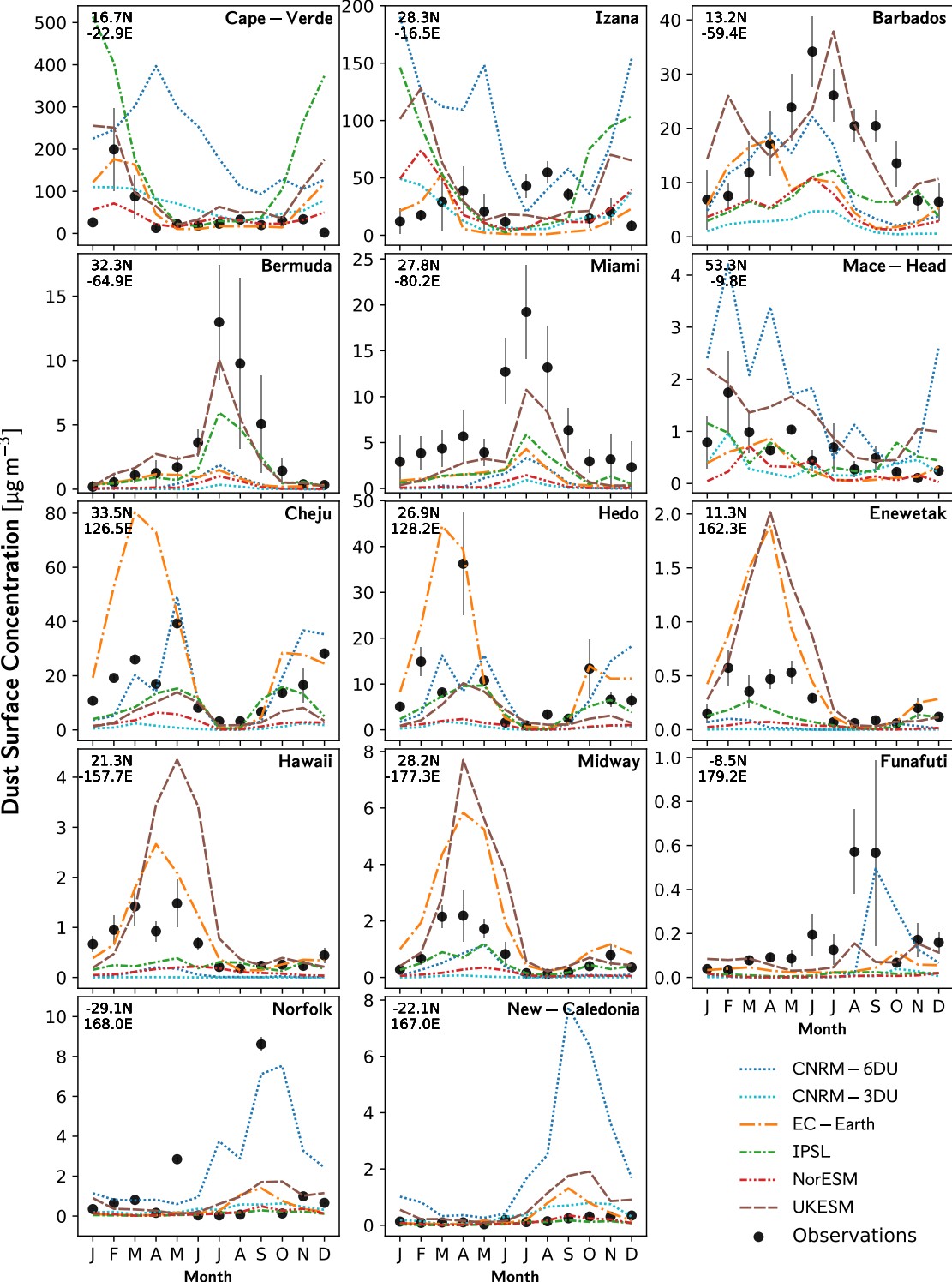

**Figure 14.** Comparison of ESM models (PD) of dust surface concentration with a station based climatological dataset. For PI and PDN experiment see the supplement figures S.DSC.7 and S.DSC.8.

CNRM-3DU underestimates dust concentrations over the Pacific Ocean. This behaviour over regions remote from dust sources could be partly due to the non-conservative semi-Lagrangian transport scheme that accentuates the differences with the distance of transport (a fact also consistent with their values of the Pearson correlation, mainly in nudged-simulations). All models except IPSL underestimate the concentrations in the Antarctica station. This could be due to the larger emissions from Patagonia that cause the increase in correlation coefficient for this model. Over Northern Europe all models, except CNRM, tend to underestimate dust concentrations and do not reproduce the range of variability found in the observations. When comparing PD and PDN simulations, IPSL and NorESM models show slightly better agreement in PDN conditions, whereas the two CNRM models show higher correlations when using nudged-winds but similar differences over the Pacific Ocean.

The correlation between the models and observations is significant for all models. The RMSE values are influenced by the stations with the highest concentrations and hence are more representative of the concentrations near the Sahara desert and the Middle East. In this regard, the NorESM and CNRM-3DU models show the best agreement over these regions. The EC-Earth model shows however the smallest bias because it better captures dust concentrations over Japan and East China, where the other models underestimate concentrations. Values of normalised bias and normalised mean absolute error complement the previous metrics and give us a characterisation of global differences accounting equally for the stations with the lowest concentrations (see Table 13). The normalised statistics indicate that the nudged-wind simulations generally show a better agreement with observations.

Although the 36 stations are covering many regions, a complete assessment of the model performance at the surface is not possible due to the absence of stations in South America and Asia, and only one station inland over North America and Africa. Therefore, the global observational constraints, in terms of the surface concentrations, are only partial.

The comparison of the seasonal cycle of surface concentrations against 14 stations is shown in Figure 14 for the PD experiment. The stations Cape-Verde and Barbados are in the same latitude at opposite sides of the Atlantic, therefore they have a signature of the transatlantic transport of mineral dust from the Sahel region. The IPSL, CNRM-6DU and UKESM models overestimate the early winter contributions to the seasonal cycle in Cape-Verde. The models reproduce the concentrations within a factor of 2 from May to September (except CNRM-6DU model) with, in general, an overestimation except for EC-Earth. However in the case of Barbados UKESM after April and CNRM-6DU before May reproduce very well the surface concentrations. All the other models, although with a similar seasonal cycle, underestimate the total surface concentrations by a factor from 2 to 4. The stations Izaña, Bermudas and Miami have also similar latitudes and represent the Atlantic transport from West-Sahara. Izaña Observatory is not at sea level and all the models have difficulties to reproduce the seasonal cycle. The seasonal cycles of Bermuda and Miami are well reproduced with a general underestimation of the surface concentrations values, where only UKESM and IPSL show a consistency within a factor of 2. Cheju and Hedo are stations on the Western Pacific Coast and their measurements represent the dust transport from China. The EC-Earth model reproduces well the seasonal cycle but with an overestimation of spring concentrations by a factor of 3. The seasonal cycle and values are well represented by the CNRM-6DU and IPSL models. Enewetak is located between Philippines and Hawaii in the middle of the Pacific Ocean, and EC-Earth and UKESM overestimate the spring concentrations whereas all the other models underestimate them. A similar

situation is found in Hawaii and Midway. The rest of the stations are in the Southern Hemisphere where the dust concentrations are smaller and the seasonal cycle is only partially reproduced. The results for the PDN experiment (see supplement DSC) are similar with a slight improvement in the seasonal cycle but with a general underestimation of surface concentrations. All the models with nudged winds exhibit problems in reproducing the observations in Izaña.

## 6   Discussion and conclusions

The analysis of the results provides insight into how the combination of modelling and measurements of dust can be used to improve our understanding of the dust cycle.

A first approach to the evaluation of the dust cycle relies on in the total dust loads and emissions. In this regard, we have shown that the model ensemble values of total emissions with nudged winds has less dispersion. We stress, however, that dust column loads are a better quantity when comparing models with different size distributions at emission than comparing total emission fluxes, since gravitational settling gets rid of the very large particles over a short time span. For dust loads, all models in PDN experiments are in a range between 9.1 and 15.2 Tg which can serve as a baseline to study model improvements. Because new studies support the important role of the coarse mode of dust (Huang et al., 2020), it is recommended to compare the contributions to dust load for fine and coarse modes separately. The range of dust loadings that we obtained is smaller than recent estimations (Kok et al., 2021) that propose values $\gtrsim 20$ Tg with a multi-model comparison with models with geometric diameters up to 20μm but based on a new methodology where the dust diagnostics are including observational constraints (Kok et al., 2020). Actually, Adebiyi and Kok (2020) propose that the total load of dust in the atmosphere is higher than what is estimated typically, and give a mean value close to 30 Tg, where the contribution of the coarse mode is more important than the fine mode.

Therefore, annual global dust emissions from climate models are dependent on the dust particle size distribution (DPSD) representation. The first result we observe is that those models that account for particles with diameters larger than 10 μm produce higher total fluxes. However, although an important diversity in the total emissions depends on the upper threshold, also the specific boundaries of the bin for largest particles used in a sectional scheme seems critical. We observed large differences in total emissions between UKESM1 and CNRM-6DU where an important difference is the lower-boundary of the last bin diameter: 20 μm and 10 μm, respectively. For this reason we have proposed two classifiers for further model analysis, but still we need a reasonable metric to compare the emissions at grid-cell scale.

To overcome the challenge of comparing models with different DPSD at emission, we introduced normalised emission maps, showing first (by a comparison between PD and PDN simulations) that wind fields do not substantially affect these normalised emission estimates in terms of spatial patterns when we analyse the 15 year emissions means of the PD and PDN simulations. This led us to interpret differences in regions where dust was emitted as reflecting differences in the underlying dust *effective* soil erodibility information (DESEI) among models. However, the DESEI is also including a sort of meteorological factors because the role of soil moisture in the emission process, together with specific properties of the dust scheme like the threshold in friction velocity or how the soil texture is translated into a dust size distribution. Note that the simulations compared in

our study share the same sea-surface temperatures which reduces the model diversity in terms of precipitation. Nonetheless, the consistency we report between PD and PDN normalised emission maps needs further investigation at smaller spatial and temporal scales, in particular at daily and sub-daily scales.

Beyond the interpretation of the re-gridded normalised emission maps, they allow us to compare the relative intensity contribution to dust emissions on the same spatial scale. It is a useful tool, as a direct balance of the several source functions is complex. For example, with the aim of reproducing dust observations at different model resolutions, models have introduced correction factors to their dust soil erodibility (see for example Albani et al. (2014) and Knippertz and Todd (2012)). In contrast, our normalised emissions can indicate effective model differences, both in intensity and location, on preferential dust sources. We found that these differences are the largest over Asia and are also significant over Australia. Hence, we identified these regions as two source regions that would benefit from further comparison of dust emission observations with actual model occurrences in emission fluxes. Moreover, the diversity in Asia emissions is investigated by Kok et al. (2021) obtaining also important differences with Aerocom Phase I models, and suggesting an underestimation of dust emission from East Asian deserts. Finally, additional research is also needed to ascertain seasonality disagreements in dust sources, which our 15 year mean normalised emission maps are not showing, but where seasonal normalised emission maps would be an useful tool.

Regarding dust deposition, another important point of discrepancy between models is the ratio between wet and dry deposition over similar particle size ranges, indicating that specific sensitivity studies should focus on the treatment of deposition. Interestingly, we have found that there is not a correlation between the modelling of largest particles and the value of this ratio. Finally, evidence of significant differences is also found in deposition over the oceans, in particular over the Indian Ocean and over the Pacific West, both of which are affected by dust source distributions over Asia.

To properly evaluate the impact of the dust in the climate system, it is important to determine an uncertainty range of the direct radiative effects for each model. Based on a calculation with 4 modes over a range from 0.1 to 100 μm, we observe that those models without the smallest particles (without mode $m_1$) will underestimate the short-wave contribution at the TOA by up to 20%. Models without the largest particles (those represented by the $m_{22}$, i.e. for bins with diameter larger than 40μm) are expected however to not be significantly affected in their estimations of DRE in the SW. Nevertheless, we need additional studies to conclude whether these estimates are consistent with other models with the same range of modelled dust size particles. In particular, it is recommended to attribute diversity in the context of the several refractive indices.

The dust optical depth is a key diagnostic in comparative studies. It appears to be logical to try to constrain the dust cycle by relying on dust optical depth (DOD) estimated from satellite observations. This is because the dust emissions depend on mineralogy, on land surface properties and on regional meteorology. Therefore a few in-situ measurements are not sufficient to constrain the dust cycle at any possible scale. Indeed, Ridley et al. (2016) used retrievals from instruments on-board MODIS and MISR to estimate global values for DOD between 0.020 and 0.035 which place two models (CNRM-3DU and UKESM) outside this observational range. Note, however, that there are difficulties to estimate DOD from satellite retrievals with the method of (Ridley et al., 2016) because it still relies on model simulations to ascertain the fraction of non-dust optical depth. As shown by our results in the supplement material (Section DOD), the non-dust fraction of optical depth can have large inter-model differences. Furthermore, an important result is that, although DOD should be proportional to the mineral dust

total column, models with the lowest dust loadings are not those with the smallest DOD. This is illustrated in the differences in

mass extinction efficiency (MEE) between the different models. The magnitude of MEE is a good indicator of intrinsic model

properties due to its relatively small seasonal cycle, an aspect in which all the CRESCENDO-ESMs match. But also, because

mass extinction efficiency is affected by the DPSD and optical properties of mineral dust modelled, it is also a useful property

to compare with observations.

Our analysis of dust optical depth include a study at regional scale. Specifically, the regional dust optical depth over dust

source regions relies on a comparison with MODIS satellite estimates of DOD based on the algorithm described in (Pu and

Ginoux, 2018b). This comparison allowed us to evaluate the skill of each model by evaluating the correlation between the

regional time series of observations versus each model. A significant increase in the skill was revealed for the simulations

using nudged winds, indicating that a consistent reproduction of the seasonal cycle depends critically on how the strong

surface winds are represented (with a improvement with the use of re-analysis wind data sets). However, the correlation (skill)

is not useful in determining differences in the scale of the signal, and Figure S.DOD.3 shows that there are regions where

the seasonal cycle is well reproduced but the mean annual signal is actually underestimated, see also Pu and Ginoux (2018b).

A further example of the difficulties in specific regions is given in the newly incorporated stations over Asia compared with

Huneeus et al. (2011), because these stations has been proven to be challenging for the CRESCENDO-ESMs in terms of the

820 comparison provided by Taylor diagrams (see figure 12).

## 7 Future research directions

Currently, the dust source disagreements/differences between models make it difficult to quantify the fraction of the uncertain-

ties of dust emission due to those small-scale atmospheric phenomena that are not well represented by global models. The use

of wind fields from reanalysis data sets reduces the differences between models, but a benchmark reference dataset regarding

dust sources is needed to establish a range for those uncertainties. In particular, specific model comparisons based on a com-

mon soil erodibility information would illuminate on specific model improvements to decrease diversity. Indeed, these studies

should use a similar prescribed seasonal vegetation fraction and bare soil distribution to improve the seasonal consistency.

The dust particle size distribution is a key point of research for current ESM. Specifically, the global description of the

dust cycle in terms of the amount of aerosol mass mobilised needs to be extended to larger particles as they can significantly

increase the total emissions. At the same time according to recent studies the fraction of dust mass in the atmosphere due to

the coarser particles would be dominant with respect to fine mode (Adebiyi and Kok, 2020). A further complication, we found

in our analysis is that the method by which the largest particles are incorporated in the models can drive strong differences

in total emissions with ranges from $3500\ \mathrm{Tg\,yr^{-1}}$ in CNRM-6DU to about $7000\ \mathrm{Tg\,yr^{-1}}$ in UKESM model. In particular,

the specific bins used to model the contribution of largest particles are critical to understand model diversity. Additionally, a

835 better discrimination of particles larger than 10μm but smaller than about 20 to 30 μm will conclude if the results in the Table

7 are consistent between different models. This also illustrates that comparisons where the particle size distribution is resolved

(comparisons based on the contributions of each DPSD bin) is needed to understand better the source of model discrepancies,

in that regard CRESCENDO-ESMs simulations were designed with these future evaluations in mind. Also we created specific tools to estimate binned contributions from models based on modal DPSD (Checa-Garcia, 2020a) to support these comparisons.

However, these differences in total emissions are not directly translated into proportional loadings because of the differences in deposition between models, and therefore in the lifetime. In particular, regarding total deposition one priority should be given to analysing the large differences in the ratio between dry and wet deposition between models which is only partially explained by the modelled size distribution. From the aerosol micro-physical point of view differences in the dominance of wet scavenging over ocean regions could account for part of these differences. However, as indicated by Shao et al. (2011)

observations of dry deposition velocities in wind tunnels are not reproduced by current dry deposition schemes. In this scenario it becomes necessary to compare with measurements of wet and dry deposition separately (Marticorena et al., 2017). In fact, although our ensemble mean global contributions of gravitational settling, wet deposition and dry deposition without sedimentation are similar, there is a large model diversity. To explain better the model diversity in sedimentation a first step is to ensure that gravitational settling is estimated for all atmospheric levels before a comparison of sedimentation for each size

range. Because, wet deposition involve the modelling dust-cloud and dust-rainfall interactions the model diversity is partially conditioned by other parts of climate models (Croft et al., 2010). However, sensitivity studies for each model based on the plausible range of values of their dust scavenging coefficients (in-cloud and below-cloud) can provide valuable information on the actual range of uncertainties expected for each model.

    The models exhibit important differences in preferential dust sources, in particular a better agreement of preferential sources

found over Asia and Australia would give us more consistency in global dust transport over the Indian and the Pacific Oceans. Although there is a scarcity of measurement campaigns over Asia compared to the Sahara and Sahel, studies based on empirical relationships between visibility and dust surface concentrations give us an additional insight into dust sources over these regions (Shao and Dong, 2006). This information, supported by new regional studies is needed to suggest best lines of model improvements in these regions.

Given that the optical depth depends on column load rather than dust emission fluxes, the inter-model convergence can be reasonably achieved even for those models that are not implementing particles with radius larger than 10µm. Also, an inter-model convergence in terms of optical depth is important to better constrain the dust radiative forcings and direct radiative effects (DRE). However, as said earlier, the link between dust loads and dust optical depth, i.e. the MEE, shows important model differences. Additional MEE observations to better constrain the expected values would definitively help modellers to

improve the dust load description by comparing with satellite dust optical depth estimates. Finally, given the different role of each mode: fine, coarse, super-coarse and giant in the dust-radiation interaction, further studies, not only on the mineral composition but also in possible dependence of the composition with size of dust particles would improve our estimates of dust radiative forcings and direct radiative effects.

## Appendix A:  Method to estimate Direct Radiative Effects in multi-modal size distributions

In section 5.1 the direct radiative effects for a dust scheme with several dust modes were shown. Here we present the methods used to obtain the results of Table 7. The *direct radiative effect* of a species is defined by the *earth's instantaneous imbalance* at the top of the atmosphere due to that specific atmospheric species/component. It has been introduced at Boucher and Tanré (2000) and discussed by Bellouin et al. (2013) and Heald et al. (2014). This imbalance is conceptually different from the radiative forcing (either defined as a stratospherically adjusted instantaneous radiative forcing or by an effective radiative forcing) which is a comparison between two different time periods, usually between pre-industrial time and present day. In our case the estimations of direct radiative effects are estimated during a single simulation with present day conditions but with multiple calls to the radiative transfer model implemented in the climate model. The aerosols in the climate model have actually direct, indirect and semi-direct effects in the simulation but the method only estimated the direct radiative effects due to scattering and absorption of specific aerosol species. Therefore there are observational based estimations of the direct radiative effects of the aerosols (Yu et al., 2006). However, from the point of view of aerosol modelling based on multi-modal approaches, differences have been reported (Di Biagio et al., 2020) between (a) the calculation by the sum of each mode contribution estimated individually, and (b) the estimation for the joint multi-modal directly.

In this appendix two different approaches and a joint new method with four calls to the radiative scheme are described to decrease these differences.

In general, in the calculation done by current radiative transfer schemes it is considered a state of the atmosphere with several aerosols species $\mathcal{X}, \mathcal{Y}, \ldots$ where each species is possibly described by a multi-modal distribution with modes $X_1, \ldots, X_n$. The state with all the aerosol species is named hereafter $\mathcal{A}$, therefore $\mathcal{A} = \mathcal{X} \cup \mathcal{Y} \cup \mathcal{Z} \cup \ldots$. We define another state named $\widetilde{\mathcal{A}}$ that includes all the modes of every aerosol specie except those modes corresponding to the species $\mathcal{X}$. Therefore, $\mathcal{A} = \widetilde{\mathcal{A}} \cup \mathcal{X}$. The radiative effect of the aerosol $\mathcal{X}$ described by several modes $X_1, ..., X_n$, is defined by,

$$\widehat{\mathcal{F}_X} = \mathcal{R}(\mathcal{A}, \delta) - \mathcal{R}(\widetilde{\mathcal{A}}, \delta)$$

where $\mathcal{R}$ represents the radiance obtained in our radiative transfer scheme which is intrinsically a non-linear forward model. $\delta$ represents all others elements considered by our radiative scheme beyond the aerosol species which are invariant for both estimations of the radiance.

However, in order to disentangle the contribution of each mode $X_i$ of the specie $\mathcal{X}$, results differ depending on the methodology used due to the non-linearity of $\mathcal{R}$. We define here two methods: the first approach considers each $X_i$ mode added individually to $\widetilde{\mathcal{A}}$ with respect to the experiment given by $\widetilde{\mathcal{A}}$, hereafter we name this as *method in*. The second approach compares an experiment $\mathcal{A}$ with a scenario $\widetilde{\widetilde{\mathcal{A}}}$ where all the modes $X_j$ with $j \neq i$ are included, named hereafter *method out*. Visually, the *method in* would compare a base state without any mode of the target component with a state where the specific mode is added (therefore, *in*). The *method out* compares a state with all the modes of a target component with a state in which the specific mode is removed (therefore named *out*).

The *method in* is written for the radiative effects of $X_i$ as,

$$\widehat{\mathcal{F}_{X_i}} = \mathcal{R}(\widetilde{\mathcal{A}} \cup X_i, \delta) - \mathcal{R}(\widetilde{\mathcal{A}}, \delta)$$

whereas the *method out* is written as,

$$\mathcal{F}_{X_i} = \mathcal{R}(\mathcal{A}, \delta) - \mathcal{R}(\mathcal{A} \cup X_i^*, \delta) \quad \text{with} \quad X_i^* = \cup_{i \neq j} X_j$$

and we note that $\mathcal{F}_X = \widehat{\mathcal{F}_X}$ but $\mathcal{F}_{X_i} \neq \widehat{\mathcal{F}_{X_i}}$. In particular, we have both, $\sum_i \mathcal{F}_{X_i} \neq \mathcal{F}_X$ and $\sum_i \widehat{\mathcal{F}_{X_i}} \neq \widehat{\mathcal{F}_X}$.

However, the results for 4 modes of mineral dust of IPSL-4DU, shown at Table 7, indicate that $\frac{1}{2} \sum_i (\widehat{\mathcal{F}_{X_i}} + \mathcal{F}_{X_i}) \approx \widehat{\mathcal{F}_X} = \mathcal{F}_X$.

Therefore the joint method described based on four calls to the radiative transfer scheme to calculate the direct radiative effect is providing estimates per mode that combine linearly to reproduce the multi-modal direct radiative effect.

*Code availability.* The core functions of the software used for data-analysis are available on the reference (Checa-Garcia, 2020a) and its related open-source code repository. The open-source code used to prepare and test IPSL diagnostics is (Checa-Garcia, 2020b).

*Author contributions.* RC-G and YB designed the research. RC-G analysed the data and wrote the manuscript with input from YB, SA, PN, DO, FMO'C and TvN. Data from climate model simulations were provided by: TB, PLS and TvN for EC-Earth, MS and DO for NorESM, FMO'C and CD for the UKESM, MM and PN for the CNRM, RC-G for IPSL and IPSL-4DU. SA, YB and AC developed the IPSL-4DU dust scheme. BM and JMP provided observational data sets used in the analysis.

*Competing interests.* The authors declare that they have no conflict of interest.

*Acknowledgements.* This work has been supported by the European Union's Horizon 2020 research and innovation programme under grant agreement No 641816 (CRESCENDO). SA acknowledges funding from the European Union's Horizon 2020 research and innovation program under the Marie Sklodowska-Curie grant agreement 708119, for the project DUS3C. The French National Observatory Service IN-DAAF is supported by the INSU/CNRS, the IRD (Institut de Recherche pour le Développement) and the Observatoires des Sciences de l'Univers EFLUVE and Observatoire Midi-Pyrénées. The authors would like to thank the French and African PIs and operators for maintaining the stations and providing the PM10 concentrations. The INDAAF data are distributed on the web site : https://indaaf.obs-mip.fr/. The authors would like to acknowledge the contribution of Mohit Dalvi, Jane Mulcahy and Stephanie Woodward from the UK Met Office Hadley Centre in developing and/or running the UKESM1 simulations. RC-G and YB gratefully acknowledge the hospitality of the Institut Pascal during the Paris-Saclay Indices Program 2019, supported by ANR-11-IDEX-0003-01. We also thank comments of the three anonymous reviewers.

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
