# Peer review of "Evaluation of natural aerosols in CRESCENDO-ESMs: Mineral Dust"

_Atmospheric Chemistry and Physics, 2020_

## Referee Comment (RC1) · Anonymous Referee #1 · 18 Dec 2020

In this study, the authors compare a small multi-model ensemble of mineral dust simulations to observations of mineral dust deposition, surface concentrations, and optical depth. Models perform in diverse ways against the different metrics. The comparison is complicated by the different coverage of mineral dust size distribution by the different models. This a typical issue in mineral dust inter-comparison papers, which the authors try to work around to an extent but cannot really avoid.

The paper gives an avalanche of figures and numbers and comes with a chunky supplementary document. That is not an issue in itself, but the discussion and conclusion sections should make more of an effort to summarise and add value to the analysis. In my mind, the questions that the discussion should clearly answer are:

[Figure]

- Do non-dust differences dominate model disagreement? It seems to me that non-dust factors dominate – at least for emitted mass and load. The authors downplay the contribution of different wind fields, at least in a normalised sense, but highlight the contribution of the "effective" soil erodibility.

- How can model disagreement be resolved? The paper tries to explain some of the model differences, but the Crescendo simulations are not sufficient to go beyond speculations. But the results indicate where future inter-comparisons must do more: a simulation with prescribed soil properties, including moisture, is clearly required. Mass diagnostics integrated over the size distribution are clearly a barrier to understanding, so must be replaced with diagnostics for the different size modes/bin of each model, which can be remapped to common bins for comparison purposes.

- What observations can support further progress? The paper uses existing observations very well. I was struck by the absence of aircraft data, which seems to imply that all those expensive aircraft campaigns dedicated to mineral dust do not measure the quantities that are needed to improve models. The authors have the opportunity to say what those quantities are: size distribution, clearly – with the need to go beyond case studies and constrain the climatology. Mass extinction efficiency looks important too. Something else?

**1 Other comments**

- Page 1, line 8: "uncertainty": "diversity" would be preferable because it is unlikely that 5 models sample the full uncertainty range.

- Page 1, line 9: how many models in that subset?

- Page 1, line 10: "better consistency between models": all models, or the subset?

- Page 1, lines 14-17: The abstract needs to say what the conclusions of these two tasks were.

- Page 2, line 23: Could say that the estimate by Kok et al. (2017) comes from observations and models

- Page 2, line 29: Could note that the impact of mineral dust on the phosphorus budget of the Amazon may be smaller that previously thought, based on Prospero et al. 2020 https://doi.org/10.1029/2020GB006536

- Page 4, lines 21-22: What are the differences reported by Yu et al. 2019 due to?

- Page 6, line 21: "non-mixed": it is more usual to say "externally mixed"

- Page 6, line 23: what experiments?

- Page 7, lines 24-26: That seems to be an example of the processes mentioned in line 20, so could be moved there.

- Page 7, line 27: The information in Section 2.1 would be better described by a table of experiments.

- Page 11, Table 3: the units of MEE are given as $m^2$ $g^{-1}$ in Table 5. It would be good to harmonise that.

- Page 12, line 4: What is meant by "along the seasonal cycle"?

- Page 14, lines 29-32: The low regard given by the author to Pearson correlation is surprising since that measure is used extensively throughout the paper. I suggest toning down that statement or clarifying that it only applies to specific comparisons.

- Page 19, line 27: "being the only model" – is that CNRM-6DU?

- Page 23, lines 5-6: But does CNRM-6DU match the Adebiyi and Kok (2020) estimates for the right reasons? Adebiyi and Kok (2020) estimate the burden of coarse-mode (larger than 5 microns) dust to be 17 Tg. Does the model also match that number?

- Page 28, line 25: Is that so remarkable? The models must prescribe fairly similar soil properties.

- Page 30, lines 11-13: Did the CNRM model do something specific to represent Hoggar emissions?

- Page 45, lines 25-30: What about the LW? It would probably be the other way around, so there should be cancellation of error in size distribution between the two spectra.

- Page 44 line 31: What is the difference in terms of content between section 6 Discussion and section 7 Conclusion? They seem to both be a mix of summary and further discussion, so could be merged.

- Page 45 line 6-8: Where has the discussion on effective erodibility taken place? It is the first time the paper mentions that concept.

**2  Technical comments**

- Page 3, caption of Table 1: extra word "of about"

- Page 3, line 20: "indicates" -> "indicate"

- Page 42, line 10: "correspond at" -> "correspond to"

- Page 44, lines 14-15: What is meant in the part starting with "although with"?

- Page 47, line 48: typo: "an scarcity"

- Page 47, line 12: Rephrase "which resulted to be challenging"

- Page 50: Grammar of the last sentence of the acknowledgment could be improved.

---

## Referee Comment (RC2) · Anonymous Referee #2 · 28 Dec 2020

In this manuscript, the authors examined the simulated dust cycle by five Earth System Models. They compared the simulated the dust dry and wet depositions, dust surface concentrations, and dust optical depths against measurements across the world at both annual and seasonal scales. Their results confirmed what's known to the dust research community, including that (i) the cutoff maximum dust size is important to the dust emission magnitude, and (ii) the ratio of dry to wet depositions is highly divergent between models. Furthermore, the authors found what's less known to the dust research community, including that (i) using identical nudged winds among models can improve the consistency between models in the dust cycle, and (ii) the divergent mass extinction efficiency between models explains why similar dust loads result in a large difference in optical depth.

I agree with Anonymous Referee #1 that the presentation quality needs to be improved. The authors offered extensive and interesting results (in both the main text and the supplement). However, clear and compact leading and ending sentences per paragraph are missing. This could give readers an excuse to stop reading, and thus decrease the impact of the manuscript.

The manuscript is overall excellent science. I do have some comments that the authors should address before publication.

- Throughout the manuscript, the West Pacific and the East Pacific are defined problematically. To me, the West Pacific is the side where East Asia, Southeast Asia, and Australia are, and the East Pacific is the side where North and South Americas are located. However, the authors treated them reversely (see Fig. 1a as an example). This mismatch concerns me a lot, and can cause unnecessary misunderstandings in future studies. I suggest the authors correct the two regions throughout the manuscript systematically.

- The cutoff maximum dust diameters of the seven models need to be better presented. First, the maximum diameters of the 4 modal models (in Table S. MD. 9) are missing. Second, the maximum sizes of the 3 sectional models (Table S.MD.8) do not match Page 19 lines 10-15. For example, in Table S. MD.8, the maximum diameter of CNRM-6DU is 100 $\mu$m, however, in Page 19 line 12, the maximum diameter is 50 $\mu$m. A similar issue exists in CNRM-3DU. Since maximum diameters are critical to this manuscript, I suggest the authors address these two issues, and make the maximum diameters very clear in Section 2.

- In Page 35 lines 8-10, the authors compared simulated deposition flux in Asia, and implicitly indicating that EC-Earth is better than all the other models. However, the tricky thing is that there is only one station in the Asian region (as seen in Fig. 1a). (Similarly, there is only one station in South Atlantic, South America, and Egypt). The sample number is too small to draw a significant conclusion for a continent. Thus, I

suggest the authors make it clear that the sample number is one, and add the numbers of observational stations at all the regions in the legends of Figs. 8, 9, and 13.

- Recent progress in dust shape and its impact on dry deposition needs to be added in Page 7 lines 16-26. Jasper Kok's group has a recent paper (Huang et al., 2020) that compiled 27 measurements of realistic dust shape worldwide (including Li and Osada, 2007). They find that dust asphericity increases gravitational settling lifetime by 20% at all sizes.

- Recent progress in dust cycle needs to be added. A recent paper (Kok et al., 2020) that diagnosed the dust cycle is very similar to this manuscript, but used different models.

Minor comments:

- Typo in Page 6 line 33: correct "clay/silk" to "clay/silt"

- Typo in Table 3: the unit of grid cell area should be m^2 instead of kg

- Typo in Page 12 line 7: correct "19 stations" to "18 stations"

- Table 7 only offers emissions of 14 of the 16 regions. Regions "Mid-Atlantic" and "Sahel/Gulf of Guinea" are missing and should be added. Note that after adding the two regions, authors need to update the order of the top 10 regions with dust emission intensity.

- Typos exist in Table 7's order of the top 10 regions. For instance, for EC-Earth, there are two 4th largest sources (i.e., the North Sahara and the Taklamakan), which are clearly problematic. Typos also exist in models CNRM-3DU and IPSL.

References:

Huang, Y., Kok, J. F., Kandler, K., Lindqvist, H., Nousiainen, T., Sakai, T., Adebiyi, A. and Jokinen, O.: Climate models and remote sensing retrievals neglect substantial desert dust asphericity, Geophys. Res. Lett., 47(6), 1–11,

doi:10.1029/2019GL086592, 2020.

Kok, J. F., Adebiyi, A. A., Albani, S., Balkanski, Y., Checa-Garcia, R., Chin, M., Colarco, P. R., Hamilton, D. S., Huang, Y., Ito, A., Klose, M., Leung, D. M., Li, L., Mahowald, N. M., Miller, R. L., Obiso, V., García-pando, C. P., Rocha-Lima, A., Wan, J. S. and Whicker, C. A.: Improved representation of the global dust cycle using observational constraints on dust properties and abundance, Atmos. Chem. Phys. Discuss., (November), 1–45, doi:10.5194/acp-2020-1131, 2020.

Li, J. and Osada, K.: Preferential settling of elongated mineral dust particles in the atmosphere, Geophys. Res. Lett., 34(17), L17807, doi:10.1029/2007GL030262, 2007.

Li, J. and Osada, K.: Water-Insoluble Particles in Spring Snow at Mt. Tateyama, Japan: Characteristics of the Shape Factors and Size Distribution in Relation with Their Origin and Transportation, J. Meteorol. Soc. Japan. Ser. II, 85(2), 137–149, doi:10.2151/jmsj.85.137, 2007.

---

## Referee Comment (RC3) · Anonymous Referee #3 · 30 Dec 2020

General comments:

This manuscript presents the results of five Earth System Models simulations of the global dust cycle, emissions, dry and wet deposition, optical depths, and surface concentrations comparatively to satellites and in situ observations. The authors explore global and regional variability between models in three different simulated experiments: PD (calculated winds), PDN (reanalyzed winds), and PI (prescribed chemistry and aerosols). Overall, the manuscript is well written and provides ample content. Having said that, this manuscript is quite extensive and important information is left for the reader to find in the supplement. The content of this manuscript could be divided in two different publications. In the first one, you could explore the differences between the five models, and then, in the second, you could explore more deeply the differences

between the three simulated experiment scenarios. At least, I would include a figure of the particle size distributions used in each model into the main text.

Specific Comments:

Page 6, Line 7-31: For this part of the text, it would be very useful to have a plot overlapping the particle size distributions in each of the 5 models. This would be similar to what you have in the bottom panel of Figure 4.

Page 7, Line 11: "Therefore those optical properties are representative for the global mineralogical composition rather than a description of the soil-type dependence of the mineralogy that would imply local differences on emitted optical properties." The point you are raisin here is important, but it is still not clear what optical properties you have actually used. For instance, do all five models use exactly the same spectral complex refractive indices? Which databases/references are you using in each model?

Page 3, Table 1: It would useful for the reader to include the specific particle sizes ranges used in each model simulation in column DPSD. Alternatively, you could move Tables S.MD.8 and S.MD.9 from the supplement to the main text. Please, also include the meaning of PD, PDN, and PI in the title of Table 1.

Page 8, Line 2: "A last simulation where aerosols and chemistry emissions are pre-scribed for 1850 (named PI)". Why? Could you add a few words explaining why such simulation is relevant and how do you use its results specifically in this study, covering the years between 2000 and 2014?

Page 12, Line 3: Could you please clarify the criterion used to select optical depths? What does "all – aer" mean here?

Page 15, Figure 4: What do you mean by "samples are the marks on x-axis"? Does the "sample" correspond to a given year? Which are the years you consider here? Include the time-period in the figure caption.

Page 22, Line 21: Could you clarify if the model simulations were sampled at MODIS

and MISR times in Figure 4? What is the main reason for having UKESM's AOD so high?

Page 22, Line 29: Do you know why the EC-Earth and the NorESM have MEE values that differ from the other models? If so, I suggest you to discuss the main reasons in the text.

Page 23, Line 7 to 24: The plot overlapping all size distributions would again be helpful here.

Page 28, Line 7: "On the side the modeled wind surface friction velocity and speed agree better with actual meteorological conditions, and on the other side the description of the soil surface properties has become more accurate." These are important points. Could you provide references of experimental studies that support these two statements?

Page 30, Line 26: "... they indicate that although there are important differences between PD and PDN experiments in terms of total emissions, ...". It is difficult to see those differences here. Although it might be out of the scope of this paper, I think it would be interesting to comment on how the re-analyzed wind fields in PDN differ from the calculated wind fields in PD near dusty regions. Are there significant differences between the wind fields in PD and PDN? How much would be that difference?

Page 35, Line 22: "All the other models underestimate total depositions fluxes over stations where fluxes exceed 100gm-2yr-1." What do you think is the main reason for that?

Page 44, Line 7 to 30: Could you comment on the temporal resolution of the surface concentration observations? Are those monthly means based on continuous daily observations? Are these observations for one or the average value for multiple years? Include the years of the observations and the simulations in the figure caption.

Technical Corrections:

Page 8, Figure 1: The aspect ratio of the Figures 1a and 1b seems strange.

Page 14, Line 27: Replace "has" by "have".

Page 16, Line 25: Replace "in the main paper" by "this paper".

Page 17, Table 6: Explain the meaning of "in and out".

Page 33, Line 11: "Figures Dep.11 and Dep.12 show . . .". I cannot find these figures.

Page 44, Line 25: "sitation"
* * *

---

## Author Comment (AC1) · 14 Mar 2021

We thank the referee for the comments and questions. They help us to improve our manuscript and to clarify several points. Here we are indicating our answers in boxed frames after each point raised by the reviewer and our changes/actions in the manuscript within a green colour box. Informative data are given in orange boxes.

**INFORMATION**: The Table 6 has been double-checked by the different modelling groups. CNRM reported that, instead of our previous estimate, their diagnostics of dry deposition are not including sedimentation which means different values of total and dry deposition (without sedimentation). With this revision the CNRM-6DU model has a larger bias due to an unclosed budget but the CNRM-3DU decreases the previous bias by a factor 2. In this situation we have removed the model CNRM-6DU from the multi-model mean, but we kept the CNRM-3DU. Given scale of the differences between models and observations, the comparison of total deposition draws the same conclusions and the results are very similar. Because the dust emission scheme is not affected by the bias, we kept their results in the analysis. All the Tables and Figures has been revised, and several of them improved according to the new information.

**1 General comments**

In this study, the authors compare a small multi-model ensemble of mineral dust simulations to observations of mineral dust deposition, surface concentrations, and optical depth. Models perform in diverse ways against the different metrics. The comparison is complicated by the different coverage of mineral dust size distribution by the different models. This is a typical issue in mineral dust inter-comparison papers, which the authors try to work around to an extent but cannot really avoid. The paper gives an avalanche of figures and numbers and comes with a chunky supplementary document. That is not an issue in itself, but the discussion and conclusion sections should make more of an effort to summarise and add value to the analysis.
Regarding the large amount of figures/tables this is partially a consequence of the CRESCENDO approach. In this project for each model, we have a set of several simulations to be analysed and compared. This means that the number of analysis/results (therefore Figures and Tables) are multiplied by a factor 3 with respect to other comparisons, because of the three experiments: PD (present-day), PDN (present-day-nudged) and PI (pre-industrial) described in the main paper.

We have reorganized the discussion and conclusion sections providing additional information. It has been added a new final section with future research.

In my mind, the questions that the discussion should clearly answer are:

(1) Do non-dust differences dominate model disagreement? It seems to me that non-dust factors dominate – at least for emitted mass and load. The authors downplay the contribution of different wind fields, at least in a normalised sense, but highlight the contribution of the "effective" soil erodibility.

We did not try to specifically downplay the role of non "dust scheme" differences. In the introduction we explain the several processes leading to dust activation events. The data analysis and discussion of emission maps has followed a step by step analysis:

1. Prepare for each model, emission maps for both simulations: the one with nudged winds (PDN) and the other one with non-nudged-winds (PD).

2. Compare non-normalised emission maps PD & PDN, named here $E$ and $\mathcal{E}$ respectively.

3. Compare the normalised emission maps PD & PDN, named here $\epsilon$ and $\varepsilon$ respectively.

We observe that for each model the differences between **normalised** PD and PDN are small (Figure 7), with important differences between models. Let's write the PD normalised emission map of model m with their expected dependencies: $\epsilon_m(\mathbf{r}; \mathbf{v_m}, \phi_m)$ and for the PDN experiments $\varepsilon_m(\mathbf{r}; \mathbf{v_{ERA}}, \phi_m)$. With the definitions: $\mathbf{r}$ is the location at surface, $\phi_m$ represents the dust scheme parameters including soil information (for model m), $\mathbf{v_m}$ are the wind velocities for each model m, and $\mathbf{v_{ERA}}$ the ERA-Interim nudged-winds. Our results indicate that:

$$\mathsf{E}_m(\mathbf{r}; \mathbf{v_m}, \phi_m) \neq \mathcal{E}_m(\mathbf{r}; \mathbf{v_{ERA}}, \phi_m) \quad \forall \mathbf{r}, m$$

but

$$\epsilon_m(\mathbf{r}; \mathbf{v_m}, \phi_m) \simeq \varepsilon_m(\mathbf{r}; \mathbf{v_{ERA}}, \phi_m) \quad \forall \mathbf{r}, m$$

For one specific model it is possible that $\mathbf{v_m} \simeq \mathbf{v_{ERA}}$ and we can not derive a conclusion of the functional arguments from the last relation. But given that non-normalised maps are different for all models m ($E_m \neq \mathcal{E}_m$), then we consider that it is reasonable to suppose that $\mathbf{v_m} \neq \mathbf{v_{ERA}}$ are different enough, and the functional dependence in the normalised emissions maps of wind fields is less relevant than $\phi_m$. In this context we have the interpretation that the comparison $\epsilon_m$ for each m is a comparison of a dust *effective* soil erodibility information (DESEI). We remark the two points implicit in our study:

A. The comparison of maps, and therefore the interpretations, are **15 years mean**.

B. Experiments PD and PDN have prescribed and identical sea-surface temperatures.

In the context of (B) a reduced diversity between the models is expected compared to fully coupled models. Because of (A) all our discussion is for our estimate of climatology emission maps. Note that we understand that the wind speed thresholds in the dust scheme are part of the DESEI, and the DESEI may also depend on the dust particle size distribution imposed during the emission process. Finally, the DESEI has still a sort of "meteorology" as far as it includes information of, for example, soil moisture.

We have clarified better the process we followed, and we have added some emphasis on the fact that each normalized emission map is calculated monthly and we presented the 15 years mean. Future research will analyse possible seasonal discrepancies.

**In the discussion section we modified our previous text by:**

To overcome the challenge of comparing models with different DPSD at emission, we introduced normalised emission maps, showing first (by a comparison between PD and PDN simulations) that wind fields do not substantially affect these normalised emission estimates in terms of spatial patterns when we analyse the 15 year emissions means of the PD and PDN simulations. This led us to interpret differences in regions where dust was emitted as reflecting differences in the underlying dust *effective* soil erodibility information (DESEI) among models. However, the DESEI is also including a sort of meteorological factors because the role of soil moisture in the emission process, together with specific properties of the dust scheme like the threshold in friction velocity or how the soil texture is translated into a dust size distribution. Note that the simulations compared in our study share the same sea-surface temperatures which reduces the model diversity in terms of precipitation. Nonetheless, the consistency we report between PD and PDN normalised emission maps needs further investigation at smaller spatial and temporal scales, in particular at daily and sub-daily scales.

How can model disagreement be resolved? The paper tries to explain some model differences, but the CRESCENDO simulations are not sufficient to go beyond speculations. But the results indicate where future inter-comparisons must do more: a simulation with prescribed soil properties, including moisture, is clearly required. Mass diagnostics integrated over the size distribution are clearly a barrier to understanding, so must be replaced with diagnostics for the different size modes/bin of each model, which can be remapped to common bins for comparison purposes.

We agree that the set of models used and the number of simulations (our sample size) although reasonably comprehensive of different dust scheme approaches, still is not exhaustive. We tried, when possible, to translate our study into numerical assessments, but we are cautious about directly extend our conclusions (or "ensemble values") to models/comparisons outside our analysis.
Once this premise is clear, still our analysis highlights several paths to overcome the model diversity:

- First we have proposed a simulation with prescribed soil properties by using a *benchmark reference dataset* regarding soil information, but we will clarify better this point in the discussion.

- In the CRESCENDO design of the simulations a key point has been to provide some diagnostics per-bin or per-mode. This first study includes already a large amount of analyses and comparisons. This means that a comparison per size range (and details about vertical distribution) will be incorporated into future publications. But we totally agree about the key role of analyses per bin/modes. As part of the set of software tools that we have created for this and future studies, we have included a set of methods to translate modal distribution variables into diagnostics over specific bins to perform that set of comparisons (see the following link to read the online manual with examples: FunFAN software manual).

- From the point of view of the optical properties we have shown that the loadings can be very different despite a similar dust optical depth. This points to an analysis on how the optical properties of dust are implemented to have a better convergence. First at the level of the refraction index but also the full set of hypothesis that explain the different optical parameters.

What observations can support further progress? The paper uses existing observations very well. I was struck by the absence of aircraft data, which seems to imply that all those expensive aircraft campaigns dedicated to mineral dust do not measure the quantities that are needed to improve models.

We absolutely agree about the important role of the aircraft data for mineral dust (and also other aerosols). The absence of a comparison with aircraft measurements doesn't mean, from our side, that they are not useful to improve the models. It can be actually the opposite, these measurements deserve a specific study about the representation of the vertical structure of dust in the models (with satellite soundings and aircraft measurements). Note that we also did not compare with specific measurements of wet vs dry deposition flux, or fine vs coarse optical depth. Both are important and useful, and should be part of future research.

The authors have the opportunity to say what those quantities are: size distribution, clearly – with the need to go beyond case studies and constrain the climatology. Mass extinction efficiency looks important too. Something else?

In the case of dust, and looking to aircraft measurements a key point (beyond those already commented by the reviewer) is the collection of mineralogy samples. First, it will provide information to understand the divergences in terms of optical properties beyond the size distribution. Specific minerals also produce indirect effects like those of heterogeneous chemistry, or cloud droplets formation for instance. The shape of the particles is important to quantify uncertainties in lifetime. Finally, we need more studies about the mineral fractions per size bin to understand better future paths in modelling of the largest particles.

**2 Other comments**

- Page 1, line 8: "uncertainty": "diversity" would be preferable because it is unlikely that 5 models sample the full uncertainty range.

  We agree with the terminology *diversity*, we will use it along the paper. Thank you.

- Page 1, line 9: how many models in that subset?

  There are 6 different models/dust schemes, with two models including explicitly the largest particles. Therefore, this subset has 4 models (6 minus 2). We indicate it better.

- Page 1, line 10: "better consistency between models": all models, or the subset?

  For all models in PDN experiment. We have clarified it better.

- Page 1, lines 14-17: The abstract needs to say what the conclusions of these two tasks were.

  **We added**: The global localization of source regions is correlated with MODIS, but the actual time-series per region has a diversity of values per model and differences with observations.

- Page 2, line 23: Could say that the estimate by Kok et al. (2017) comes from observations and models.

  Thank you for the comment, we added this information.

- Page 2, line 29: Could note that the impact of mineral dust on the phosphorus budget of the Amazon may be smaller than previously thought, based on Prospero et al. 2020

  Thank you for the comment, we added this information.

- Page 4, lines 21-22: What are the differences reported by Yu et al. 2019 due to?

We think that the differences should be related with different dust activation processes, in the case of Taklamakan previous studies like Ge et al. (2016) proposed an important role of nocturnal low-level jets. This explains the seasonal differences in the frequency of dust events between Taklamakan and Gobi deserts.

We have improved the text in the main paper with:
Recently, Yu et al. (2019) reported differences in the frequency of dust events between the Gobi (very high frequency of dust events in March and April) and Taklamakan (more than half of the events from May to September) deserts, which can be interpreted by a larger role in dust activation of the nocturnal low-level jet in Taklamakan Ge et al. (2016).

- Page 6, line 21: "non-mixed": it is more usual to say "externally mixed"

Thank you, we agree, we have changed the "non-mixed" to "externally mixed".

- Page 6, line 23: what experiments?

We are speaking about Denjean et al. (2016), Ryder et al. (2018) and Ryder et al. (2019). We have improved the sentence.

- Page 7, lines 24-26: That seems to be an example of the processes mentioned in line 20, so could be moved there.

Thank you. We have followed your comment and reordered the paragraph, and added also a reference recommended by other reviewer.

- Page 7, line 27: The information in Section 2.1 would be better described by a table of experiments.

> Thank you. We have added a new table to summarise the model experiments. Here it is also shown:
>
> CRESCENDO-ESM experiments analysed: PD (Present Day), PDN (Present Day with nudged winds), PI (Pre-Industrial aerosol and chemistry forcings). The sea-surface temperatures (SSTs) and ice cover are prescribed based on CMIP6-DECK-AMIP (Durack and Taylor, 2018). The solar forcing is using the input4MIPs dataset (Matthes et al., 2017) but NorESM uses the previous dataset. The gas and aerosol emissions are consistent with CMIP6 but depending on the complexity of the gas-phase species, ozone can be prescribed with either ozone concentrations from a previous full chemistry simulation or the input4MIPs ozone forcing dataset (Checa-Garcia et al., 2018; Hegglin et al., 2016). Wind fields used for the specified dynamics are obtained from re-analysis of ERA-Interim (Dee et al., 2011)
>
> | | PD | PDN | PI |
> |---|---|---|---|
> | Time Period | 2000-2014 | 2000-2014 | 2000-2014 |
> | SST and ice cover | prescribed | prescribed | prescribed |
> | Aerosol Precursors | Present-Day | Present-Day | 1850 |
> | Anthropogenic Emissions | Present-Day | Present-Day | 1850 |
> | Solar Forcing | Present-Day | Present-Day | Present-Day |
> | Wind Fields | modelled | prescribed | modelled |

- Page 11, Table 3: the units of MEE are given as $m2g - 1$ in Table 5. It would be good to harmonise that.

> We aimed to include the units of the original CRESCENDO diagnostics or derived in SI, but we have added a note to clarify this difference with Table 5. Thank you.

- Page 12, line 4: What is meant by "along the seasonal cycle"?

> We have changed that by "all the months of the year". This means that the relation $\tau_{440}^{dust} > 0.5\tau_{440}^{all-aer}$ should be true for all the models, and for each model for all the months of the year.

- Page 14, lines 29-32: The low regard given by the author to Pearson correlation is surprising since that measure is used extensively throughout the paper. I suggest toning down that statement or clarifying that it only applies to specific comparisons.

> It is not easy to tone down more the description about Pearson correlation as they are mathematical properties. But we have clarified that the reason to compare with another correlation estimator is that we can not show the scatter-plots of the involved variables to visually ascertain the performance of the statistic used.

> Thank you for the comment. **We have changed our previous sentence by**: *Given that this statistics is not robust and only representative of linear relationships, the skill is also estimated based on the Spearman rank correlation.*

- Page 19, line 27: "being the only model" – is that CNRM-6DU?

> Yes. We have been more explicit.

- Page 23, lines 5-6: But does CNRM-6DU match the Adebiyi and Kok (2020) estimates for the right reasons? Adebiyi and Kok (2020) estimate the burden of coarse-mode (larger than 5 microns) dust to be 17 Tg. Does the model also match that number?

> Here we refer only to total values, we did not evaluate per-bin differences (coarse vs fine modes).

- Page 28, line 25: Is that so remarkable? The models must prescribe fairly similar soil properties.

> We agree that this property is expected, but we considered it worth to be mentioned. It is true that it seems that all models are prescribing similar soil properties in Bodélé, but not in other regions like Australia and several Asian regions. So we highlighted the agreements by indicating also those regions with fair consistency (Bodélé).

- Page 30, lines 11-13: Did the CNRM model do something specific to represent Hoggar emissions?

> Not specifically, it is a result of the dust source information implemented.

- Page 45, lines 25-30: What about the LW? It would probably be the other way around, so there should be cancellation of error in size distribution between the two spectra.

> This is a question that we asked ourselves. However, the estimate of DRE in the LW done by the RRTM (the radiative transfer model used in our calculations) is not including the LW scattering (only absorption) therefore we considered it better to not conclude about LW and SW error cancellation explicitly as we can't support it with calculations. We have added a text to explain this important point.

> **We added (at the end of Section 5.1)**: It is important to note that the DRE shown in Table 6 is estimated without scattering in the LW (only absorption). In the case of mineral dust to neglect the LW scattering implies an underestimation of TOA-DRE-LW (Dufresne et al, 2020), mostly in cloud conditions.

- Page 44 line 31: What is the difference in terms of content between section 6 Discussion and section 7 Conclusion? They seem to both be a mix of summary

and further discussion, so could be merged.

> Thank you. We have refactored both sections, and we have followed the advice to merge both discussion and conclusions. We have extracted the recommendations for future research to have a specific section.

- Page 45 line 6-8: Where has the discussion on effective erodibility taken place? It is the first time the paper mentions that concept.

> We have reorganized the discussion and conclusion sections of the paper and improved the cross-references in the paper. As commented before we introduced this concept to interpret the normalised emissions maps. As far as, we have identical wind-fields in the wind-nudged simulations, we consider that these maps highlight differences in terms of mixed soil erodibility properties: soil properties like texture, surface roughness length, bare soil fraction, area efficiency factors etc. We named this dust effective soil erodibility information (DESEI) as we can not separate explicitly each component.

**3 Technical comments**

- Page 3, caption of Table 1: extra word "of about"

> It is corrected. Thank you.

- Page 3, line 20: "indicates" -> "indicate"

> It is corrected. Thank you.

- Page 42, line 10: "correspond at" -> "correspond to"

> It is corrected. Thank you.
- Page 44, lines 14-15: What is meant in the part starting with "although with"?

  It is Corrected. Added ','

- Page 47, line 48: typo: "an scarcity"

  Corrected by "a scarcity". Thank you.

- Page 47, line 12: Rephrase "which resulted to be challenging"

  We improved the sentence: resulted -> improved. Thank you.

- Page 50: Grammar of the last sentence of the acknowledgment could be improved.

  We improved the sentence. Thank you.

**References**

J. M. Ge, H. Liu, J. Huang, and Q. Fu. Taklimakan desert nocturnal low-level jet: climatology and dust activity. *Atmospheric Chemistry and Physics*, 16(12):7773–7783, 2016. doi: 10.5194/acp-16-7773-2016. URL https://acp.copernicus.org/articles/16/7773/2016/.

Y. Yu, O. V. Kalashnikova, M. J. Garay, and M. Notaro. Climatology of asian dust activation and transport potential based on misr satellite observations and trajectory analysis. *Atmospheric Chemistry and Physics*, 19(1):363–378, 2019. doi: 10.5194/acp-19-363-2019. URL https://www.atmos-chem-phys.net/19/363/2019/.

C. Denjean, F. Cassola, A. Mazzino, S. Triquet, S. Chevaillier, N. Grand, T. Bourrianne, G. Momboisse, K. Sellegri, A. Schwarzenbock, E. Freney, M. Mallet, and P. Formenti. Size distribution and optical properties of mineral dust aerosols transported in the western mediterranean. *Atmospheric Chemistry and Physics*, 16(2):1081–1104, 2016. doi: 10.5194/acp-16-1081-2016. URL https://www.atmos-chem-phys.net/16/1081/2016/.

C. L. Ryder, F. Marenco, J. K. Brooke, V. Estelles, R. Cotton, P. Formenti, J. B. McQuaid, H. C. Price, D. Liu, P. Ausset, P. D. Rosenberg, J. W. Taylor, T. Choularton, K. Bower, H. Coe,

Interactive
comment

M. Gallagher, J. Crosier, G. Lloyd, E. J. Highwood, and B. J. Murray. Coarse-mode mineral dust size distributions, composition and optical properties from aer-d aircraft measurements over the tropical eastern atlantic. *Atmospheric Chemistry and Physics*, 18(23):17225–17257, 2018. doi: 10.5194/acp-18-17225-2018. URL https://www.atmos-chem-phys.net/18/17225/2018/.

C. L. Ryder, E. J. Highwood, A. Walser, P. Seibert, A. Philipp, and B. Weinzierl. Coarse and giant particles are ubiquitous in saharan dust export regions and are radiatively significant over the sahara. *Atmospheric Chemistry and Physics*, 19(24):15353–15376, 2019. doi: 10.5194/acp-19-15353-2019. URL https://acp.copernicus.org/articles/19/15353/2019/.

Paul J. Durack and Karl E. Taylor. Pcmdi amip sst and sea-ice boundary conditions version 1.1.4, 2018. URL https://doi.org/10.22033/ESGF/input4MIPs.2204.

Katja Matthes, Bernd Funke, Tim Kruschke, and Sebastian Wahl. input4mips.solaris-heppa.solar.cmip.solaris-heppa-3-2, 2017. URL https://doi.org/10.22033/ESGF/input4MIPs.1122.

Ramiro Checa-Garcia, Michaela I. Hegglin, Douglas Kinnison, David A. Plummer, and Keith P. Shine. Historical tropospheric and stratospheric ozone radiative forcing using the cmip6 database. *Geophysical Research Letters*, 45(7):3264–3273, 2018. doi: 10.1002/2017GL076770. URL https://agupubs.onlinelibrary.wiley.com/doi/abs/10.1002/2017GL076770.

Michaela Hegglin, Douglas Kinnison, Jean-Francois Lamarque, and David Plummer. Ccmi ozone in support of cmip6 - version 1.0, 2016. URL https://doi.org/10.22033/ESGF/input4MIPs.1115.

D. P. Dee, S. M. Uppala, A. J. Simmons, P. Berrisford, P. Poli, S. Kobayashi, U. Andrae, M. A. Balmaseda, G. Balsamo, P. Bauer, P. Bechtold, A. C. M. Beljaars, L. van de Berg, J. Bidlot, N. Bormann, C. Delsol, R. Dragani, M. Fuentes, A. J. Geer, L. Haimberger, S. B. Healy, H. Hersbach, E. V. Hólm, L. Isaksen, P. Kållberg, M. Köhler, M. Matricardi, A. P. McNally, B. M. Monge-Sanz, J.-J. Morcrette, B.-K. Park, C. Peubey, P. de Rosnay, C. Tavolato, J.-N. Thépaut, and F. Vitart. The era-interim reanalysis: configuration and performance of the data assimilation system. *Quarterly Journal of the Royal Meteorological Society*, 137(656):553–597, 2011. doi: https://doi.org/10.1002/qj.828. URL https://rmets.onlinelibrary.wiley.com/doi/abs/10.1002/qj.828.

---

## Author Comment (AC2) · 14 Mar 2021

We thank the referee for the comments and questions. They help us to improve our manuscript, in particular those aspects related to improve the readability. Here we are indicating our answers in boxed frames after each point raised by the reviewer and our changes/actions in the manuscript within a green colour box.
**INFORMATION**: The Table 6 has been double-checked by the different modelling groups. CNRM reported that, instead of our previous estimate, their diagnostics of dry deposition are not including sedimentation which means different values of total and dry deposition (without sedimentation). With this revision the CNRM-6DU model has a larger bias due to an unclosed budget but the CNRM-3DU decreases the previous bias by a factor 2. In this situation we have removed the model CNRM-6DU from the multi-model mean, but we kept the CNRM-3DU. Given scale of the differences between models and observations, the comparison of total deposition draws the same conclusions and the results are very similar. Because the dust emission scheme is not affected by the bias, we kept their results in the analysis. All the Tables and Figures has been revised, and several of them improved according to the new information.

In this manuscript, the authors examined the simulated dust cycle by five Earth System Models. They compared the simulated dust dry and wet depositions, dust surface concentrations, and dust optical depths against measurements across the world at both annual and seasonal scales. Their results confirmed what's known to the dust research community, including that (i) the cutoff maximum dust size is important to the dust emission magnitude, and (ii) the ratio of dry to wet depositions is highly divergent between models. Furthermore, the authors found what's less known to the dust research community, including that (i) using identical nudged winds among models can improve the consistency between models in the dust cycle, and (ii) the divergent mass extinction efficiency between models explains why similar dust loads result in a large difference in optical depth.

I agree with Anonymous Referee 1 that the presentation quality needs to be improved. The authors offered extensive and interesting results (in both the main text and the supplement). However, clear and compact leading and ending sentences per paragraph are missing. This could give readers an excuse to stop reading, and thus decrease the impact of the manuscript.

> Thank you for the suggestion. We have improved the flow in several sections, in particular the section of discussion and conclusion, that now are a single new section. We have separated the recommendations to modellers about future research to be a single section because this is one of the specific objectives of the CRESCENDO project.

- Throughout the manuscript, the West Pacific and the East Pacific are defined problematically. To me, the West Pacific is the side where East Asia, Southeast Asia, and Australia are, and the East Pacific is the side where North and South Americas are located. However, the authors treated them reversely (see Fig. 1a as an example). This mismatch concerns me a lot, and can cause unnecessary misunderstandings in future studies. I suggest the authors correct the two regions throughout the manuscript systematically.

> Thank you for pointing us to this nomenclature aspect.

> We have changed the West vs East Pacific Ocean nomenclature across the paper: figures, tables and main text in the manuscript text.

- The cutoff maximum dust diameters of the seven models need to be better presented. First, the maximum diameters of the 4 modal models (in Table S.MD.9) are missing. Second, the maximum sizes of the 3 sectional models (Table S.MD.8) do not match Page 19 lines 10-15. For example, in Table S.MD.8, the maximum diameter of CNRM-6DU is 100 $\mu$m, however, in Page 19 line 12, the maximum diameter is 50 $\mu$m. A similar issue exists in CNRM-3DU. Since maximum diameters are critical to this manuscript, I suggest the authors address these two issues, and make the maximum diameters very clear in Section 2.

> **About the Page 19**: There is a mismatch of values between Page 19 line 12 and Table S.MD.8 due to radius vs diameter (thank you for detecting this issue).

**About the other points**: In the case of models with modal aerosol particle size distributions it is not easy to unambiguously define a cut-off. Some models implement a cut-off because internally they have a sectional dust emission scheme which is mapped into a modal aerosol scheme. In these cases, it is commented in the main text when we described the dust emission scheme. Other models, like IPSL models, don't implement any kind of cut-off. They have a very low probability to find the largest particles in the tail of the log-normal distribution that, in practical terms, implies modelling or not large particles. We have clarified these aspects in the caption of Table S.MD.9. In this situation we think that might be misleading to give one single number to describe the modelling of the largest particles.

**Added to manuscript**: In order to resolve the important comment of the reviewer. We have added in the Table 1 additional information about the modelling of large particles. Explicitly, we have defined two classifiers:

1. If the dust scheme aims to model or not particles with diameters larger than $10\mu m$.

2. If specific separated (or mixed) modelling of particles larger than $20\mu m$ are included.

The aim is to capture better the differences of our ensemble. For example, CNRM-6DU is modelling particles up to $100\mu m$ as the last bin is covering those diameters, however with a single bin from $10\mu m$ to $100\mu m$. It is possible to argue that UKESM1 is modelling more explicitly large particles because the last bin is from $20\mu m$ to $62\mu m$ even if the upper-threshold is smaller. We think that two classifiers are more descriptive than a single number. In the case of CNRM-6DU they will be (yes, mix) and for UKESM (yes, yes).

- In Page 35 lines 8-10, the authors compared simulated deposition flux in Asia, and implicitly indicating that EC-Earth is better than all the other models. However, the tricky thing is that there is only one station in the Asian region (as seen in Fig. 1a). (Similarly, there is only one station in South Atlantic, South America, and Egypt). The sample number is too small to draw a significant conclusion for a continent. Thus, I suggest the authors make it clear that the sample number is one, and add the numbers of observational stations at all the regions in the legends of Figs. 8, 9, and 13.

We agree with the reviewer in this aspect. We have commented about the low number of measurements in several regions of the Earth. In the paragraph of Page 35, we mentioned *Asia station* to refer to a single station but not the whole region. We used the term "Asia" because it might be more easy for the reader to follow the location of the station. But we agree that this can lead to misunderstandings.

We have replaced the *Asia station* by *Asia single station*. We have done the same for South Atlantic station.

- Recent progress in dust shape and its impact on dry deposition needs to be added in Page 7 lines 16-26. Jasper Kok's group has a recent paper (Huang et al., 2020) that compiled 27 measurements of realistic dust shape worldwide (including Li and Osada, 2007). They find that dust asphericity increases gravitational settling lifetime by 20% all sizes.

Thank you for the references. They are very relevant for our discussion.

We have added these references, and we have discussed this important point.

- Recent progress in dust cycle needs to be added. A recent paper (Kok et al., 2020) that diagnosed the dust cycle is very similar to this manuscript, but used different models.

> Thank you for the reference.

> **We have included a new paragraph it in the discussion and conclusion section**: The range of dust loadings that we obtained is smaller than recent estimations Kok et al. (2021) that propose values $\gtrsim$ 20 Tg with a multi-model comparison with models with geometric diameters up to $20\mu m$ but based on a new methodology where the dust diagnostics are including observational constraints Kok et al. (2020). Actually, Adebiyi and Kok (2020) propose that the total load of dust in the atmosphere is higher than what is estimated typically, and give a mean value close to 30 Tg, where the contribution of coarse mode is more important than the fine mode.

**1 Minor comments**

- Typo in Page 6 line 33: correct "clay/silk" to "clay/silt"

  > Corrected. Thank you.

- Typo in Table 3: the unit of grid cell area should be $m^2$ instead of kg

  > Corrected. Thank you.

- Typo in Page 12 line 7: correct "19 stations" to "18 stations"

  > Corrected. Thank you.

- Table 7 only offers emissions of 14 of the 16 regions. Regions "Mid-Atlantic" and "Sahel/Gulf of Guinea" are missing and should be added. Note that after adding the two regions, authors need to update the order of the top 10 regions with dust emission intensity.

  Thank you for this comment. We agree that a clarification is needed. It has been a typographical error from our side to name the region as "Sahel/Gulf of Guinea" as the top latitude for this region is 9N. Therefore the region is not covering the Sahel which would begin at 10N. Both regions, Gulf of Guinea and Mid-Atlantic don't have an important role in emissions, so we initially excluded them from the table.

  **We have improved the text with**:
    – The emissions from Gulf of Guinea have been added to the table (they don't change at all any of the results either the order or the analysis).
    – The order of the regions by emissions has been double-checked.
    – The values of Mid-Atlantic are zero as it is not including dust sources (only ocean). We added a note in the Table.

- Typos exist in Table 7's order of the top 10 regions. For instance, for EC-Earth, there are two 4th largest sources (i.e., the North Sahara and the Taklamakan), which are clearly problematic. Typos also exist in models CNRM-3DU and IPSL.

  Thank you. We have introduced some errors when reformatting the table for final submission. We have now revised and corrected the Table 7.

**References**

J. F. Kok, A. A. Adebiyi, S. Albani, Y. Balkanski, R. Checa-Garcia, M. Chin, P. R. Colarco, D. S. Hamilton, Y. Huang, A. Ito, M. Klose, L. Li, N. M. Mahowald, R. L. Miller, V. Obiso, C. Pérez García-Pando, A. Rocha-Lima, and J. S. Wan. Contribution of the world's main dust source regions to the global cycle of desert dust. *Atmospheric Chemistry and Physics Discussions*, 2021:1–34, 2021. doi: 10.5194/acp-2021-4. URL https://acp.copernicus.org/preprints/acp-2021-4/.

J. F. Kok, A. A. Adebiyi, S. Albani, Y. Balkanski, R. Checa-Garcia, M. Chin, P. R. Colarco, D. S. Hamilton, Y. Huang, A. Ito, M. Klose, D. M. Leung, L. Li, N. M. Mahowald, R. L. Miller, V. Obiso, C. Pérez García-Pando, A. Rocha-Lima, J. S. Wan, and C. A. Whicker. Improved representation of the global dust cycle using observational constraints on dust properties and abundance. *Atmospheric Chemistry and Physics Discussions*, 2020:1–45, 2020. doi: 10.5194/acp-2020-1131. URL https://acp.copernicus.org/preprints/acp-2020-1131/.

Adeyemi A. Adebiyi and Jasper F. Kok. Climate models miss most of the coarse dust in the atmosphere. *Science Advances*, 6(15):eaaz9507, apr 2020. doi: 10.1126/sciadv.aaz9507. URL https://doi.org/10.1126/sciadv.aaz9507.

---

## Author Comment (AC3) · 14 Mar 2021

We thank the referee for comments and questions. They help us to explain better several aspects of the paper. Here we are indicating our answers in boxed frames after each point raised by the reviewer and our changes/actions in the manuscript within a green colour box.
**INFORMATION**: The Table 6 has been double-checked by the different modelling groups. CNRM reported that, instead of our previous estimate, their diagnostics of dry deposition are not including sedimentation which means different values of total and dry deposition (without sedimentation). With this revision the CNRM-6DU model has a larger bias due to an unclosed budget but the CNRM-3DU decreases the previous bias by a factor 2. In this situation we have removed the model CNRM-6DU from the multi-model mean, but we kept the CNRM-3DU. Given scale of the differences between models and observations, the comparison of total deposition draws the same conclusions and the results are very similar. Because the dust emission scheme is not affected by the bias, we kept their results in the analysis. All the Tables and Figures has been revised, and several of them improved according to the new information.

**1  General comments**

This manuscript presents the results of five Earth System Models simulations of the global dust cycle, emissions, dry and wet deposition, optical depths, and surface concentrations comparatively to satellites and in situ observations. The authors explore global and regional variability between models in three different simulated experiments: PD (calculated winds), PDN (reanalysed winds), and PI (prescribed chemistry and aerosols). Overall, the manuscript is well written and provides ample content. Having said that, this manuscript is quite extensive and important information is left for the reader to find in the supplement. The content of this manuscript could be divided in two different publications. In the first one, you could explore the differences between the five models, and then, in the second, you could explore more deeply the differences between the three simulated experiment scenarios.
The reason to have a single publication for the mineral dust evaluation is that we have a publication plan within CRESCENDO-ESMs: one study per aerosol species. This also explains why we are including here the PI experiment (with prescribed chemistry and aerosols precursor emissions to pre-industrial values). In the case of mineral dust, we don't expect important differences between PI and PD experiments. For other aerosols however, we expect larger differences, and within this scope it is convenient to compare all three experiments systematically. As commented to the first reviewer, other analysis based on diagnostics per bin-size, vertical structure of dust concentrations or long dust transport have not been added to this study although a few of these results have been shown in conferences already.

At least, I would include a figure of the particle size distributions used in each model into the main text.

During the manuscript preparation we considered representing the dust particle size distribution (DPSD) as suggested by the reviewer. However, the shape of DPSD depends on the localization of the grid cell of the model and the time. We have two options: a representation at emission at specific locations, or a kind of global *mean* particle size distribution.

- For the first option, we have to deal with the several kinds of dust emission schemes. For those based in the brittle fragmentation, it is possible to have a representation of a normalised size distribution (like Figure 1 in Di Biagio et al. (2020)), as far as, the different modes have a prescribed mixing factor. However, several models proceed with a sectional emission scheme that is mapped into a multimodal log-normal size distribution, and it is not easy to have a unique multimodal dust size distribution at emission to compare with.

- For the second option we did not find a suitable average in space and time that can help us in the discussion.

For this reason we have decided to include a detailed description of the several dust emission schemes rather than plot a qualitative DPSD.

We have introduced two classifiers related with the modelling of coarse/large particles to improve the description and discussion of those aspects related with the DPSD (more information in our answers to reviewer 2).

**2   Specific Comments**

- Page 6, Line 7-31: For this part of the text, it would be very useful to have a plot overlapping the particle size distributions in each of the 5 models. This would be similar to what you have in the bottom panel of Figure 4.

  As commented above we did not find how to include a plot of the overlapping DPSD without being specific of a location at emission, or without defining an average method that can not be representative of the several microphysical processes in different regions.

- Page 7, Line 11: "Therefore those optical properties are representative for the global mineralogical composition rather than a description of the soil-type dependence of the mineralogy that would imply local differences on emitted optical properties." The point you are raisin here is important, but it is still not clear what optical properties you have actually used. For instance, do all five models use exactly the same spectral complex refractive indices? Which databases/references are you using in each model?

Thanks for this comment. We agree that this information is important in the paper. In CRESCENDO each model is using a different refractive index. Also, each model implements slight differences in the pre-computation of lookup tables for each optical parameter.

**We have added this information in the Table 1.**

- Page 3, Table 1: It would be useful for the reader to include the specific particle sizes ranges used in each model simulation in column DPSD. Alternatively, you could move Tables S.MD.8 and S.MD.9 from the supplement to the main text. Please, also include the meaning of PD, PDN, and PI in the title of Table 1.

Thank you for pointing it out. We have included new information in the main manuscript and finally the Tables S.MD.8 and S.MD.9 remain in the Supplementary information for further reference.

**We have added**:
  – Two columns in Table 1 to classify the modelling of large dust particles by the different models.
  – The meaning of PD, PDN and PI in the Table caption.
  – A new table with the information of the several model experiments.

- Page 8, Line 2: "A last simulation where aerosols and chemistry emissions are prescribed for 1850 (named PI)". Why? Could you add a few words explaining why such simulation is relevant and how do you use its results specifically in this study, covering the years between 2000 and 2014?

We have explained the initial motivation above. For mineral dust the hypothesis to be tested is that the differences between PD and PI are small and the general behaviour is the same. Any kind of indirect effect is much smaller than the role of wind fields (PD vs PDN). If, as a consequence of prescribed emissions at 1850, we have slight differences in clouds or precipitation this potentially could have a slight effect on dust cycle. Our comparison suggests that those effects are not conditioning the global behaviour of dust global cycle.

**We have added to Section 2.1**: The comparison between the PD and PDN experiments inform about the role of wind fields to explain model diversity. The difference between PD and PI dust emissions allow us to evaluate whether the effects in the climate system due to non-dust emissions have a discernible impact on the global dust cycle (as both PD and PI have been prescribed with the same SSTs). A summary of the properties of the model experiments is given in Table 2.

- Page 12, Line 3: Could you please clarify the criterion used to select optical depths? What does "all − aer" mean here?

Thank you. $\tau^{all-aer}$ means the total optical depth of all aerosols in contrast with $\tau^{dust}$ that is only for dust. We have added this information in the main text.

> **We added a better description in the main text**:
> $\tau_{440}^{dust} > 0.5\tau_{440}^{all-aer}$ for all models and all the months of the year, where $\tau_{440}^{all-aer}$ refers to optical depth at 440 nm of all aerosols and $\tau_{440}^{dust}$ is the optical depth of mineral dust aerosols at 440 nm.

- Page 15, Figure 4: What do you mean by "samples are the marks on x-axis"? Does the "sample" correspond to a given year? Which are the years you consider here? Include the time-period in the figure caption.

> Thank you for this comment. We improved the sentence. In the Figure 4, each sample (the value for a year) is represented by grey-dots (for the top panel) and by coloured-marks just over the x-axis. The idea is that each year is a sample, and the plot is more informative with these dots and marks.

> **We improved the caption by explaining that**:
> - The grey-dots (top-panel) over the box-plot represent each of the annual values.
> - In the bottom panels our sample values per model are represented by the coloured vertical marks just above the x-axis.
> - For both, the models and the observations (MISR and MODIS), the estimates are for time-period 2000-2014.

- Page 22, Line 21: Could you clarify if the model simulations were sampled at MODIS and MISR times in Figure 4? What is the main reason for having UKESM's AOD so high?

Thank you for this comment. Yes, the years are the same for MODIS and MISR in Figure 4 bottom right panel. We have added this information in the caption. In the context of the paper, we know that the large values of UKESM are not due to mineral dust, so it is related with other aerosol species. In principle, most of the models (in other multi-model evaluations like AEROCOM) show global values of aerosol optical depth smaller than MODIS. However, the reason for that is an active research topic at this moment. In our figure the estimates of AOD are also in general smaller than MISR. The best scenario is a progressive convergence between models, but also between observations. Although only with two satellite platforms, the Figure 4 also suggests that the diversity of total aerosols optical depths is larger in models than in observations. The inter-annual variability on AOD by MODIS is also the largest one.

**We have added to the Section 5.1**: *The bottom right panel of Figure 4 indicates model discrepancies in the magnitude of the inter-annual variability (as measured by the width of the distribution) and an overall underestimation of AOD at 550 nm with respect to satellite platforms.*

- Page 22, Line 29: Do you know why the EC-Earth and the NorESM have MEE values that differ from the other models? If so, I suggest you to discuss the main reasons in the text.

> We think that MEE values of the EC-Earth and the NorESM differ (from other models) by a combination of factors. Both models have the lowest dust loadings, in the case of NorESM, also with the smallest inter-annual variability. In the EC-Earth the dust scheme has a cut-off at $8\mu m$ and NorESM an accumulation mode with smaller particles than the equivalent mode of IPSL or EC-Earth so there are differences in modelling of the particle size distribution. In the case of NorESM the imaginary part of the refractive index is also the largest.

- Page 23, Line 7 to 24: The plot overlapping all size distributions would again be helpful here.

> Thank you. We have introduced additional information in Table 1 and improved the text.

- Page 28, Line 7: "On the side the modelled wind surface friction velocity and speed agree better with actual meteorological conditions, and on the other side the description of the soil surface properties has become more accurate." These are important points. Could you provide references of experimental studies that support these two statements?

> Thank you. These are mostly general assessments about the improvement of the climate models in the boundary layer with respect to wind fields, and in the land surface. These aspects are not restricted to mineral dust science, and belong more to the evolution of climate models.

**We have added two additional references in our revised manuscript** (for more general discussions good references are: (Shao, 2008; Knippertz and Stuut, 2014)):

On the one hand the modelled wind surface friction velocity and speed agree better with actual meteorological conditions, and on the other hand, the description of the soil surface properties has become more accurate due to both, improvements in the soil texture databases, and the use of satellite retrievals to better describe the roughness length, e.g Prigent et al. (2005); Menut et al. (2013).

• Page 30, Line 26: ". . . they indicate that although there are important differences between PD and PDN experiments in terms of total emissions, . . .". It is difficult to see those differences here.

Thank you. The important differences are in the global emissions, for example, CNRM-6DU has 3450 Tg/yr for the PD experiment and 1278 Tg/yr for the PDN experiment. However, these large differences in global emissions are not discernible in the *normalized* emission maps.

• Although it might be out of the scope of this paper, I think it would be interesting to comment on how the re-analyzed wind fields in PDN differ from the calculated wind fields in PD near dusty regions. Are there significant differences between the wind fields in PD and PDN? How much would be that difference?

> Thank you. This is an interesting comparison. We expect differences that explain the decrease on the model diversity of mineral dust when we have consistency for the wind fields. Several previous studies have been done for a single model. We think that the ideal situation is to compare at several temporal resolutions that can be relevant for dust related processes to actually explain the improvement in agreement between models for the diagnostics analysed. It is an interesting suggestion for future evaluations with global climate models, but the analysis would be too extensive to include in this already long paper.

- Page 35, Line 22: "All the other models underestimate total depositions fluxes over stations where fluxes exceed 100gm-2yr-1." What do you think is the main reason for that?

> This fact is probably correlated with the largest emissions of UKESM model and its modelling of particles with diameters larger than $20\mu m$.

> We added this interpretation to the discussion.

- Page 44, Line 7 to 30: Could you comment on the temporal resolution of the surface concentration observations? Are those monthly means based on continuous daily observations? Are these observations for one or the average value for multiple years? Include the years of the observations and the simulations in the figure caption.

There are few available observations of dust surface concentrations. The general treatment is to consider them as **climatological dataset** (although we know that it is an approximation) and compare their values with the values for our 15 years of simulation (from 2000 to 2014). The same hypothesis has been applied by Huneeus et al. (2011) and Albani and et al (2021). We further investigate possible discrepancies due to different sampling years of measurements. In this regard, we added those of the reference Cheng et al. (2008) to the Table S.MD.4 in the Supplementary information to visualize the role of annual sampling. We observe that measurements from different years are only slightly different but not discernible in the figures of the paper, and the conclusions are the same. The values are included in the table S.MD.4 in brackets. We only had access to the raw INDAAF dataset where the observations of PM10 are measured with high-temporal resolution (several measurements per day).

We refer better to Table S.MD.4 in brackets and explained the interpretation as a climatology dataset.

**3 Technical Corrections**

- Page 8, Figure 1: The aspect ratio of the Figures 1a and 1b seems strange.

Thank you. This figure has been improved.

- Page 14, Line 27: Replace "has" by "have".

Thank you. We have changed the text of the paragraph.

- Page 16, Line 25: Replace "in the main paper" by "this paper".

  > Thank you. We replaced the word "main" by "this".

- Page 17, Table 6: Explain the meaning of "in and out".

  > The definitions are in the Appendix A. But we have added a less mathematical and more visual definition in the Table 6: the *method in add* each specific mode to a case without any mode of dust, the *method out remove* that specific mode to a case with all the modes of dust. The DRE estimates give different values for each method.

- Page 33, Line 11: "Figures Dep.11 and Dep.12 show . . .". I cannot find these figures.

  > Thank you. We included the right names of the figures of the supplement.

- Page 44, Line 25: "sitation"

  > Thank you. It should be "situation".

**References**

C. Di Biagio, Y. Balkanski, S. Albani, O. Boucher, and P. Formenti. Direct radiative effect by mineral dust aerosols constrained by new microphysical and spectral optical data. *Geophysical Research Letters*, 47(2), January 2020. doi: 10.1029/2019gl086186. URL https://doi.org/10.1029/2019gl086186.

Yaping Shao. *Physics and Modelling Wind Erosion*, volume 23. Springer-Werlag, 01 2008. ISBN 1402088949.

Peter Knippertz and Jan-Berend W. Stuut, editors. *Mineral Dust - A Key Player in the Earth System*. Springer-Verlag, 2014. ISBN 978-94-017-8977-6. doi: 10.1007/978-94-017-8978-3.

Catherine Prigent, Ina Tegen, Filipe Aires, Béatrice Marticorena, and Merhez Zribi. Estimation of the aerodynamic roughness length in arid and semi-arid regions over the globe with the ers scatterometer. *Journal of Geophysical Research: Atmospheres*, 110(D9), 2005. doi: https://doi.org/10.1029/2004JD005370. URL https://agupubs.onlinelibrary.wiley.com/doi/abs/10.1029/2004JD005370.

Laurent Menut, Carlos Pérez, Karsten Haustein, Bertrand Bessagnet, Catherine Prigent, and Stéphane Alfaro. Impact of surface roughness and soil texture on mineral dust emission fluxes modeling. *Journal of Geophysical Research: Atmospheres*, 118(12):6505–6520, 2013. doi: https://doi.org/10.1002/jgrd.50313. URL https://agupubs.onlinelibrary.wiley.com/doi/abs/10.1002/jgrd.50313.

N. Huneeus, M. Schulz, Y. Balkanski, J. Griesfeller, J. Prospero, S. Kinne, S. Bauer, O. Boucher, M. Chin, F. Dentener, T. Diehl, R. Easter, D. Fillmore, S. Ghan, P. Ginoux, A. Grini, L. Horowitz, D. Koch, M. C. Krol, W. Landing, X. Liu, N. Mahowald, R. Miller, J.-J. Morcrette, G. Myhre, J. Penner, J. Perlwitz, P. Stier, T. Takemura, and C. S. Zender. Global dust model intercomparison in aerocom phase i. *Atmospheric Chemistry and Physics*, 11(15):7781–7816, 2011. doi: 10.5194/acp-11-7781-2011. URL https://www.atmos-chem-phys.net/11/7781/2011/.

Samuel. Albani and et al. The global dust cycle in the IPSL climate model revisited: particle size distributions and dependence of emissions on land surface properties. *in preparation*, 2021.

T. Cheng, Y. Peng, J. Feichter, and I. Tegen. An improvement on the dust emission scheme in the global aerosol-climate model ECHAM5-HAM. *Atmospheric Chemistry and Physics*, 8 (4):1105–1117, February 2008. doi: 10.5194/acp-8-1105-2008. URL https://doi.org/10.5194/acp-8-1105-2008.

---

## Author Response (AR2)

**Authors's reply to Editor**

**May 11, 2021**

We thank the editor for the last feedback, and we tried to resolve all the minor correction proposed.

**Publish subject to minor revisions**

Thank you for submitting your manuscript entitled "Evaluation of natural aerosols in CRESCENDO-ESMs: Mineral Dust" to Atmospheric Chemistry and Physics, and for your efforts in addressing the Referee comments, as well as the clear information regarding the update to the CNRM results. After reviewing your response to the Referee comments and tracked changes, my decision is "Publish subject to minor revisions".

While the Referee comments have largely been addressed, I would like to ask you to strengthen the response to a few of the comments raised. In particular:

1. The discussions on the topics of "How can model disagreement be resolved?" and "which observations can support further progress" (Referee 1) seem to me to be important. I would like to encourage you to consider whether some version of the comments you have provided in the review response could be fruitfully incorporated into the manuscript.

Several points had been added to the last section named **Future research directions**, but we agree that the point related with the comparisons based on dust particle size distribution has not been full clarified. So we have added the following text in the section 7:

This also illustrates that comparisons where the particle size distribution is resolved (comparisons based on the contributions of each DPSD bin) is needed to understand better the source of model discrepancies, in that regard CRESCENDO-ESMs simulations were designed with these future evaluations in mind. Also we created specific tools to estimate binned contributions from models based on modal DPSD to support these comparisons.

2. On the topic of the Pearson correlation, raised by Referee 1: to avoid the impression of a subjective value judgment, please make more explicit (in the manuscript text) what you mean by "not robust", or otherwise clarify this sentence.

Thank for the comment. We have improved the sentence and we have added a new reference where it is explained what we meant by "not robust". Now it reads:

Given that this statistic is not robust, because its instability in the presence of outliers [2], and only representative of linear relationships, the skill is also estimated based on the Spearman rank correlation to ensure the robustness of the results. For the other comparisons the scatter-plots are informative of the quality of the correlation estimator.

3. On the discussion of "why the EC-Earth and the NorESM have MEE values that differ from the other models", this discussion also contains useful information. Again, I would like to encourage you to consider whether it would be appropriate to include some of this information in the manuscript text.

We added few remarks to explain the possible reasons of these differences:

The larger MEE values of the EC-Earth and the NorESM models can be due a combination of factors: they have the lowest dust loadings, and both are not modelling particles larger than about 8  $\mu$ m, also in the case of NorESM the imaginary part of the refractive index is also the largest of all the models analysed.

4. Referee 3 had pointed out, in the comment on Page 28, Line 7, statements where additional referencing may be appropriate. The revised version of the text adds referencing and additional details for improvements to the description of soil properties. However, it is still not clear what evidence the statement "the modelled wind surface friction velocity and speed agree better with actual meteorological conditions" is based on. Please provide a source of information or otherwise clarify the basis for this statement.

This sentence relies on the general improvement of current models to reproduce the surface wind speed and the logical impact in the dust emission. There are aspects that contribute to improve the description of wind in the boundary layer, e.g the progressive increase of vertical levels close to surface, but to keep brief this part of the paper we have just included a new citation based on the book reference Knippertz and Stuut [1] where the role of the evolution/progress in modelling the dust schemes is commented.

5. Referee 3 comment on Page 30, Line 26: Adding a reference to Table 8 in this sentence would eliminate any potential for confusion by readers.

Thank you. We have added the reference to Table 8.

**References**

- [1] Peter Knippertz and Jan-Berend W. Stuut, eds. *Mineral Dust.* Springer Netherlands, 2014. DOI: 10.1007/978-94-017-8978-3. URL: https://doi.org/10.1007/978-94-017-8978-3.
- [2] Zh. V. Li, G. L. Shevlyakov, and V. I. Shin. "Robust estimation of a correlation coefficient for *ϵ*-contaminated bivariate normal distributions". In: *Automation and Remote Control* 67.12 (Dec. 2006), pp. 1940–1957. DOI: 10.1134/s0005117906120071. URL: https://doi.org/10.1134/s0005117906120071.